# Adversarial Training from Mean Field Perspective

**Soichiro Kumano**
The University of Tokyo
kumano@cvm.t.u-tokyo.ac.jp

**Hiroshi Kera**
Chiba University
kera@chiba-u.jp

**Toshihiko Yamasaki**
The University of Tokyo
yamasaki@cvm.t.u-tokyo.ac.jp

## Abstract

Although adversarial training is known to be effective against adversarial examples, training dynamics are not well understood. In this study, we present the first theoretical analysis of adversarial training in random deep neural networks without any assumptions on data distributions. We introduce a new theoretical framework based on mean field theory, which addresses the limitations of existing mean field-based approaches. Based on the framework, we derive the (empirically tight) upper bounds of $\ell_q$ norm-based adversarial loss with $\ell_p$ norm-based adversarial examples for various values of $p$ and $q$. Moreover, we prove that networks without shortcuts are generally not adversarially trainable and that adversarial training reduces network capacity. We also show that the network width alleviates these issues. Furthermore, the various impacts of input and output dimensions on the upper bounds and time evolution of weight variance are presented.

## 1 Introduction

Adversarial training [38, 58] is one of the most effective approaches against adversarial examples [89]. Various studies aimed to improve the performance of adversarial training [28, 94, 113], leading to numerous observations. Adversarial training improves accuracy for adversarial examples but decreases it for clean images [58, 88]. Moreover, it requires additional training data [14, 40, 77] and achieves high robust accuracy in a training dataset but not in a test dataset [58]. To improve the reliability of adversarial training and address the aforementioned challenges, a theoretical analysis is essential. Specifically, it is crucial to gain insight into network evolution during adversarial training, conditions for adversarial trainability, and differences between adversarial and standard training.

However, the theoretical understanding of network training is challenging, even for standard training, due to the non-convexity of loss surface and optimization stochasticity. Recent studies employed the mean field theory and analyzed the early stage of standard training for randomly initialized deep neural networks (random networks) [71, 81]. Some studies explored network trainability regarding gradient vanishing/explosion [81, 104, 105] and dynamical isometry [69, 79, 100]. Others examined network representation power [49, 71, 101]. The theoretical results of the early stage of training have been empirically observed to fit well with fully trained networks [25, 84], partially supported by recent theoretical results [46]. However, existing mean field-based approaches cannot manage the probabilistic properties of an entire network (e.g., the distribution of a network Jacobian) and dependence between network inputs and parameters, which are crucial for analyzing adversarial training.

In this study, we propose a mean field-based framework that addresses the aforementioned limitations (Thm 4.1), and apply it to the adversarial training analysis. While previous studies on adversarial

37th Conference on Neural Information Processing Systems (NeurIPS 2023).

training rely on strong assumptions (e.g., Gaussian data [14, 80] and linear classifiers [76, 107]), the proposed framework includes various scenarios (e.g., $\ell_p$ norm-based adversarial examples and deep neural networks with or without shortcuts, i.e., residual or vanilla networks) without any assumptions on data distributions. Our analysis reveals various adversarial training characteristics that have been only experimentally observed or are unknown. The results are summarized as follows.

**Upper bounds of adversarial loss.** We derive the upper bounds of adversarial loss, quantifying the adverse effect of adversarial examples for various combinations of $\ell_q$-adversarial loss with the $\ell_p$-norm $\epsilon$-ball ($p, q \in \{1, 2, \infty\}$) (Thm 5.1). Numerical experiments confirm the tightness of these bounds. We also investigate the impacts of input and output dimensions on these bounds, and discover that for the $(p, q) = (2, \infty)$ combination, the bound is independent of these dimensions.

**Time evolution of weight variance.** We present the time (training step) evolution of weight variance in training (Thms 5.4 and G.9). Weight variance has been used to assess training properties [49, 71, 81]. Our analysis indicates that adversarial training significantly regularizes weights and exhibits consistent weight dynamics across different norm choices.

**Vanilla networks are not adversarially trainable under mild conditions.** We show that gradient vanishing occurs in vanilla networks (without shortcuts) with large depths and small widths, making them untrainable via adversarial training, even with careful weight initialization (Thm 5.7). This contradicts standard training, where deep vanilla networks can be trained with proper initialization [81, 100]. However, residual networks are adversarially trainable *even without proper initialization* (Thm 5.8 and Prop G.10), proving the importance of shortcuts for adversarial training.

**Degradation of network capacity and role of network width.** As adversarial robustness requires high network capacity [63], deep networks can be used in adversarial training. However, we reveal that network capacity, measured based on the Fisher–Rao norm [56], sharply degrades in deep networks during adversarial training (Thms 5.9 and G.14). Specifically, we confirm that the capacity at training step $t$ is $\Theta(L - tL^2/N)$, where $L$ and $N$ denote the network depth and width, respectively. Interestingly, this result contrasts the roles of depth and width, i.e., the depth increases the initial capacity but decays it as training proceeds, whereas the width preserves it. While our theory is validated only during the initial stages of training, our experiments confirm that the adversarial robustness after full training is significantly influenced by network width (cf. Tabs. A5 and A6) as demonstrated in our theorems.

**Other contributions.** Furthermore, we show the followings cases: (a) Equality of the adversarial loss is obtained instead of inequality (upper bound) under several assumptions (Props G.2 and G.3). (b) Adversarial training leads to faster weight decay and less stable gradients compared with $\ell_2$ weight regularization. (c) Capacity degradation is discussed for metrics other than the Fisher–Rao norm. (d) Adversarial risk cannot be mitigated while maintaining trainability and expressivity. (e) Discussion on ReLU-like activations extends to Lipschitz continuous activations under certain conditions. (f) A single-gradient descent attack can find adversarial examples that flip the prediction of a binary classifier (Prop K.1). Contributions (b)–(f) are found in Appx. K.

Although this study focuses on adversarial training, our theoretical framework can be applied to other training methods that consider the probabilistic property of an entire network and dependence between network inputs and parameters. Consequently, we believe that this study can potentially contribute to the theoretical understanding of adversarial training and various deep learning methods.

## 2 Related work

Here, we summarize the full version in Appx. A. A technical discussion follows in Sec. 5.1.

**Mean field theory.** Mean field theory in machine learning investigates the training dynamics of random networks in chaotic and ordered phases [71]. Networks can be trained at the boundary between these phases [81]. The theory has been extended to networks with shortcuts [105, 106], recurrent connections [15, 69], and batch normalization [104]. It has been employed to study dynamical isometry [15, 69, 70, 79], and a subsequent study achieved training of 10,000-layer networks without

shortcuts [100]. Moreover, the theory has been applied to analyze network representation power [49, 71, 101]. However, existing mean field-based analysis cannot handle the properties of an entire network and input–parameter dependence, which is a drawback for some deep learning methods, e.g., adversarial training. Thus, we propose a new framework to address these limitations.

**Adversarial training.** Various questions related to adversarial training have been theoretically addressed by some studies, including the robustness-accuracy trade-off [29, 47, 75, 76, 92, 113], generalization gap [6, 50, 102, 107], sample complexity [1, 14, 62, 80, 110], large model requirement [63], and enhanced transfer learning performance [27]. However, these results are obtained in limited settings (e.g., Gaussian data and linear classifiers) and are not easily extended to deep neural networks or realistic data distributions. To explore more general settings, recent studies used the neural tangent kernel theory [4, 46, 55]. In the kernel regime, adversarial training, even with a heuristic attack, finds a robust network [35, 115]. In our study, we investigate adversarial training dynamics based on a mean field perspective, covering general multilayered networks with or without shortcuts and without assumptions about data distributions.

# 3 Preliminaries

## 3.1 Setting

Notations are summarized in Tab. A2. For an integer $n \in \mathbb{N}$, let $[n] := \{1, \ldots, n\}$. In this study, we focus on random deep neural networks with ReLU-like activations, called random ReLU-like networks. This is formally defined as follows:

**Definition 3.1** (ReLU-like network). A network is called a ReLU-like network if all its activation functions are $\phi(z) := uz$ for $z \geq 0$ and $vz$ for $z < 0$, with $u, v \in \mathbb{R}$.

ReLU-like activations [33, 57] are widely used in theoretical and practical applications [42, 45, 53, 85, 109]. In Appx. K, we extend our theorems to networks with Lipschitz continuous activations.

A ReLU-like network, $\boldsymbol{f} : \mathbb{R}^d \to \mathbb{R}^K$, comprises $L \in \mathbb{N}$ trainable layers and two non-trainable layers for adjusting input and output dimensions. The input layer projects $\boldsymbol{x}^{\mathrm{in}} \in \mathbb{R}^d$ to an $N$-dimensional vector $\boldsymbol{x}^{(0)} \in \mathbb{R}^N$ using the random matrix $\boldsymbol{P}^{\mathrm{in}} \in \mathbb{R}^{N \times d}$. Subsequently, $L$ consecutive affine transformations and activations are applied by $\boldsymbol{g} : \mathbb{R}^N \to \mathbb{R}^N$. Then, $\boldsymbol{g}(\boldsymbol{x}^{(0)})$ is multiplied by a random matrix $\boldsymbol{P}^{\mathrm{out}} \in \mathbb{R}^{K \times N}$ to obtain the output vector $\boldsymbol{f}(\boldsymbol{x}^{\mathrm{in}})$. Finally, the network function is provided by $\boldsymbol{f}(\boldsymbol{x}^{\mathrm{in}}) := \boldsymbol{P}^{\mathrm{out}} \boldsymbol{g}(\boldsymbol{P}^{\mathrm{in}} \boldsymbol{x}^{\mathrm{in}})$. We assume that $d$ and $K$ are sufficiently large, and each entry of $\boldsymbol{P}^{\mathrm{in}}$ and $\boldsymbol{P}^{\mathrm{out}}$ is i.i.d. and sampled from Gaussians $\mathcal{N}(0, 1/d)$ and $\mathcal{N}(0, 1/N)$, respectively.

An $L$-layer neural network $\boldsymbol{g}$ comprises weights $\boldsymbol{W}^{(l)} = (W_{ij}^{(l)}) \in \mathbb{R}^{N \times N}$ and biases $\boldsymbol{b}^{(l)} = (b_1^{(l)}, \ldots, b_N^{(l)})^\top \in \mathbb{R}^N$, where $l \in [L]$ denotes the layer index. The network is assumed to possess a sufficiently large width (i.e., $N$ is sufficiently large). The $l$-th pre- and post-activation are defined as $\boldsymbol{h}^{(l)} := \boldsymbol{W}^{(l)} \boldsymbol{x}^{(l-1)} + \boldsymbol{b}^{(l)}$ and $\boldsymbol{x}^{(l)} := \phi(\boldsymbol{h}^{(l)})$, respectively, where ReLU-like activation $\phi$ operates entry-wise. The weight $W_{ij}^{(l)}$ and bias $b_i^{(l)}$ are i.i.d. and sampled from $\mathcal{N}(0, \sigma_w^2/N)$ and $\mathcal{N}(0, \sigma_b^2)$, respectively. The network function is represented by Eq. (1). For a residual network setting, refer to Appx. C.

$$\boldsymbol{f}(\boldsymbol{x}^{\mathrm{in}}) := \boldsymbol{P}^{\mathrm{out}} \phi(\boldsymbol{W}^{(L)} \phi(\cdots \phi(\boldsymbol{W}^{(1)} \boldsymbol{P}^{\mathrm{in}} \boldsymbol{x}^{\mathrm{in}} + \boldsymbol{b}^{(1)}) \cdots) + \boldsymbol{b}^{(L)}). \tag{1}$$

## 3.2 Background

**Mean field theory.** Mean field theory employs probabilistic methods to analyze the properties of random deep neural networks. It assumes that $\boldsymbol{h}^{(l)}$ follows a Gaussian, justified by the central limit theorem when width $N$ is sufficiently large [71]. Here, we review the mean field-based approach to analyze the forward and backward dynamics of a network. Let $\mathcal{L} : \mathbb{R}^d \to \mathbb{R}$ represent the loss function. The mean squared pre-activation $\mathbb{E}[(h_i^{(l)})^2]$ and gradient $\chi^{(l)} := \mathbb{E}[(\partial \mathcal{L}(\boldsymbol{x}^{\mathrm{in}})/\partial x_i^{(l)})^2]$, where $i$ denotes the neuron index, are calculated as follows [71, 81]:

$$\mathbb{E}[(h_i^{(l)})^2] = \sigma_w^2 \mathbb{E}[\phi(h_i^{(l-1)})^2] + \sigma_b^2, \qquad \chi^{(l)} = \sigma_w^2 \mathbb{E}[\phi'(h_i^{(l+1)})^2] \chi^{(l+1)}. \tag{2}$$

These equations represent the dynamics between adjacent layers. We can infer that gradients vanish when $\sigma_w^2 \mathbb{E}[\phi'(h_i^{(l+1)})^2] < 1$ and explode when $\sigma_w^2 \mathbb{E}[\phi'(h_i^{(l+1)})^2] > 1$, indicating that a network is trainable only if $\sigma_w^2 \mathbb{E}[\phi'(h_i^{(l+1)})^2] \approx 1$.

**Adversarial training.** We define adversarial loss as follows:

$$\mathcal{L}_{\text{adv}}(\boldsymbol{x}^{\text{in}}) := \max_{\|\boldsymbol{\eta}\|_p \leq \epsilon} \big\| \boldsymbol{f}(\boldsymbol{x}^{\text{in}} + \boldsymbol{\eta}) - \boldsymbol{f}(\boldsymbol{x}^{\text{in}}) \big\|_q \tag{3}$$

$$= \max_{\|\boldsymbol{\eta}\|_p \leq \epsilon} \left\| \begin{array}{l} \boldsymbol{P}^{\text{out}}\phi(\boldsymbol{W}^{(L)}\phi(\cdots\phi(\boldsymbol{W}^{(1)}\boldsymbol{P}^{\text{in}}(\boldsymbol{x}^{\text{in}} + \boldsymbol{\eta}) + \boldsymbol{b}^{(1)})\cdots) + \boldsymbol{b}^{(L)}) \\ -\boldsymbol{P}^{\text{out}}\phi(\boldsymbol{W}^{(L)}\phi(\cdots\phi(\boldsymbol{W}^{(1)}\boldsymbol{P}^{\text{in}}\boldsymbol{x}^{\text{in}} + \boldsymbol{b}^{(1)})\cdots) + \boldsymbol{b}^{(L)}) \end{array} \right\|_q, \tag{4}$$

where $\epsilon > 0$ and $p, q \in \{1, 2, \infty\}$. The adversarial loss aims to minimize the difference between the network outputs of natural and adversarial inputs. Networks are trained by minimizing the sum of the standard loss $\mathcal{L}_{\text{std}} : \mathbb{R}^d \times \mathcal{Y} \to \mathbb{R}$ (e.g., cross-entropy loss), where $\mathcal{Y}$ denotes a label set, and the adversarial loss $\mathcal{L}_{\text{adv}}$. The mean field analysis typically assumes gradient independence for loss functions (Appx. B) [81]. We use this assumption for $\mathcal{L}_{\text{std}}$, but not $\mathcal{L}_{\text{adv}}$. Although Eq. (3) differs from the standard adversarial loss based on cross-entropy [58], our definition is employed in more robust methods, e.g., TRADES [113] and is theoretically simpler to analyze. Therefore, herein, the aforementioned loss is analyzed. However, even this simplified definition (Eq. (3)) poses a challenge for theoretical analysis due to the complex nested structure of a deep neural network (cf. Eq. (4)).

## 4 Theoretical framework

### 4.1 Limitations of existing mean field-based approaches

We propose a new theoretical framework based on mean field theory to analyze adversarial training. Here, we describe two limitations of existing mean field-based approaches, e.g., Eq. (2).

**Layer-wise approach.** Existing approaches focus on the dynamics between adjacent layers (cf. Eq. (2)). However, analyzing the adversarial loss (Eq. (3)) requires a framework that handles the probabilistic properties of an entire network instead of adjacent layers. This analysis becomes difficult due to the complex nested structure of networks (cf. Eqs. (1) and (4)). For example, there is no clarity on the the probabilistic behavior of $\boldsymbol{f}(\boldsymbol{x}^{\text{in}})$, distribution of $\boldsymbol{f}(\boldsymbol{x}^{\text{in}} + \boldsymbol{\eta}) - \boldsymbol{f}(\boldsymbol{x}^{\text{in}})$, and dependence between $\boldsymbol{f}(\boldsymbol{x}^{\text{in}})$ and inputs. We need a framework that disentangles the nested structure of networks and manages the probabilistic properties of an entire network. Recent studies on the mean field theory studies [15, 36, 69, 70, 100] have concepts related to ours; the comparative analysis is given in Sec. 5.1.

**Difficulty in analyzing input–parameter dependence.** The analysis of adversarial training requires consideration of input-parameter dependence since adversarial perturbations are designed based on network parameters (cf. Eq. (3)). However, this cannot be readily addressed using existing approaches because their frameworks (e.g., Eq. (2)) do not offer a clear view of the dependence between perturbation $\boldsymbol{\eta}$ and network parameters $W_{1,1}^{(1)}, W_{1,2}^{(1)}, \ldots,$ and $W_{N,N}^{(L)}$.

The proposed framework resolves these limitations and provides a simple network representation that allows us to capture the entire network property with clear input–parameter dependence.

### 4.2 Proposed framework

The current mean field-based approaches cannot capture the probabilistic properties of an entire network due to the complex nested structure of deep neural networks. Moreover, it is difficult to consider the dependence between inputs and numerous number of parameters. To address these limitations, we employ a linear-like representation of a ReLU-like network and propose its probabilistic properties. As ReLU-like networks are piecewise linear, a vanilla ReLU-like network can be represented as:

$$\boldsymbol{f}(\boldsymbol{x}^{\text{in}}) = \boldsymbol{J}(\boldsymbol{x}^{\text{in}})\boldsymbol{x}^{\text{in}} + \boldsymbol{a}(\boldsymbol{x}^{\text{in}}), \tag{5}$$

$$\boldsymbol{J}(\boldsymbol{x}^{\text{in}}) := \boldsymbol{P}^{\text{out}}\boldsymbol{D}(\phi'(\boldsymbol{h}^{(L)}(\boldsymbol{x}^{\text{in}})))\boldsymbol{W}^{(L)}\boldsymbol{D}(\phi'(\boldsymbol{h}^{(L-1)}(\boldsymbol{x}^{\text{in}})))\boldsymbol{W}^{(L-1)}\cdots\boldsymbol{W}^{(1)}\boldsymbol{P}^{\text{in}}, \tag{6}$$

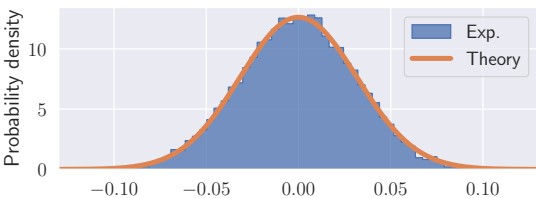

Figure 1: Distribution of $J(\boldsymbol{x}^{\text{in}})_{1,1}$ in the vanilla ReLU network with $d = 1,000$, $K = 1$, $N = 5,000$, $L = 10$, $\sigma_w^2 = 2$, and $\sigma_b^2 = 0.01$. The blue histogram represents the experimental results (10,000-time samplings), and the orange curve is predicted by Thm 4.1.

Table 1: Values of $\beta_{p,q}$. Under further assumptions, we can obtain equality of the adversarial loss rather than inequality (upper bound). Values marked with † represent the equality when $\epsilon$ is sufficiently small. Values marked with ◇ are applicable if $\epsilon$ is sufficiently small and $K = 1$.

|  | $q = 1$ | $q = 2$ | $q = \infty$ |
|---|---|---|---|
| $p = 1$ | $\sqrt{\frac{2K^2}{\pi d}}^{\dagger}$ | $\sqrt{\frac{K}{d}}^{\dagger}$ | $\sqrt{\frac{2\ln K}{d}}$ |
| $p = 2$ |  | $1^{\diamond} + \sqrt{\frac{K}{d}}$ | $1^{\dagger\diamond}$ |
| $p = \infty$ |  |  | $\sqrt{\frac{2d}{\pi}}^{\dagger\diamond}$ |

where $\boldsymbol{D}(\,\cdot\,)$ denotes a diagonal matrix and $\boldsymbol{a}(\boldsymbol{x}^{\text{in}})$ is defined similar to $\boldsymbol{J}(\boldsymbol{x}^{\text{in}})$ (cf. Eq. (A48)). For a residual network, Eq. (A73) can be referred. Importantly, this representation does not rely on approximations, e.g., Taylor expansions. As $\boldsymbol{W}^{(l)}$ and $\boldsymbol{b}^{(l)}$ are randomly sampled, $\boldsymbol{J}^{(l)}(\boldsymbol{x}^{\text{in}})$ and $\boldsymbol{a}^{(l)}(\boldsymbol{x}^{\text{in}})$ denote a random matrix and vector, respectively. Unlike the original network definition (Eq. (1)), this representation (Eq. (5)) is non-nested and focuses only two parameters, thereby simplifying network analysis. Remarkably, we show the following properties of $\boldsymbol{J}(\boldsymbol{x}^{\text{in}})$ and $\boldsymbol{a}(\boldsymbol{x}^{\text{in}})$:

**Theorem 4.1** (Properties and distributions of $\boldsymbol{J}(\boldsymbol{x}^{\text{in}})$ and $\boldsymbol{a}(\boldsymbol{x}^{\text{in}})$). *Suppose that the width $N$ is sufficiently large. Then, for any $\boldsymbol{x}^{\text{in}} \in \mathbb{R}^d$, (I) $\boldsymbol{J}(\boldsymbol{x}^{\text{in}})$ and $\boldsymbol{a}(\boldsymbol{x}^{\text{in}})$ are independent. (II) each entry of $\boldsymbol{J}(\boldsymbol{x}^{\text{in}})$ and $\boldsymbol{a}(\boldsymbol{x}^{\text{in}})$ is i.i.d. and follows the Gaussian below:*

$$J(\boldsymbol{x}^{\text{in}})_{ij} \sim \mathcal{N}\left(0, \frac{\omega^L}{d}\right), \qquad a(\boldsymbol{x}^{\text{in}})_i \sim \mathcal{N}\left(0, \alpha\sigma_b^2 \sum_{k=1}^{L} \omega^{k-1}\right), \qquad (7)$$

*where $\alpha := (u^2 + v^2)/2$ (cf. Defn 3.1) and $\omega$ is $\omega_{\text{v}} := \alpha\sigma_w^2$ for vanilla networks and $\omega_{\text{r}} := 1 + \alpha\sigma_w^2$ for residual networks.*

A significance of this theorem lies in that **despite being functions of $\boldsymbol{x}^{\text{in}}$, the distributions of $\boldsymbol{J}(\boldsymbol{x}^{\text{in}})$ and $\boldsymbol{a}(\boldsymbol{x}^{\text{in}})$ do not depend on $\boldsymbol{x}^{\text{in}}$.**[1] In other words, although $\boldsymbol{J}(\boldsymbol{x}^{\text{in}})$ and $\boldsymbol{a}(\boldsymbol{x}^{\text{in}})$ are determined by (a fixed) $\boldsymbol{x}^{\text{in}}$ for an initialized network, they become different for each sampling of weights and biases, and the selection of these values is independent of $\boldsymbol{x}^{\text{in}}$. Besides, $\boldsymbol{J}(\boldsymbol{x}^{\text{in}})$ and $\boldsymbol{a}(\boldsymbol{x}^{\text{in}})$ are independent and their distributions are Gaussian, which exhibits convenient properties. A sketch of proof is given in Appx. D and formal one is in Appxs. E and F.

To validate Thm 4.1, we conducted a numerical experiment and present the results in Fig. 1. We randomly sampled 10,000 vanilla ReLU networks and computed $J(\boldsymbol{x}^{\text{in}})_{1,1}$ for each network using the identical input $\boldsymbol{x}^{\text{in}}$. Additional experimental results can be found in Appx. L.

**Broader applicability.** The proposed framework (Thm 4.1) manages an entire network using only two Gaussians. While we focus on adversarial training, Thm 4.1 can be valuable for other deep neural network analyses based on the mean field theory. For example, contrastive learning [16] can be investigated, as it aims to minimize the distance between original and positive samples while maximizing it for negative samples. In this context, instead of considering the adversarial loss, $\|\boldsymbol{f}(\boldsymbol{x}^{\text{in}} + \boldsymbol{\eta}) - \boldsymbol{f}(\boldsymbol{x}^{\text{in}})\|$, we can analyze loss functions, e.g., $\|\boldsymbol{f}(\boldsymbol{x}_{\text{pos}}^{\text{in}}) - \boldsymbol{f}(\boldsymbol{x}_{\text{ori}}^{\text{in}})\|$ and $\|\boldsymbol{f}(\boldsymbol{x}_{\text{neg}}^{\text{in}}) - \boldsymbol{f}(\boldsymbol{x}_{\text{ori}}^{\text{in}})\|$, where $\boldsymbol{x}_{\text{ori}}^{\text{in}}$, $\boldsymbol{x}_{\text{pos}}^{\text{in}}$, and $\boldsymbol{x}_{\text{neg}}^{\text{in}}$ represent the original, positive, and negative samples, respectively. The complex nested structure of a network makes it challenging to consider the difference between two network outputs; however, Thm 4.1 helps theoretically manageable analyses.

## 5 Analysis of adversarial training

The proof of each theorem is described in Appx. G.

---

[1]This does not imply that random variables, $\boldsymbol{J}(\boldsymbol{x})$ and $\boldsymbol{J}(\boldsymbol{y})$, are identical for $\boldsymbol{x} \neq \boldsymbol{y}$. In addition, $\boldsymbol{J}(\boldsymbol{x})$ and $\boldsymbol{J}(\boldsymbol{y})$ are not always independent. Please also refer to the empirical results in Appx. L.

## 5.1 Upper bounds of adversarial loss

As the ReLU-like network $\boldsymbol{f}$ is locally linear and its input Jacobian at $\boldsymbol{x}^{\mathrm{in}}$ is $\boldsymbol{J}(\boldsymbol{x}^{\mathrm{in}})$ (cf. Eq. (5)), we can consider a more tractable form of the adversarial loss instead of Eq. (4) as follows:

$$\mathcal{L}_{\mathrm{adv}}(\boldsymbol{x}^{\mathrm{in}}) \leq \max_{\boldsymbol{x} \in \mathbb{R}^d, \|\boldsymbol{\eta}\|_p \leq \epsilon} \|\boldsymbol{J}(\boldsymbol{x})\boldsymbol{\eta}\|_q = \epsilon \max_{\boldsymbol{x} \in \mathbb{R}^d} \|\boldsymbol{J}(\boldsymbol{x})\|_{p,q}, \tag{8}$$

where $\|\boldsymbol{J}(\boldsymbol{x})\|_{p,q} := \max_{\|\boldsymbol{\eta}\|_p=1} \|\boldsymbol{J}(\boldsymbol{x})\boldsymbol{\eta}\|_q$ denotes the $(p,q)$-operator norm of $\boldsymbol{J}(\boldsymbol{x})$. Using Thm 4.1, which describes the property of $\boldsymbol{J}(\boldsymbol{x})$, we transform Ineq. (8) and obtain the following:

**Theorem 5.1** (Upper bounds of adversarial loss)**.** *Suppose that the input dimension $d$, output dimension $K$, and width $N$ are sufficiently large. Then, for any $\boldsymbol{x}^{\mathrm{in}} \in \mathbb{R}^d$, the following inequality holds:*

$$\mathcal{L}_{\mathrm{adv}}(\boldsymbol{x}^{\mathrm{in}}) \leq \epsilon\beta_{p,q}\omega^{L/2} = \begin{cases} \epsilon\beta_{p,q}(\frac{\alpha}{LN}\sum_{W\in\mathcal{W}} W^2)^{L/2} & \text{(vanilla)} \\ \epsilon\beta_{p,q}(1 + \frac{\alpha}{LN}\sum_{W\in\mathcal{W}} W^2)^{L/2} & \text{(residual)} \end{cases}, \tag{9}$$

*where $\mathcal{W} := \{W_{1,1}^{(1)}, W_{1,2}^{(1)}, \ldots, W_{N,N}^{(L)}\}$ denotes the set of all network weights. The constant $\beta_{p,q}$ for each norm pair $(p,q)$ is described in Tab. 1.*

For some choices of $(p,q)$, we cannot derive upper bounds, and thus, Tab. 1 contains blank. Numerical experiments show the tightness of the bounds (cf. Fig. 2). The theorem indicates that (i) the bounds increase linearly with the perturbation budget $\epsilon$, (ii) the effects of the input and output dimensions depend on the norms $(p,q)$ (cf. Tab. 1), and (iii) network depth $L$ exponentially impacts the bounds, with $\omega = 1$ as a threshold between order and chaos. Further, the square sum of the weights in Ineq. (9) suggests that adversarial training exhibits a weight regularization effect, which is compared to $\ell_2$ weight regularization in Appx. I. Besides, we derive equality rather than inequality (upper bound) under specific assumptions (e.g., small $\epsilon$) for some $(p,q)$ in Appx. K, indicated by † and ◇ in Tab. 1.

The input and output dimensions, $d$ and $K$, influence the bounds through the $(p,q)$-dependent parameter $\beta_{p,q}$. In Tab. 1, $\beta_{p,q}$ displays a wide range of dependencies on $d$ and $K$. When $d \to \infty$ and $q = \infty$, $d$ significantly affects the two phases of the adversarial loss, where $p = 2$ **marks the transition point from order ($p = 1$) to chaos ($p = \infty$)**. In contrast, under realistic assumptions with $d \gg K$, $K$ affects negligibly. Interestingly, **the dimensions do not impact the upper bounds when** $(p,q) = (2,\infty)$. In practice, we scale the perturbation budget and adversarial loss according to the choice of $(p,q)$, respectively, and discuss the scaling effects in Appx. K.

**Comparison with other studies.** We can consider studies on global Lipschitz of networks in certified adversarial defenses [3, 18, 93] and spectral regularization [60, 108] to analyze Ineq. (8). A key difference is that the proposed probabilistic approach contradicts their deterministic approach. By imposing probabilistic constraints on network parameters, we can obtain exponentially tighter, interpretable, and more theoretically manageable bounds, which facilitates the subsequent section's discussion. The mathematical comparison is described in Appx. H.

Results obtained from [15, 36, 69, 70, 100] can be used to analyze the Jacobian's singular value distribution. Compared to their approaches, which are limited to $(p,q) = (2,2)$, the proposed method offers greater generality and flexibility, providing upper bounds for various $(p,q)$. Moreover, Thm 4.1 enables the derivation of equality instead of inequality (upper bound), Props G.2 and G.3, which is not achievable using the approaches mentioned in the aforementioned studies because it cannot incorporate perturbation-Jacobian dependence. Moreover, we do not consider their assumption that the variance $\mathbb{V}[h_i^{(l)}]$ is constant for all $l \in [L]$, which is often difficult to satisfy.

Further, we refer to [78], which established a theoretical link between adversarial training and the $(p,q)$-operator norm of a Jacobian. Their findings support Ineq. (8) in training scenarios using heuristic attacks, e.g., projected gradient descent [58]. In this study, we derive concrete upper bounds beyond their theoretical link, enabling further investigation of adversarial training properties.

## 5.2 Time evolution of weight variance

Weight variance plays a critical role in determining deep neural network properties [49, 81]. We substitute the adversarial loss definition (Eq. (3)) with $\mathcal{L}_{\mathrm{adv}} := \epsilon\beta_{p,q}\omega(t)^{L/2}$, where $t \geq 0$ denotes the

continuous training step. Considering gradient descent with an infinitely small learning rate (gradient flow), the model parameter $\theta(t)$ at step $t$ is updated as:

$$\frac{\mathrm{d}\theta(t)}{\mathrm{d}t} := -\frac{\partial\mathcal{L}_{\text{std}}}{\partial\theta(t)} - \frac{\partial\mathcal{L}_{\text{adv}}}{\partial\theta(t)}. \tag{10}$$

We make the following assumption.

**Assumption 5.2.** For $0 \leq t \leq T \ll N$, model parameters are independent, and weight and bias follow Gaussian $\mathcal{N}(0, \sigma_w^2(t)/N)$ and $\mathcal{N}(0, \sigma_b^2(t))$, respectively.

This assumption ensures that the properties of the model parameters remain close to their initial values during the early stages of training ($t \leq T$). Under Asm 5.2, the original and proposed mean field theories (Thm 4.1) remain valid during training. For a moderately small value of $T$, Asm 5.2 is not strong because the model parameters change minimally and retain their initialized states with sufficiently small learning rates. Recent neural tangent kernel studies partially supported this assumption [4, 46, 55], and it is known that random network theories align well with fully trained networks [25, 84]. For example, $T = 160$ is reasonable in a specific training setting (cf. Fig. 4).

Now, we summarize other assumptions for reference as follows:

**Assumption 5.3.** The input dimension $d$, output dimension $K$, and width $N$ are sufficiently large. We apply Asm B.1 to the standard loss function $\mathcal{L}_{\text{std}}$. The adversarial loss is defined as $\mathcal{L}_{\text{adv}} := \epsilon\beta_{p,q}\omega(t)^{L/2}$ (cf. Thm 5.1).

Based on the aforementioned settings, we obtain the time evolution of the weight variance.

**Theorem 5.4** (Weight time evolution of vanilla network in adversarial training). *Suppose that Asms 5.2 and 5.3 hold. Then, the time evolution of $\sigma_w^2$ of a vanilla network in adversarial training is given by:*

$$\sigma_w^2(t) = \left(1 - \frac{\epsilon\alpha\beta_{p,q}\omega_{\text{v}}(0)^{L/2-1}}{N}t\right)\sigma_w^2(0). \tag{11}$$

A similar result is obtained for residual networks (Thm G.9). The theorem reveals that the weight variance linearly decreases with $t$, which can be attributed to the weight regularization effect of adversarial training (cf. Sec. 5.1). In addition, the norm pair $(p, q)$ affect only time-invariant constant $\beta_{p,q}$, and the dynamics of the weight variance can be represented as a consistent function of $t$ regardless of $(p, q)$. In other words, **adversarial training exhibits consistent weight dynamics irrespective of norm selection**, with a scale factor varying.

## 5.3 Vanilla networks are not adversarially trainable under mild conditions

We show that in adversarial training, vanilla networks can fit a training dataset in limited cases (small depth and large width), but residual networks can in most cases. This result suggests that residual networks are better suited for adversarial training. First, we present the trainability condition based on the concept in [81, 105].

**Definition 5.5** ($(M, m)$-trainability condition). A network is said to be $(M, m)$-trainable if a network satisfies $m \leq \chi^{(0)}/\chi^{(L)} \leq M$, where $0 \leq m \leq 1$ and $M \geq 1$.[2]

The value $\chi^{(l)}$ denotes the squared length of the gradient in the $l$-th layer. A near-zero $\chi^{(0)}/\chi^{(L)}$ suggests gradient vanishing, while a large value implies gradient explosion. Hence, Defn 5.5 is directly linked to successful training. In contrast to the existing definition, $\chi^{(l-1)}/\chi^{(l)} \approx 1$ [81, 105], Defn 5.5 incorporates $M$ and $m$ for the subsequent discussion. Then, we establish specific $(M, m)$-trainability conditions for ReLU-like networks.

**Lemma 5.6** (Vanilla and residual $(M, m)$-trainability condition). *Suppose that the width $N$ is sufficiently large. Then, the $(M, m)$-trainability conditions for vanilla and residual networks are respectively given by:*

$$m^{1/L} \leq \alpha\sigma_w^2 \leq M^{1/L} \text{ (vanilla)}, \qquad \alpha\sigma_w^2 \leq M^{1/L} - 1 \text{ (residual)}. \tag{12}$$

---

[2]Trainability depends on other factors such as the number of parameters; however, these are not considered herein to maintain simplicity. It is empirically known that this condition represents trainability well [81].

Using the weight time evolution (Thm 5.4) and $(M, m)$-trainability condition (Lemma 5.6), the following theorem can be readily derived.

**Theorem 5.7** (Vanilla networks are not adversarially trainable). *Consider a vanilla network. Suppose that Asms 5.2 and 5.3 hold, and the $(M, m)$-trainability condition holds at $t = 0$ and $\alpha\sigma_w^2(0) = 1$. If*

$$T \geq \frac{(1 - m^{1/L})N}{\epsilon\alpha\beta_{p,q}}, \tag{13}$$

*then there exists $0 < \tau \leq T$ such that the $(M, m)$-trainability condition does not hold for $\tau \leq t \leq T$.*

This indicates **the potential untrainability of vanilla networks in adversarial training, even when satisfying the $(M, m)$-trainability condition at initialization, contradicting with standard training where extremely deep networks can be trained if initialized properly [81]**. This issue arises from the inconsistency between the trainability condition of a vanilla network, i.e., $m^{1/L} \leq \alpha\sigma_w^2$ (cf. Lemma 5.6) and monotonically decreasing nature of $\sigma_w^2(t)$ in adversarial training (cf. Thm 5.4). As stated in Asm 5.2, we assume that $T$ is small. Therefore, if the right-hand term of Eq. (13) becomes large, the assumption and Thm 5.7 are violated. In summary, **for large $L$ (many layers) and $\epsilon$ (large perturbation constraint), vanilla networks are not adversarially trainable**. Moreover, **a large $N$ (a wide network) can mitigate this issue in vanilla networks**. For example, the vanilla network with $L = 20$, $N = 256$, and $\epsilon = 0.3$ is not adversarially trainable; however, when the width is increased to 512, it becomes trainable (cf. Fig. 4). In contrast, we can claim the following for residual networks:

**Theorem 5.8** (Residual networks are adversarially trainable). *Consider a residual network. Suppose that Asms 5.2 and 5.3 hold, and the $(M, m)$-trainability condition holds at $t = 0$ and $\alpha\sigma_w^2(0) \ll 1$. Then, $(M, m)$-trainability condition always holds for $0 \leq t \leq T$.*

This occurs due to the $(M, m)$-trainability condition for residual networks, which does not have any lower bound (cf. Lemma 5.6), and the monotonically decreasing nature of $\sigma_w^2$ (cf. Thm G.9). Besides, residual networks are adversarially trainable even without careful weight initialization (Prop G.10). These theorems highlight the robust stability of adversarial training in residual networks.

## 5.4 Degradation of network capacity

We demonstrate that adversarial training degrades network capacity. We use the Fisher–Rao norm as a metric [56], with alternative metrics discussed in Appx. K. The Fisher–Rao norm is defined as:

$$\|\boldsymbol{w}\|_{\mathrm{FR}} := \boldsymbol{w}^\top \boldsymbol{F}(\boldsymbol{x}^{\mathrm{in}})\boldsymbol{w}, \qquad \boldsymbol{F}(\boldsymbol{x}^{\mathrm{in}}) := \sum_{i=1}^{K} \left(\frac{\partial f_i(\boldsymbol{x}^{\mathrm{in}})}{\partial \boldsymbol{w}}\right)^\top \frac{\partial f_i(\boldsymbol{x}^{\mathrm{in}})}{\partial \boldsymbol{w}}, \tag{14}$$

where $\boldsymbol{w} := (W_{11}^{(1)}, W_{12}^{(1)}, \ldots, W_{NN}^{(L)})^\top$ represents the vector of all the network weights and $\boldsymbol{F}$ denotes the empirical Fisher information matrix when only one training data point is considered (for simplicity). The Fisher–Rao norm is preferred over other norm-based capacity metrics [8, 65–67] owing to its invariance to node-wise rescaling [56]. We present the following theorem based on [49]:

**Theorem 5.9** (Adversarial training degrades network capacity). *Consider a vanilla network. Suppose that Asms 5.2 and 5.3 hold, Asm B.1 is applied to the network output, and the $(M, m)$-trainability condition holds at $t = 0$ and $\alpha\sigma_w^2(0) = 1$. Assume $\|\boldsymbol{x}^{\mathrm{in}}\|_2 = \sqrt{d}$ and $\sigma_b^2(t) = 0$. Then, the expectation of the Fisher–Rao norm is given by:*

$$\mathbb{E}[\|\boldsymbol{w}(t)\|_{\mathrm{FR}}] = LK\left(1 - \frac{\epsilon\alpha\beta_{p,q}L}{N}t\right). \tag{15}$$

A related result for residual networks is presented in Thm G.14. As described in [63], adversarial robustness necessitates high capacity. However, Thm 5.9 indicated that capacity decreases linearly with step $t$ in adversarial training. To address this conflict, large depth (large $L$), which increase the initial capacity with $\Theta(L)$, can be considered, but this scenario accelerates the degradation speed with $\Theta(L^2)$. To preserve capacity, we must increase the width $N$ accordingly. Consequently, **to achieve high capacity in adversarial training, it is necessary to increase not only the number of layers $L$ but also the width $N$ to keep $L^2/N$ constant**. Although Thms 5.9 and G.14 have been established in the early stages of training, numerical experiments proved that the adversarial robustness following full training is significantly influenced by network width (cf. Tabs. A5 and A6).

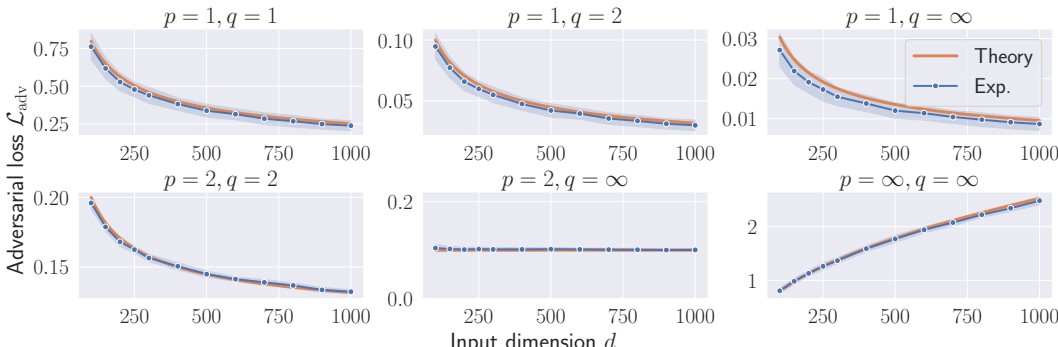

Figure 2: Adversarial loss (Eq. (3)) in vanilla networks with $N = 40,000$, $K = 100$, $L = 3$, and $\epsilon = 0.1$. We generated 100 adversarial examples for each input dimension. The blue curves and bands represent the mean and standard deviation of the adversarial loss, respectively, whereas the orange curves (upper bounds) are predicted based on Thm 5.1. Some samples slightly exceed the upper bounds because we used the finite network width (cf. Appx. L).

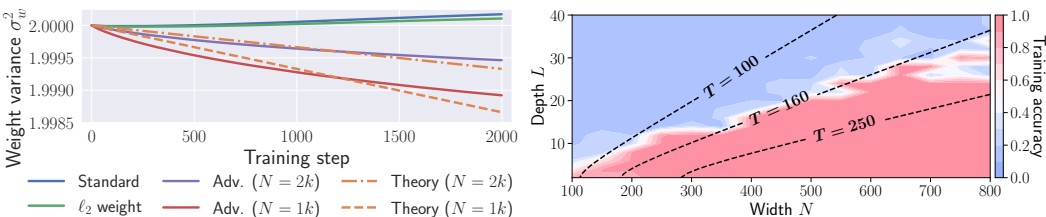

Figure 3: Time evolution of the weight variance in the vanilla network with $L = 10$, $p = \infty$, $q = \infty$, and $\epsilon = 0.3$. We used $N = 1,000$ for standard and $\ell_2$ regularized training. The solid lines represent experimental results. The dashed lines are predicted by Thm 5.4.

Figure 4: Heat map of the training accuracy of vanilla networks with $p = \infty$, $q = \infty$, and $\epsilon = 0.3$. The dashed lines represent the condition of $T$ in Thm 5.7 with $m = 0.0001$. In standard training, high accuracy is obtained across all the depths and widths (cf. Fig. A19).

## 6 Experimental results

We validate Thms 5.1, 5.4 and 5.7 via numerical experiments. The vanilla ReLU networks were initialized with $\sigma_w^2 = 2$ and $\sigma_b^2 = 0.01$ to meet the $(M, m)$-trainability conditions (Lemma 5.6). To verify Thms 5.4 and 5.7, we used MNIST [26]. Setup details and more results are given in Appx. L.

**Verification of Thm 5.1.** We created adversarial examples for initialized networks and computed the adversarial loss (Eq. (3)). As shown in Fig. 2, the upper bounds in Thm 5.1 were considerably tight. Some samples slightly exceeded the upper bounds because we used the finite network width while the infinite width is assumed (cf. Appx. L).

**Verification of Thm 5.4.** We trained vanilla networks normally (with and without $\ell_2$ regularization) and adversarially. The time evolution of weight variance is shown in Fig. 3. Adversarial training significantly reduces weight variance, whereas wide width (i.e., large $N$) suppresses it. The validity of Thm 5.4, which forms the basis of our theorems such as Thms 5.7 and 5.9, supports our theorems.

**Verification of Thm 5.7.** We trained vanilla networks considering various depth and width settings and monitored the training accuracy. As shown in Fig. 4, it was difficult for vanilla networks to fit the training dataset when the depth was large and the width was small; increased width helps in fitting. Although we currently lack a theoretical prediction of the boundary between trainable and untrainable areas determined based on Eq. (13), empirical evidence suggests that $T = 160$ is relevant.

# 7 Limitations

The mean field theory offers valuable insight into network training. However, its applicability is restricted to the initial stages of training. Although recent studies suggested empirically [25, 84] and theoretically [46] that the analysis of early-stage training extends well to full training, the strict relationship is yet to be explored. Our results have the same limitations. Although some theorems accurately capture the behavior during the initial stages of training and even after full training (cf. Tabs. A5 and A6), as training progresses, some theorems begin to diverge from the actual behavior (cf. Fig. A17). Another caveat is that the mean field theory assumes infinite network width, which is practically infeasible. Empirically, our theorems hold well when the width approximately exceeds 1,000 (cf. Fig. A7), while Thm 5.1 requires larger width approximately exceeds 10,000 for $(p, q) = (2, 2)$ (cf. Fig. A14). These limitations also derive from the mean field theory and are not unique to our study. Despite these limitations, we consider that this study provides a powerful theoretical framework that extends the applicability of the mean field theory to various training methods, and the results obtained for adversarial training are insightful.

# 8 Conclusions

We proposed a framework based on the mean field theory and conducted a theoretical analysis of adversarial training. The proposed framework addressed the limitations of existing mean field-based approaches, which could not handle the probabilistic properties of an entire network and dependence between network inputs and parameters [71, 81]. Based on this framework, we examined adversarial training from various perspectives, unveiling upper bounds of adversarial loss, relationships between adversarial loss and network input/output dimensions, the time evolution of weight variance, trainability conditions, and the degradation of network capacity. The theorems of this study were validated via numerical experiments. The proposed theoretical framework is highly versatile and can help analyze various training methods, e.g., contrastive learning.

## Acknowledgments and Disclosure of Funding

We would like to thank Huishuai Zhang for useful discussions. This work was supported by JSPS KAKENHI Grant Number JP23KJ0789 and JP22K17962, by JST, ACT-X Grant Number JPM-JAX23C7, JAPAN, and by Microsoft Research Asia.

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

Table A2: Notation. While $\boldsymbol{h}^{(l)}$ is a function that takes $\boldsymbol{x}^{\mathrm{in}}$ as input, we omit the argument as $\boldsymbol{h}^{(l)} := \boldsymbol{h}^{(l)}(\boldsymbol{x}^{\mathrm{in}})$ for notational simplicity. Other symbols sometimes follow this.

| Notation | Description | Cf. |
|---|---|---|
| $d \in \mathbb{N}$ | Input dimension of the network | Sec. 3.1 |
| $K \in \mathbb{N}$ | Output dimension of the network | Sec. 3.1 |
| $L \in \mathbb{N}$ | Number of layers in the network, i.e., network depth | Sec. 3.1 |
| $N \in \mathbb{N}$ | Number of neurons in a layer, i.e., network width | Sec. 3.1 |
| $u, v \in \mathbb{R}$ | Slope of a ReLU-like function | Defn 3.1 |
| $p \in \{1, 2, \infty\}$ | Perturbation constraint norm, $\|\boldsymbol{\eta}\|_p \leq \epsilon$ | Eq. (3) |
| $q \in \{1, 2, \infty\}$ | Output difference norm, $\left\|\boldsymbol{f}(\boldsymbol{x}^{\mathrm{in}} + \boldsymbol{\eta}) - \boldsymbol{f}(\boldsymbol{x}^{\mathrm{in}})\right\|_q$ | Eq. (3) |
| $m \in [0, 1]$ | $(M, m)$-trainability condition | Defn 5.5 |
| $\epsilon \geq 0$ | Perturbation constraint, $\|\boldsymbol{\eta}\|_p \leq \epsilon$ | Eq. (3) |
| $\alpha \geq 0$ | $\alpha := (u^2 + v^2)/2$ | Thm 4.1 |
| $\beta_{p,q} \geq 0$ | Time-invariant constant determined by $(p, q)$ | Thm 5.1 |
| $t \geq 0$ | Continuous training step | Eq. (10) |
| $T \geq 0$ | Upper limit of training steps | Asm 5.2 |
| $\sigma_w^2 \geq 0$ | Weight variance, $W_{ij} \sim \mathcal{N}(0, \sigma_w^2/N)$ | Sec. 3.1 |
| $\sigma_b^2 \geq 0$ | Bias variance, $b_i \sim \mathcal{N}(0, \sigma_b^2)$ | Sec. 3.1 |
| $\omega \geq 0$ | $\omega_{\mathrm{v}} := \alpha\sigma_w^2$ (vanilla) and $\omega_{\mathrm{r}} := 1 + \alpha\sigma_w^2$ (residual) | Thm 4.1 |
| $M \geq 1$ | $(M, m)$-trainability condition | Defn 5.5 |
| $\boldsymbol{x}^{\mathrm{in}} \in \mathbb{R}^d$ | Input vector | Sec. 3.1 |
| $\boldsymbol{\eta} \in \mathbb{R}^d$ | Perturbation, $\|\boldsymbol{\eta}\|_p \leq \epsilon$ | Eq. (3) |
| $\boldsymbol{W}^{(l)} \in \mathbb{R}^{N \times N}$ | $l$-th weight, $W_{ij}^{(l)} \sim \mathcal{N}(0, \sigma_w^2/N)$ | Sec. 3.1 |
| $\mathcal{W} \subset \mathbb{R}$ | Set of all network weights, $\{W_{11}^{(1)}, W_{12}^{(1)}, \ldots, W_{NN}^{(L)}\}$ | Thm 5.1 |
| $\boldsymbol{w} \in \mathbb{R}^{LN^2}$ | Vector of all the weights, $(W_{11}^{(1)}, \ldots, W_{11}^{(L)})^\top$ | Eq. (14) |
| $\boldsymbol{b}^{(l)} \in \mathbb{R}^N$ | $l$-th bias, $b_i^{(l)} \sim \mathcal{N}(0, \sigma_b^2)$ | Sec. 3.1 |
| $\boldsymbol{P}^{\mathrm{in}} \in \mathbb{R}^{N \times d}$ | Rand. mat. for input adjustment, $P_{ij}^{\mathrm{in}} \sim \mathcal{N}(0, \frac{1}{d})$ | Sec. 3.1 |
| $\boldsymbol{P}^{\mathrm{out}} \in \mathbb{R}^{K \times N}$ | Rand. mat. for output adjustment, $P_{ij}^{\mathrm{out}} \sim \mathcal{N}(0, \frac{1}{N})$ | Sec. 3.1 |
| $\boldsymbol{P}^{(l)} \in \mathbb{R}^{N \times N}$ | Rand. mat. in the $l$-th shortcuts, $P_{ij}^{(l)} \sim \mathcal{N}(0, \frac{1}{N})$ | Eq. (A16) |
| $\phi : \mathbb{R} \to \mathbb{R}$ | ReLU-like activation, $\phi := uz(z \geq 0); vz(z \leq 0)$ | Defn 3.1 |
| $\chi^{(l)} : \mathbb{R}^d \to \mathbb{R}$ | Mean squared $l$-th gradient, $\mathbb{E}[(\partial\mathcal{L}(\boldsymbol{x}^{\mathrm{in}})/\partial x_i^{(l)})^2]$ | Sec. 3.1 |
| $\boldsymbol{h}^{(l)} : \mathbb{R}^d \to \mathbb{R}^N$ | $l$-th pre-activation, $\boldsymbol{h}^{(l)} := \boldsymbol{W}^{(l)}\boldsymbol{x}^{(l-1)}(\boldsymbol{x}^{\mathrm{in}}) + \boldsymbol{b}^{(l)}$ | Sec. 3.1 |
| $\boldsymbol{x}^{(l)} : \mathbb{R}^d \to \mathbb{R}^N$ | $l$-th post-activation, $\boldsymbol{x}^{(l)} := \phi(\boldsymbol{h}^{(l)}(\boldsymbol{x}^{\mathrm{in}}))$ | Sec. 3.1 |
| $\boldsymbol{F} : \mathbb{R}^d \to \mathbb{R}^{LN^2 \times LN^2}$ | Empirical Fisher information matrix | Eq. (14) |
| $\boldsymbol{f} : \mathbb{R}^d \to \mathbb{R}^K$ | Overall network, $\boldsymbol{f}(\boldsymbol{x}^{\mathrm{in}}) := \boldsymbol{P}^{\mathrm{out}}\boldsymbol{g}(\boldsymbol{P}^{\mathrm{in}}\boldsymbol{x}^{\mathrm{in}})$ | Sec. 3.1 |
| $\boldsymbol{g} : \mathbb{R}^N \to \mathbb{R}^N$ | $L$-trainable layer ReLU-like network | Sec. 3.1 |
| $\boldsymbol{J} : \mathbb{R}^d \to \mathbb{R}^{K \times d}$ | Slope of a piecewise linear region at $\boldsymbol{x}^{\mathrm{in}}$ | Thm 4.1 |
| $\boldsymbol{a} : \mathbb{R}^d \to \mathbb{R}^K$ | Bias of a piecewise linear region at $\boldsymbol{x}^{\mathrm{in}}$ | Thm 4.1 |
| $\boldsymbol{D} : \mathbb{R}^N \to \mathbb{R}^{N \times N}$ | Diagonal matrix | Eq. (5) |
| $\mathcal{L}_{\mathrm{std}} : \mathbb{R}^d \times \mathcal{Y} \to \mathbb{R}$ | Standard loss function, where $\mathcal{Y}$ represents a label set | Sec. 3.2 |
| $\mathcal{L}_{\mathrm{adv}} : \mathbb{R}^d \to \mathbb{R}_{\geq 0}$ | Adversarial loss function | Eq. (3) |

# A  Additional related work

Please also refer to Secs. 2 and 5.1.

## A.1  Random networks and mean field theory

Understanding general deep neural networks is challenging due to the non-convexity of loss surfaces and stochastic nature of optimization. Researchers have explored random networks, which have random parameters instead of trained parameters. Random networks have been primarily studied in three areas: compositional kernels [17, 22, 23], neural network Gaussian processes [21, 44, 54, 59,

64, 68, 95, 103], and mean field theory. These fields share close relationships, particularly between neural network Gaussian processes and mean field theory. For example, the forwarding dynamics in mean field theory (Eq. (2)) is equivalent to the kernel representation of a Gaussian process [64, 71]. Furthermore, the accuracy of a Gaussian process is significantly influenced by the edge of chaos, which has been studied in mean field theory [68]. We provide a more detailed review of the mean field theory.

Mean field theory for neural networks was first introduced in [87] and later extended in [71, 81]. This theory examines the signal propagation, dynamics, and trainability of random networks in two phases: ordered and chaotic. The ordered phase is characterized by decreasing layer output variance and vanishing gradients during backpropagation, while the chaotic phase is characterized by expanding variance and exploding gradients. Effective training of deep neural networks occurs near the boundary between these two phases [71, 81]. Researchers have applied mean field theory to study various network architectures, including dropout [81], batch normalization [104], residual network [105, 106], recurrent network [15], quantized network [10], and Swish [41]. Recent research [15, 36, 69, 70, 100] has also utilized the mean field theory to analyze dynamical isometry, where all singular values of the Jacobian are one. Some of these findings can be applied to analyze Ineq. (8) in this study, with detailed comparisons provided in Sec. 5.1. Other studies have explored network representation power [49, 71, 101]. This work utilized the proof idea of [49] in the derivation of Thms 5.9 and G.14.

In this study, we employed mean field theory to analyze adversarial training behavior. However, the original theory has two limitations that make it unsuitable for this purpose (cf. Sec. 4.1). To address these limitations, we introduced a new mean field-based framework (cf. Sec. 4.2). Our theory is also applicable to mean field-based analyses of other deep learning methods beyond adversarial training.

## A.2 Adversarial examples

### A.2.1 Adversarial examples in random neural networks

We summarize the literature on adversarial examples in random neural networks [7, 11, 24, 25, 61, 84, 112]. It has been reported that adversarial perturbations can be found through gradient flow in most random ReLU networks with decreasing layer widths [24]. This result was extended to two-layer random networks with greater width than input dimension [11] and generalized to random ReLU networks with constant depth and wide width [7]. Recently, similar results were presented without width restrictions and for local Lipschitz continuous activation [61]. Additionally, it has been demonstrated that adversarial noises generated by a single-step attack are linearly separable in a two-layer random network and the neural tangent kernel regime [112]. More information on [25, 84] can be found in Appx. A.2.2.

Although the context differs somewhat, we mention [34], which provides empirical evidence that batch normalization leads to adversarial vulnerability. This observation aligns with previous results from mean field theory suggesting that batch normalization causes gradient explosion [104].

In this study, we focused on early stage adversarial training properties rather than adversarial examples. A key difference is that we primarily investigated the maximum difference between network outputs for standard and adversarial inputs rather than misclassification which was the main focus of previous studies [7, 11, 24, 61]. Nonetheless, some findings in Sec. 5.1 can be applied to the understanding of adversarial examples in random networks. In addition, in Appx. K, we proved the existence of adversarial examples in random networks to demonstrate the effectiveness of Thm 4.1.

### A.2.2 Adversarial examples and input dimension

We present research investigating the relationship between adversarial examples and input dimensions [2, 25, 30–32, 37, 38, 82, 84]. Apart from [38],[3] most studies have indicated that adversarial example threats increase with the square root of the input dimension [25, 30–32, 37, 82, 84]. Some studies have focused on specific data distributions or simple classifiers [30–32, 37, 82], while others targeted random networks [25, 84]. The study in [25] has examined the distance from the decision boundary in random networks, and concluded that adversarial examples deceive classifiers more easily with the square root of the input dimension. Another study has utilized the first-order Taylor

---

[3]Indeed, they omitted weight scaling for simplicity, and their results were essentially consistent with those of other studies when the scaling was considered.

expansion of a loss function to analyze the loss gradient with respect to an input [84]. While a direct comparison between our work and [84] is challenging due to differing loss functions, our theorems offer two advantages. First, we considered both the input and output dimensions, whereas they addressed only input dimensions. Second, our theorems are applicable to residual networks, whereas their assumptions are not.

In this study, we examined the relationship between an input dimension and adversarial risk in Sec. 5.1. Our analysis did not depend on specific data distributions or architectures, except for ReLU-like activations and random parameters, which is advantageous. In addition, we assessed it for a wide variety of norms, demonstrating that the impact of adversarial examples is not limited to the square root of the input dimension alone (cf. Tab. 1). Moreover, our bound considers not only the input dimension but also the number of classes.

## A.3  Adversarial training

Numerous empirical adversarial defenses have been proposed; however, most are ineffective against stronger attacks [5, 12, 13, 20, 90]. Some studies have focused on theoretically certified defenses [19, 39, 43, 74, 96], but these are often only applicable to specific or small networks, or are weaker than empirical methods. Adversarial training [38, 58] is considered the most effective empirical defense against various attacks [5, 20, 90]. This involves training a classifier using a dataset that contains natural images and adversarial examples [38] or solely adversarial examples [58]. Various forms of adversarial training exist, including more effective loss functions [28, 48, 58, 94, 113], time-efficient frameworks [83, 97, 111], and procedures that preserve the clean accuracy [72, 113, 114]. Recent studies have demonstrated that combining adversarial training and data augmentation with unlabeled or generated data results in high robustness [40, 77]. Despite significant progress in empirical methods, a theoretical understanding of adversarial training remains incomplete. We provide a summary of theoretical studies on adversarial training, including robust generalization research.

Several studies have investigated the trade-off between robust and clean accuracy [29, 47, 75, 76, 92, 113], initially observed empirically [58, 88]. The trade-off has been proven inevitable even in the infinite data limit, assuming data is constructed from a moderately correlated single feature and weakly correlated many features [92]. Similar claims were found in [113] for different data distributions. It has been reported that the trade-off in finite data settings for linear and slightly more complex predictors can be mitigated with additional unlabeled data [76]. While comparable results were reported in [75], a contrasting study also exists [47]. Moreover, the trade-off has been shown to depend on class imbalance in a dataset using Gaussian classification models [29].

Various studies have examined the generalization gap of adversarial robust models [6, 50, 102, 107]. For example, ResNet [42] trained adversarially on CIFAR-10 [52] achieved 96% robust training accuracy but only 47% robust test accuracy [107]. The lower bound of adversarial Rademacher complexity [9] has been shown to increase with the square root of the input dimension for linear classifiers trained with $\ell_\infty$ adversarial examples [107], which was later extended in [6]. Similar results using a tree transformation approach have been reported in [50]. However, the influence of network width and depth, or other training settings, such as training with $\ell_1$ adversarial examples, on the bound remains unclear. The generalization gap has been investigated for linear regression models and two-layer neural networks with lazy training in data interpolation contexts [102].

Numerous studies have explored the sample complexity of robust generalization [1, 14, 62, 80, 110]. Robust learning may require a larger sample size than standard learning [80]. Subsequent research has indicated that unlabeled data can be sufficient to achieve robust generalization [1, 14, 62, 110]. The majority of these studies have focused on data sampled from Gaussian mixture models [1, 14, 110]. Moreover, the sample complexity of distributionally robust learning with perturbations in the Wasserstein ball has been examined [62], and some studies have suggested that the trade-off between robustness and accuracy can be mitigated using additional unlabeled data [75, 76].

Recent findings have suggested that robust classification requires complex classifiers [63], which is supported by the results in the neural tangent kernel regime [35]. In the context of transfer learning, robust classifiers have been shown to perform better [27]. Furthermore, a certifiable adversarial training procedure has been established, constraining perturbation by the distributional Wasserstein distance [86].

The aforementioned outcomes rely on specific data distributions, such as Gaussian or heuristic-tuned distributions, or simplistic models, such as linear classifiers or two-layer neural networks. The lack of theoretical research on adversarial training in deep neural networks stems from the complexity of training these models, including non-convex loss surfaces and stochastic optimization. Recent research has employed the neural tangent kernel regime to address these challenges, demonstrating that adversarial training can yield a robust network with near-zero robust loss [35]. This result was later extended in two-layer neural networks [115], eliminating the assumption in [35] that requires exponentially large width and runtime.

In this study, we conducted a theoretical analysis of adversarial training, focusing on the time evolution of network parameters during training, the conditions promoting adversarial training, and the differences between adversarial and standard training. Our analysis targeted deep neural networks without relying on any assumptions about data distribution and employed $\ell_p$ norms practically used as perturbation constraints, rather than more impractical metrics such as the distributional Wasserstein distance. To address the training difficulty of deep neural networks, we utilized mean field theory.

Finally, we mention the report from a perturbation instability perspective that increasing network width does not necessarily improve robustness in adversarial training [98]. This may seem to contradict our results, which suggest that a wider network can help the model maintain capacity during adversarial training, implying greater robustness in wider networks. However, these two claims are compatible. Robustness is determined by both perturbation instability (negative effect) and network capacity (positive effect). While the negative effect of width appears dominant in [98]'s experiments on CIFAR-10 and WideResNet, the positive effect appeared more prevalent in our experiments on MNIST, Fashion-MNIST, and fully connected networks with or without shortcuts. The dominant factor may depend on the dataset and model architectures.

## B   Gradient independence assumption

The gradient independence assumption was first introduced in [81] for backward dynamics in mean field theory and later refined in [105]. We provide a definition based on [105] as follows:

**Assumption B.1** (Gradient independence assumption [105])**.** (a) We use a different set of weights for backpropagation than those used to compute the network outputs, but sampled i.i.d. from the same distributions. (b) For any loss $\mathcal{L}$, the gradient at layer $l$, $\partial\mathcal{L}/\partial\boldsymbol{x}^{(l)}$, is independent of $\boldsymbol{h}^{(l)}$ and $\boldsymbol{x}^{(l-1)}$.

Although not strictly accurate, this assumption has been empirically found to hold well [81, 105]. It has been rigorously justified for specific architectures, including vanilla, residual, and convolutional networks [103]. In this study, we applied the assumption to the standard loss function but not to the adversarial loss function. In addition, we regard a network output as a loss and apply Asm B.1 to the network output in Sec. 5.4.

## C   Setting of residual networks

The mean field theory for residual networks is studied in [105]. Although they employed trainable weights in shortcuts, we employ the untrainable matrix. Formally, the pre- and post-activations in the $l$-th layer of a residual network are defined as follows:

$$\boldsymbol{h}^{(l)} := \boldsymbol{W}^{(l)}\boldsymbol{x}^{(l-1)} + \boldsymbol{b}^{(l)}, \qquad\qquad \boldsymbol{x}^{(l)} := \boldsymbol{x}^{(l-1)} + \boldsymbol{P}^{(l)}\boldsymbol{\phi}(\boldsymbol{h}^{(l)}), \qquad \text{(A16)}$$

where $\boldsymbol{P}^{(l)} \in \mathbb{R}^{N\times N}$ is an untrained random matrix. Each entry of $\boldsymbol{P}^{(l)}$ is i.i.d. sampled from a Gaussian $\mathcal{N}(0, 1/N)$ at initialization. The random matrix $\boldsymbol{P}^{(l)}$ is introduced for simplifying probabilistic calculations and is applied in accordance with Asm B.1(a). The definition of $\boldsymbol{x}^{(l)}$, given in Eq. (A16), is slightly different from the original definition in [105]. Therefore, we will derive the basic probabilistic properties of pre- and post-activation again, based on Eq. (A16). Following a similar approach to [71, 105], the mean squared pre- and post-activation can be calculated as follows:

$$\mathbb{E}[(h_i^{(l)})^2] = \sigma_w^2\mathbb{E}[(x_i^{(l-1)})^2] + \sigma_b^2, \qquad \mathbb{E}[(x_i^{(l)})^2] = \mathbb{E}[(x_i^{(l-1)})^2] + \mathbb{E}[\phi(h_i^{(l)})^2]. \qquad \text{(A17)}$$

Additionally, following [81, 105], the mean squared gradient with respect to pre- and post-activation can be calculated as follows:

$$\mathbb{E}\left[\left(\frac{\partial \mathcal{L}}{\partial h_i^{(l)}}\right)^2\right] = \sigma_w^2 \mathbb{E}[\phi'(h_i^{(l)})^2]\mathbb{E}\left[\left(\frac{\partial \mathcal{L}}{\partial h_i^{(l+1)}}\right)^2\right], \tag{A18}$$

$$\chi^{(l)} = (1 + \sigma_w^2 \mathbb{E}[\phi'(h_i^{(l+1)})^2])\chi^{(l+1)}. \tag{A19}$$

We can derive Eq. (A18) under Asm B.1 as follows:

$$\mathbb{E}\left[\left(\frac{\partial \mathcal{L}}{\partial h_i^{(l)}}\right)^2\right] = \mathbb{E}\left[\left(\frac{\partial \mathcal{L}}{\partial \boldsymbol{h}^{(l+1)}} \frac{\partial \boldsymbol{h}^{(l+1)}}{\partial x_i^{(l)}} \frac{\partial x_i^{(l)}}{\partial h_i^{(l)}}\right)^2\right] \tag{A20}$$

$$= \mathbb{E}\left[\left(\sum_{j=1}^N \frac{\partial \mathcal{L}}{\partial h_j^{(l+1)}} W_{ji}^{(l+1)}\right)^2 \phi'(h_i^{(l)})^2\right] \tag{A21}$$

$$= \sigma_w^2 \mathbb{E}[\phi'(h_i^{(l)})^2]\mathbb{E}\left[\left(\frac{\partial \mathcal{L}}{\partial h_j^{(l+1)}}\right)^2\right]. \tag{A22}$$

We can derive Eq. (A19) under Asm B.1 as follows.

$$\chi^{(l)} := \mathbb{E}\left[\left(\frac{\partial \mathcal{L}}{\partial x_i^{(l)}}\right)^2\right] \tag{A23}$$

$$= \mathbb{E}\left[\left(\frac{\partial \mathcal{L}}{\partial x_i^{(l+1)}}\right)^2\right] + 2\mathbb{E}\left[\sum_{j=1}^N \sum_{k=1}^N \frac{\partial \mathcal{L}}{\partial x_i^{(l+1)}} \frac{\partial \mathcal{L}}{\partial x_j^{(l+1)}} P_{jk}^{(l+1)} \phi'(h_k^{(l+1)}) W_{ki}^{(l+1)}\right]$$

$$+ \mathbb{E}\left[\sum_{j=1}^N \sum_{k=1}^N \sum_{j'=1}^N \sum_{k'=1}^N \frac{\partial \mathcal{L}}{\partial x_j^{(l+1)}} \frac{\partial \mathcal{L}}{\partial x_{j'}^{(l+1)}} P_{jk}^{(l+1)} P_{j'k'}^{(l+1)} \phi'(h_k^{(l+1)}) \phi'(h_{k'}^{(l+1)}) W_{ki}^{(l+1)} W_{k'i}^{(l+1)}\right] \tag{A24}$$

$$= (1 + \sigma_w^2 \mathbb{E}[\phi'(h_k^{(l+1)})^2])\chi^{(l+1)}. \tag{A25}$$

From Eq. (A23) to Eq. (A24), we used the following equation:

$$\frac{\partial \mathcal{L}}{\partial x_i^{(l)}} = \sum_{j=1}^N \frac{\partial \mathcal{L}}{\partial x_j^{(l+1)}} \frac{\partial x_j^{(l+1)}}{\partial x_i^{(l)}} \tag{A26}$$

$$= \sum_{j=1}^N \frac{\partial \mathcal{L}}{\partial x_j^{(l+1)}} \left(\delta_{ij} + \sum_{k=1}^N P_{jk}^{(l+1)} \frac{\partial \phi(h_k^{(l+1)})}{\partial x_i^{(l)}}\right) \tag{A27}$$

$$= \frac{\partial \mathcal{L}}{\partial x_i^{(l+1)}} + \sum_{j=1}^N \sum_{k=1}^N \frac{\partial \mathcal{L}}{\partial x_j^{(l+1)}} P_{jk}^{(l+1)} \phi'(h_k^{(l+1)}) W_{ki}^{(l+1)}. \tag{A28}$$

The second term of Eq. (A24) is rearranged as follows:

$$2\mathbb{E}\left[\sum_{j=1}^N \sum_{k=1}^N \frac{\partial \mathcal{L}}{\partial x_i^{(l+1)}} \frac{\partial \mathcal{L}}{\partial x_j^{(l+1)}} P_{jk}^{(l+1)} \phi'(h_k^{(l+1)}) W_{ki}^{(l+1)}\right]$$

$$= 2\sum_{j=1}^N \sum_{k=1}^N \mathbb{E}[P_{jk}^{(l+1)}]\mathbb{E}\left[\frac{\partial \mathcal{L}}{\partial x_i^{(l+1)}} \frac{\partial \mathcal{L}}{\partial x_j^{(l+1)}} \phi'(h_k^{(l+1)}) W_{ki}^{(l+1)}\right] \tag{A29}$$

$$= 0. \tag{A30}$$

The third term of Eq. (A24) is rearranged as follows:

$$\mathbb{E}\left[\sum_{j=1}^{N}\sum_{k=1}^{N}\sum_{j'=1}^{N}\sum_{k'=1}^{N}\frac{\partial\mathcal{L}}{\partial x_j^{(l+1)}}\frac{\partial\mathcal{L}}{\partial x_{j'}^{(l+1)}}P_{jk}^{(l+1)}P_{j'k'}^{(l+1)}\phi'(h_k^{(l+1)})\phi'(h_{k'}^{(l+1)})W_{ki}^{(l+1)}W_{k'i}^{(l+1)}\right]$$

$$=\sum_{j=1}^{N}\sum_{k=1}^{N}\mathbb{E}\left[\left(\frac{\partial\mathcal{L}}{\partial x_j^{(l+1)}}\right)^2(P_{jk}^{(l+1)})^2\phi'(h_k^{(l+1)})^2(W_{ki}^{(l+1)})^2\right] \tag{A31}$$

$$=\sum_{j=1}^{N}\sum_{k=1}^{N}\mathbb{E}\left[\left(\frac{\partial\mathcal{L}}{\partial x_j^{(l+1)}}\right)^2\right]\mathbb{E}[(P_{jk}^{(l+1)})^2]\mathbb{E}[\phi'(h_k^{(l+1)})^2]\mathbb{E}[(W_{ki}^{(l+1)})^2] \tag{A32}$$

$$=\sigma_w^2\mathbb{E}[\phi'(h_k^{(l+1)})^2]\chi^{(l+1)}. \tag{A33}$$

## D  Sketch of proof for Thm 4.1

In this section, we provide a plain but informal proof of Thm 4.1 for the stepping stone to the formal proof. We present two levels of explanation: the most straightforward one and the other that is closer to a formal proof.

First, we introduce the simplest proof of counterintuitive independence of the distribution of $J(x^{\mathrm{in}})$ from $x^{\mathrm{in}}$. As indicated in Eq. (6), the definition of $J(x^{\mathrm{in}})$ includes $\phi'(h^{(l)}(x^{\mathrm{in}}))$; thus, the distribution appears to depend $x^{\mathrm{in}}$. Here, we consider the distribution of $\phi'(h(x))$, where $h(x) := wx$ with $w \sim \mathcal{N}(0, \sigma_w^2)$ and $x \in \mathbb{R}$. From the following proposition, although $\phi'(h(x))$ is defined as a function of $x$, its distribution is independent of $x$. This characteristic holds even for $J(x^{\mathrm{in}})$, which encompasses multiple $\phi'(h^{(l)}(x^{\mathrm{in}}))$.

**Proposition D.1.** *Let $\phi(z) := uz \ (z \geq 0); vz \ (z < 0)$ be a ReLU-like function, $w \sim \mathcal{N}(0, \sigma_w^2)$ be a Gaussian variable, and $x \in \mathbb{R}$ be a fixed real number. Then, the distribution of $\phi'(wx)$ is independent of $x$.*

*Proof.* Consider the derivative of $\phi$, given by $\phi'(z) := u \ (z \geq 0); v \ (z < 0)$. The input to $\phi'$, i.e., $wx$, follows a Gaussian $\mathcal{N}(0, x^2\sigma_w^2)$. The probability of a zero-mean Gaussian being greater than or equal to zero is the same as it being less than zero, regardless of the value of $x$. Therefore, for any given $x$, the probabilities of $\phi'(wx) = u$ and $\phi'(wx) = v$ are invariant to $x$. Consequently, the claim is established. $\square$

Then, let us consider Thm 4.1 in a neural network with one activation and one weight layer, respectively, as $f(x) := P^{\mathrm{out}}\phi(W^{(1)}P^{\mathrm{in}}x^{\mathrm{in}})$. In this setting, the network Jacobian is represented by $J(x^{\mathrm{in}}) := P^{\mathrm{out}}D(\phi'(W^{(1)}P^{\mathrm{in}}x^{\mathrm{in}}))W^{(1)}P^{\mathrm{in}}$. Moreover, we assume that the uncorrelated Gaussian variables are independent. **This assumption is incorrect**. Uncorrelated Gaussian variables are not necessarily independent (cf. Remark E.3). However, for the intuition about the formal proof, we assume this. The simplified Thm 4.1 and its proof are as follows:

**Proposition D.2.** *Consider a neural network $f(x) := P^{\mathrm{out}}\phi(W^{(1)}P^{\mathrm{in}}x^{\mathrm{in}})$. Suppose that the width $N$ is sufficiently large. Assume that uncorrelated Gaussian variables are independent. Then, for any $x^{\mathrm{in}} \in \mathbb{R}^d$, each entry of $J(x^{\mathrm{in}}) := P^{\mathrm{out}}D(\phi'(W^{(1)}P^{\mathrm{in}}x^{\mathrm{in}}))W^{(1)}P^{\mathrm{in}}$ is i.i.d. and follows the Gaussian $\mathcal{N}(0, \alpha\sigma_w^2/d)$.*

*Proof.* First, we prove that each entry of $J(x^{\mathrm{in}})$ is i.i.d. and follows a Gaussian. To this end, we consider the probabilistic properties in the order of $P^{\mathrm{in}}x^{\mathrm{in}}$, $D(\phi'(W^{(1)}P^{\mathrm{in}}x^{\mathrm{in}}))$, $W^{(1)}P^{\mathrm{in}}$, and $P^{\mathrm{out}}D(\phi'(W^{(1)}P^{\mathrm{in}}x^{\mathrm{in}}))W^{(1)}P^{\mathrm{in}}$. Then, we derive the concrete distribution, i.e., its mean and variance. Comparing the distributions between $f(x^{\mathrm{in}})^2$ and $(J(x^{\mathrm{in}})x^{\mathrm{in}})^2$, we achieve this.

We first consider $P^{\mathrm{in}}x^{\mathrm{in}} =: x^{(0)}$. Recall that $x^{\mathrm{in}}$ is a deterministic vector and $P^{\mathrm{in}}$ is a random matrix where each entry is i.i.d. and follows a Gaussian. Since the weighted sum of independent Gaussian variables follows a Gaussian, each entry of $x^{(0)}$ is i.i.d. and follows a Gaussian.

Then, let us consider $\boldsymbol{W}^{(1)}\boldsymbol{x}^{(0)}$. The $i$-th entry of $\boldsymbol{W}^{(1)}\boldsymbol{x}^{(0)}$ is expressed as $\sum_{j=1}^{N} W_{ij}^{(1)} x_j^{(0)}$. Since $\boldsymbol{W}^{(1)}$ is a random matrix with i.i.d. entries, $W_{ij}^{(1)} x_j^{(0)}$ is i.i.d. with respect to $j$. By the central limit theorem with infinite $N$, the $i$-th entry of $\boldsymbol{W}^{(1)}\boldsymbol{x}^{(0)}$ follows a Gaussian. In addition, the entries of $\boldsymbol{W}^{(1)}\boldsymbol{x}^{(0)}$ follow the same Gaussian. Moreover, since $\mathbb{E}[(\sum_{j=1}^{N} W_{ij}^{(1)} x_j^{(0)})(\sum_{j=1}^{N} W_{i'j}^{(1)} x_j^{(0)})] = 0$ holds for $i \neq i'$, different entries of $\boldsymbol{W}^{(1)}\boldsymbol{x}^{(0)}$ are uncorrelated and thus independent (this is originally incorrect but guaranteed by the assumption here). In conclusion, $\boldsymbol{W}^{(1)}\boldsymbol{x}^{(0)}$ is a random vector with i.i.d. Gaussian entries. Naturally, $\boldsymbol{D}(\phi'(\boldsymbol{W}^{(1)}\boldsymbol{P}^{\mathrm{in}}\boldsymbol{x}^{\mathrm{in}}))$ is a diagonal matrix with i.i.d. entries.

Similar to the discussion above, $\boldsymbol{W}^{(1)}\boldsymbol{P}^{\mathrm{in}} =: \boldsymbol{A}$ is a random matrix with i.i.d. entries.

Finally, consider $\boldsymbol{P}^{\mathrm{out}}\boldsymbol{D}(\phi'(\boldsymbol{W}^{(1)}\boldsymbol{P}^{\mathrm{in}}\boldsymbol{x}^{\mathrm{in}}))\boldsymbol{W}^{(1)}\boldsymbol{P}^{\mathrm{in}}$. For notational simplicity, we denote $\boldsymbol{D} := \boldsymbol{D}(\phi'(\boldsymbol{W}^{(1)}\boldsymbol{P}^{\mathrm{in}}\boldsymbol{x}^{\mathrm{in}}))$. Thus, $\boldsymbol{P}^{\mathrm{out}}\boldsymbol{D}(\phi'(\boldsymbol{W}^{(1)}\boldsymbol{P}^{\mathrm{in}}\boldsymbol{x}^{\mathrm{in}}))\boldsymbol{W}^{(1)}\boldsymbol{P}^{\mathrm{in}} = \boldsymbol{P}^{\mathrm{out}}\boldsymbol{D}\boldsymbol{A}$. The entry of $i$-th row and $j$-th column of this matrix is expressed as $\sum_{k=1}^{N} P_{ik}^{\mathrm{out}} D_k A_{kj}$. Note that $\boldsymbol{P}^{\mathrm{out}}$ is independent of $\boldsymbol{D}\boldsymbol{A}$ and its entries are i.i.d. Moreover, based on network symmetry, the dependence between $D_k$ and $A_{kj}$ is invariant to $k$. Thus, $P_{ik}^{\mathrm{out}} D_k A_{kj}$ is i.i.d. with respect to $k$ and each entry of this matrix follows the same Gaussian by the central limit theorem. Moreover, since $\mathbb{E}[(\sum_{k=1}^{N} P_{ik}^{\mathrm{out}} D_k A_{kj})(\sum_{k=1}^{N} P_{i'k}^{\mathrm{out}} D_k A_{kj'})] = 0$ holds for $(i,j) \neq (i',j')$, different entries of $\boldsymbol{P}^{\mathrm{out}}\boldsymbol{D}\boldsymbol{A}$ are uncorrelated and thus independent. Therefore, each entry of $\boldsymbol{J}(\boldsymbol{x}^{\mathrm{in}})$ is i.i.d. and follows the Gaussian.

Next, we derive the mean and variance of $\boldsymbol{J}(\boldsymbol{x}^{\mathrm{in}})$. Trivially, $\mathbb{E}[\boldsymbol{J}(\boldsymbol{x}^{\mathrm{in}})_i] = 0$ since the mean of an entry of $\boldsymbol{P}^{\mathrm{out}}$ is zero. We derive the variance by comparing $\mathbb{E}[f(\boldsymbol{x})_i^2]$ and $\mathbb{E}[(\sum_{j=1}^{d} \boldsymbol{J}(\boldsymbol{x}^{\mathrm{in}})_{ij} x_j^{\mathrm{in}})^2]$. Recall $f(\boldsymbol{x})_i := \sum_{j=1}^{d} \boldsymbol{J}(\boldsymbol{x}^{\mathrm{in}})_{ij} x_j^{\mathrm{in}}$. As a preliminary, with a Gaussian variable $z \sim \mathcal{N}(0, \sigma^2)$ and its probabilistic density function $g(z)$, we derive the following equation:

$$\mathbb{E}_{z \sim \mathcal{N}(0,\sigma^2)}[\phi(z)^2] = \int_{-\infty}^{\infty} \phi(z)^2 g(z; \sigma^2) dz \tag{A34}$$

$$= \int_{0}^{\infty} u^2 z^2 g(z; \sigma^2) dz + \int_{-\infty}^{0} v^2 z^2 g(z; \sigma^2) dz \tag{A35}$$

$$= \frac{u^2}{2}\sigma^2 + \frac{v^2}{2}\sigma^2 \tag{A36}$$

$$= \alpha \sigma^2. \tag{A37}$$

In addition, if $\boldsymbol{P}^{\mathrm{out}}$ and $z$ are independent, then

$$\mathbb{E}_{z \sim \mathcal{N}(0,\sigma^2)}\left[\left(\sum_{j=1}^{N} P_{ij}^{\mathrm{out}} \phi(z)_j\right)^2\right] = \sum_{j=1}^{N} \mathbb{E}[(P_{ij}^{\mathrm{out}})^2]\mathbb{E}[\phi(z)_j^2] = \alpha \sigma^2. \tag{A38}$$

As shown in the discussion above, each entry of $\boldsymbol{W}^{(1)}\boldsymbol{x}^{(0)}$ follows the Gaussian. Moreover, we can simply derive its variance as $\sigma_w^2 \|\boldsymbol{x}^{\mathrm{in}}\|^2 / d$. Thus,

$$\mathbb{E}[f(\boldsymbol{x})_i^2] = \mathbb{E}\left[\left(\sum_{j=1}^{N} P_{ij}^{\mathrm{out}} \phi(\boldsymbol{W}^{(1)}\boldsymbol{x}^{(0)})_j\right)^2\right] = \alpha \sigma_w^2 \|\boldsymbol{x}^{\mathrm{in}}\|^2 / d. \tag{A39}$$

Since $f(\boldsymbol{x})_i := \sum_{j=1}^{d} \boldsymbol{J}(\boldsymbol{x}^{\mathrm{in}})_{ij} x_j^{\mathrm{in}}$, $\mathbb{E}[f(\boldsymbol{x})_i^2]$ is also expanded as:

$$\mathbb{E}[f(\boldsymbol{x})_i^2] = \mathbb{E}\left[\left(\sum_{j=1}^{d} \boldsymbol{J}(\boldsymbol{x}^{\mathrm{in}})_{ij} x_j^{\mathrm{in}}\right)^2\right] = \sum_{j=1}^{d} \mathbb{E}[\boldsymbol{J}(\boldsymbol{x}^{\mathrm{in}})_{ij}^2](x_j^{\mathrm{in}})^2 = \mathbb{E}[\boldsymbol{J}(\boldsymbol{x}^{\mathrm{in}})_{ij}^2]\|\boldsymbol{x}^{\mathrm{in}}\|^2. \tag{A40}$$

Comparing the two equations above, we obtain $\mathbb{E}[\boldsymbol{J}(\boldsymbol{x}^{\mathrm{in}})_{ij}^2] = \alpha \sigma_w^2 / d$. Therefore, the claim is established. $\qquad \square$

# E  Derivation of Thm 4.1 for vanilla networks

## E.1  Preliminary

First, we introduce some lemmas. These results are more general and not restricted to the context of neural networks. The notation in this section is independent of that in other sections.

We refer to a random matrix where the mean of an entry is zero as a zero-mean matrix. Formally, this is defined as follows:

**Definition E.1** (zero-mean matrix/vector)**.**  A random matrix/vector $A$ is called a zero-mean matrix/vector if $\mathbb{E}[A_{ij}] = 0$ for any $i$ and $j$.

For zero-mean matrices, the following properties hold:
*Remark* E.2 (Properties of zero-mean matrix)*.*  Let $A$ be a zero-mean matrix.

- Let $B$ be a zero-mean matrix. Then, their sum, $A + B$, is a zero-mean matrix.

- Let $B$ be a random matrix. If $A$ and $B$ are independent, then their products, $AB$ and $BA$, are zero-mean matrices.

Then, we review several properties of Gaussian variables.
*Remark* E.3 (Properties of Gaussian variables)*.*  The following statements hold:

- Any linear combination of multiple Gaussian variables follows a Gaussian if and only if they are jointly distributed Gaussian.

- A weighted sum of independent Gaussian variables is Gaussian distributed.

- Gaussian variables are jointly distributed and uncorrelated if and only if they are independent.

Next, we examine the properties of random matrices/vectors with i.i.d. Gaussian entries, called a Gaussian matrix/vector. Their formal definitions are as follows:

**Definition E.4** (Gaussian matrix/vector)**.**  A matrix/vector is called a Gaussian matrix/vector if all the entries are i.i.d. and follow a Gaussian.

Gaussian matrices and vectors satisfy the following lemmas.

**Lemma E.5.**  *If $A$ and $B$ are independent Gaussian matrices, then $A + B$ is a Gaussian matrix. In particular, when $A$ and $B$ are zero-mean, $A + B$ is a zero-mean Gaussian matrix.*

*Proof.*  By Remarks E.2 and E.3, this is trivial.  □

The following lemmas are derived for the proof of Thm 4.1.

**Lemma E.6.**  *Suppose that random matrices $A \in \mathbb{R}^{m \times n}$ and $B \in \mathbb{R}^{n \times l}$ satisfy the following properties: (a) $A$ and $B$ are independent. (b) $A$ is a zero-mean matrix. (c) All the entries of $A$ are i.i.d. (d) All the entries of $B$ are identically distributed. (e) Any two entries of $B$ from different rows are independent. (f) For any $i \in [n], j, k \in [l], j \neq k$, the dependence between $B_{ij}$ and $B_{ik}$ is invariant to $i, j, k$. (g) For any $i \in [n], j, k \in [l], j \neq k, \mathbb{E}[B_{ij}B_{ik}] = 0$ holds. Then, for a sufficiently large $n$, $AB$ is a zero-mean Gaussian matrix.*

*Proof.*  By (a), (b), and Remark E.2, $AB$ is a zero-mean matrix. The linear combination of all the entries of $AB$ is expressed as

$$c := \sum_{i=1}^{m} \sum_{j=1}^{l} s_{ij} A_{i.} B_{.j} = \sum_{i=1}^{m} \sum_{j=1}^{l} s_{ij} \sum_{k=1}^{n} A_{ik} B_{kj} = \sum_{k=1}^{n} \left( \sum_{i=1}^{m} \sum_{j=1}^{l} s_{ij} A_{ik} B_{kj} \right), \qquad \text{(A41)}$$

where $s_{ij} \in \mathbb{R}$ is a constant. By (a), (c), (d), (e), and (f), $\sum_{i=1}^{m} \sum_{j=1}^{l} s_{ij} A_{ik} B_{kj}$ is i.i.d. with respect to $k$. By the central limit theorem, $c$ follows a Gaussian. By Remark E.3, all the entries of $AB$ are jointly Gaussian distributed. Moreover, by (c) and (d), the entries of $AB$ follow the same Gaussian. By (a), (b), (c), (e), and (g), the covariance between any two entries of $AB$ is zero. Thus, by Remark E.3, the claim is established.  □

**Lemma E.7.** *Suppose that random matrices $\boldsymbol{A} \in \mathbb{R}^{m \times n}$, $\boldsymbol{B} \in \mathbb{R}^{n \times l_1}$, and $\boldsymbol{C} \in \mathbb{R}^{n \times l_2}$ satisfy the following properties: (a) $\boldsymbol{A}$, $\boldsymbol{B}$, and $\boldsymbol{C}$ are independent. (b) $\boldsymbol{A}$ is a zero-mean matrix. (c) All the entries of $\boldsymbol{A}$ are i.i.d. (d) All the entries of $\boldsymbol{B}$ are identically distributed. (e) All the entries of $\boldsymbol{C}$ are identically distributed. (f) Any two entries of $\boldsymbol{B}$ from different rows are independent. (g) Any two entries of $\boldsymbol{C}$ from different rows are independent. (h) For any $i \in [n], j, k \in [l_1], j \neq k$, the dependence between $B_{ij}$ and $B_{ik}$ is invariant to $i, j, k$. (i) For any $i \in [n], j, k \in [l_2], j \neq k$, the dependence between $C_{ij}$ and $C_{ik}$ is invariant to $i, j, k$. (j) For any $i \in [n], j \in [l_1], k \in [l_2]$, $\mathbb{E}[B_{ij} C_{ik}] = 0$ holds. Then, for a sufficiently large $n$, $\boldsymbol{AB}$ and $\boldsymbol{AC}$ are independent.*

*Proof.* By (a), (c), (d), (e), (f), (g), (h), and (i), similar to Lemma E.6, all the entries of $\boldsymbol{AB}$ and $\boldsymbol{AC}$ are jointly Gaussian distributed. By (a), (b), (c), and (j), the covariance between any two entries from $\boldsymbol{AB}$ and $\boldsymbol{AC}$ is zero. Thus, by Remark E.3, the claim is established. □

**Lemma E.8.** *Let $\boldsymbol{w}, \boldsymbol{x} \in \mathbb{R}^m$ be Gaussian vectors. Let $\phi'(z) := u \; (z \geq 0); v \; (z < 0)$. Then, for a sufficiently large $m$, $\phi'(\boldsymbol{w}^\top \boldsymbol{x}) \boldsymbol{w}$ and $\boldsymbol{x}$ are independent.*

*Proof.* We prove $\mathbb{P}[\phi'(\boldsymbol{w}^\top \boldsymbol{x}) \boldsymbol{w} = \boldsymbol{a} \mid \boldsymbol{x} = \boldsymbol{b}] = \mathbb{P}[\phi'(\boldsymbol{w}^\top \boldsymbol{x}) \boldsymbol{w} = \boldsymbol{a}]$. For any $\boldsymbol{x}$, $\boldsymbol{w} = \boldsymbol{a}/u + \boldsymbol{x}/\sqrt{m}$ or $\boldsymbol{w} = \boldsymbol{a}/v - \boldsymbol{x}/\sqrt{m}$ satisfy $\phi'(\boldsymbol{w}^\top \boldsymbol{x}) \boldsymbol{w} = \boldsymbol{a}$. Since $m$ is sufficiently large, $\boldsymbol{w} = \boldsymbol{a}/u + \boldsymbol{x}/\sqrt{m}$ and $\boldsymbol{w} = \boldsymbol{a}/v - \boldsymbol{x}/\sqrt{m}$ asymptotically approach $\boldsymbol{w} = \boldsymbol{a}/u$ and $\boldsymbol{w} = \boldsymbol{a}/v$, respectively. Thus, the claim is established. □

## E.2 Definitions of $\boldsymbol{J}(\boldsymbol{x}^{\mathrm{in}})$ and $\boldsymbol{a}(\boldsymbol{x}^{\mathrm{in}})$ for vanilla networks

Here, we define (derive) $\boldsymbol{J}(\boldsymbol{x}^{\mathrm{in}})$ and $\boldsymbol{a}(\boldsymbol{x}^{\mathrm{in}})$ for vanilla networks. The $l$-th pre- and post-activation are defined as follows:

$$\boldsymbol{h}^{(l)}(\boldsymbol{x}^{\mathrm{in}}) := \boldsymbol{W}^{(l)} \boldsymbol{x}^{(l-1)}(\boldsymbol{x}^{\mathrm{in}}) + \boldsymbol{b}^{(l)}, \qquad \boldsymbol{x}^{(l)}(\boldsymbol{x}^{\mathrm{in}}) := \phi(\boldsymbol{h}^{(l)}(\boldsymbol{x}^{\mathrm{in}})). \tag{A42}$$

Using $\phi(\boldsymbol{h}^{(l)}(\boldsymbol{x}^{\mathrm{in}})) = \boldsymbol{D}(\phi'(\boldsymbol{h}^{(l)}(\boldsymbol{x}^{\mathrm{in}}))) \boldsymbol{h}^{(l)}(\boldsymbol{x}^{\mathrm{in}})$, the $l$-th pre-activation can be rearranged as follows:

$$\boldsymbol{h}^{(l)}(\boldsymbol{x}^{\mathrm{in}}) = \boldsymbol{W}^{(l)} \boldsymbol{D}(\phi'(\boldsymbol{h}^{(l-1)}(\boldsymbol{x}^{\mathrm{in}}))) \boldsymbol{h}^{(l-1)}(\boldsymbol{x}^{\mathrm{in}}) + \boldsymbol{b}^{(l)}. \tag{A43}$$

Because ReLU-like networks are piecewise linear, the $l$-th pre-activation is also represented as follows:

$$\boldsymbol{h}^{(l)}(\boldsymbol{x}^{\mathrm{in}}) = \boldsymbol{J}^{(l)}(\boldsymbol{x}^{\mathrm{in}}) \boldsymbol{x}^{(0)}(\boldsymbol{x}^{\mathrm{in}}) + \boldsymbol{a}^{(l)}(\boldsymbol{x}^{\mathrm{in}}). \tag{A44}$$

Substituting Eq. (A44) for Eq. (A43), the equation can be rearranged as follows:

$$\begin{aligned}
\boldsymbol{h}^{(l)}(\boldsymbol{x}^{\mathrm{in}}) = {} & \boldsymbol{W}^{(l)} \boldsymbol{D}(\phi'(\boldsymbol{h}^{(l-1)}(\boldsymbol{x}^{\mathrm{in}}))) \boldsymbol{J}^{(l-1)}(\boldsymbol{x}^{\mathrm{in}}) \boldsymbol{x}^{(0)}(\boldsymbol{x}^{\mathrm{in}}) \\
& + \boldsymbol{W}^{(l)} \boldsymbol{D}(\phi'(\boldsymbol{h}^{(l-1)}(\boldsymbol{x}^{\mathrm{in}}))) \boldsymbol{a}^{(l-1)}(\boldsymbol{x}^{\mathrm{in}}) + \boldsymbol{b}^{(l)}.
\end{aligned} \tag{A45}$$

Comparing Eq. (A44) and Eq. (A45), the following equations can be derived:

$$\boldsymbol{J}^{(l)}(\boldsymbol{x}^{\mathrm{in}}) := \boldsymbol{W}^{(l)} \boldsymbol{D}(\phi'(\boldsymbol{h}^{(l-1)}(\boldsymbol{x}^{\mathrm{in}}))) \boldsymbol{J}^{(l-1)}(\boldsymbol{x}^{\mathrm{in}}), \tag{A46}$$

$$\boldsymbol{a}^{(l)}(\boldsymbol{x}^{\mathrm{in}}) := \boldsymbol{W}^{(l)} \boldsymbol{D}(\phi'(\boldsymbol{h}^{(l-1)}(\boldsymbol{x}^{\mathrm{in}}))) \boldsymbol{a}^{(l-1)}(\boldsymbol{x}^{\mathrm{in}}) + \boldsymbol{b}^{(l)}, \tag{A47}$$

where $\boldsymbol{J}^{(1)}(\boldsymbol{x}^{\mathrm{in}}) := \boldsymbol{W}^{(1)}$ and $\boldsymbol{a}^{(1)}(\boldsymbol{x}^{\mathrm{in}}) := \boldsymbol{b}^{(1)}$. Finally, $\boldsymbol{J}(\boldsymbol{x}^{\mathrm{in}})$ and $\boldsymbol{a}(\boldsymbol{x}^{\mathrm{in}})$ are defined as follows:

$$\boldsymbol{f}(\boldsymbol{x}^{\mathrm{in}}) = \boldsymbol{J}(\boldsymbol{x}^{\mathrm{in}}) \boldsymbol{x}^{\mathrm{in}} + \boldsymbol{a}(\boldsymbol{x}^{\mathrm{in}}), \tag{5}$$

$$\boldsymbol{J}(\boldsymbol{x}^{\mathrm{in}}) := \boldsymbol{P}^{\mathrm{out}} \boldsymbol{D}(\phi'(\boldsymbol{h}^{(L)}(\boldsymbol{x}^{\mathrm{in}}))) \boldsymbol{J}^{(L)}(\boldsymbol{x}^{\mathrm{in}}) \boldsymbol{P}^{\mathrm{in}}, \tag{6}$$

$$\boldsymbol{a}(\boldsymbol{x}^{\mathrm{in}}) := \boldsymbol{P}^{\mathrm{out}} \boldsymbol{D}(\phi'(\boldsymbol{h}^{(L)}(\boldsymbol{x}^{\mathrm{in}}))) \boldsymbol{a}^{(L)}(\boldsymbol{x}^{\mathrm{in}}). \tag{A48}$$

## E.3 Main derivation

First, we remark several fundamental properties of weights, biases, random projections, and pre-activations in a network.

*Remark* E.9. By definition in Sec. 3.1, the following statements hold for any $l \in [L]$:

- (Gaussian matrix/vector) $\boldsymbol{W}^{(l)}$, $\boldsymbol{P}^{\text{in}}$, and $\boldsymbol{P}^{\text{out}}$ are zero-mean Gaussian matrices and $\boldsymbol{b}^{(l)}$ is a zero-mean Gaussian vector.

- (Variance) The variance of $\boldsymbol{W}^{(l)}$, $\boldsymbol{P}^{\text{in}}$, and $\boldsymbol{P}^{\text{out}}$ is $\sigma_w^2/N$, $1/d$, and $1/N$, respectively.

- (Independence) $\boldsymbol{W}^{(l)}$ and $\boldsymbol{b}^{(l)}$ are independent of each other, as is any random variable in the layer before the $l$-th. In addition, $\boldsymbol{P}^{\text{out}}$ is independent of any random variable without a network output.

- (Dependence between pre-activation and bias) For $i = j$, the dependence between $h_i^{(l)}$ and $b_i^{(l)}$ is invariant to $i$. For $i \neq j$, $h_i^{(l)}$ and $b_j^{(l)}$ are independent.

**Lemma E.10.** *Consider a vanilla network. For any $l \in [L]$, the following hold:*

*(a)* $\boldsymbol{J}^{(l)}(\boldsymbol{x}^{\text{in}})$ *is a zero-mean Gaussian matrix.*

*(b)* $\boldsymbol{a}^{(l)}(\boldsymbol{x}^{\text{in}})$ *is a zero-mean Gaussian vector.*

*(c)* $\boldsymbol{J}^{(l)}(\boldsymbol{x}^{\text{in}})$ *and* $\boldsymbol{a}^{(l)}(\boldsymbol{x}^{\text{in}})$ *are independent.*

*Proof.* We prove the claim by induction. As a preliminary, see Remark E.9. The case with $l = 1$ is trivial. Suppose that all the claims hold for $l - 1 \in [L - 1]$.

**(a)** By Lemma E.6, the claim is established. Note that $\mathbb{E}[\phi'(h_i^{(l)})^2 \boldsymbol{J}^{(l)}(\boldsymbol{x}^{\text{in}})_{ij} \boldsymbol{J}^{(l)}(\boldsymbol{x}^{\text{in}})_{ik}] = \mathbb{E}[\phi'(h_i^{(l)})^2]\mathbb{E}[\boldsymbol{J}^{(l)}(\boldsymbol{x}^{\text{in}})_{ij}]\mathbb{E}[\boldsymbol{J}^{(l)}(\boldsymbol{x}^{\text{in}})_{ik}] = 0$ since $N$ is sufficiently large.

**(b)** By Lemma E.6, the claim is established.

**(c)** By Lemma E.7, the claim is established.

$\square$

**Lemma E.11.** *The mean squared network output is given by:*

$$\mathbb{E}[f(\boldsymbol{x}^{\text{in}})_i^2] = \frac{\omega^L}{d}\|\boldsymbol{x}^{\text{in}}\|_2^2 + \alpha\sigma_b^2 \sum_{i=1}^{L} \omega^{i-1}. \tag{A49}$$

*Proof.* By Remark E.9, the mean squared pre- and post- activation are calculated as follows (cf. Eq. (A34)):

$$\mathbb{E}[(h_i^{(l)})^2] = \sigma_w^2 \mathbb{E}[(x_i^{(l-1)})^2] + \sigma_b^2, \tag{A50}$$

$$\mathbb{E}[(x_i^{(l)})^2] = \begin{cases} \alpha\mathbb{E}[(h_i^{(l)})^2] & \text{(vanilla)} \\ \mathbb{E}[(x_i^{(l-1)})^2] + \alpha\mathbb{E}[(h_i^{(l)})^2] & \text{(residual)} \end{cases}. \tag{A51}$$

Recursively calculating Eqs. (A50) and (A51), the mean squared post-activation in a vanilla network is calculated as follows:

$$\mathbb{E}[(x_i^{(l)})^2] = \alpha\mathbb{E}[(h_i^{(l)})^2] = \omega_{\text{v}}\mathbb{E}[(x_i^{(l-1)})^2] + \alpha\sigma_b^2 = \omega_{\text{v}}^l \mathbb{E}[(x_i^{(0)})^2] + \alpha\sigma_b^2 \sum_{i=1}^{l} \omega_{\text{v}}^{i-1}. \tag{A52}$$

The mean squared post-activation in a residual network is calculated as follows:

$$\mathbb{E}[(x_i^{(l)})^2] = \mathbb{E}[(x_i^{(l-1)})^2] + \alpha(\sigma_w^2 \mathbb{E}[(x_i^{(l-1)})^2] + \sigma_b^2) \tag{A53}$$

$$= \omega_{\text{r}}\mathbb{E}[(x_i^{(l-1)})^2] + \alpha\sigma_b^2 \tag{A54}$$

$$= \omega_{\text{r}}^l \mathbb{E}[(x_i^{(0)})^2] + \alpha\sigma_b^2 \sum_{i=1}^{l} \omega_{\text{r}}^{i-1}. \tag{A55}$$

Using $x^{(0)} := \boldsymbol{P}^{\text{in}}\boldsymbol{x}^{\text{in}}$, the above results are expanded as follows:

$$\mathbb{E}[(x_i^{(l)})^2] = \omega^l \sum_{j=1}^{d} \mathbb{E}[(P_{ij}^{\text{in}})^2](x_j^{\text{in}})^2 + \alpha\sigma_b^2 \sum_{i=1}^{l} \omega^{i-1} = \frac{\omega^l}{d}\|\boldsymbol{x}^{\text{in}}\|_2^2 + \alpha\sigma_b^2 \sum_{i=1}^{l} \omega^{i-1}. \tag{A56}$$

The mean squared network output is rearranged to the mean squared $L$-th post-activation as follows:

$$\mathbb{E}[f(\boldsymbol{x}^{\mathrm{in}})_i^2] = \sum_{j=1}^{N} \mathbb{E}[(P_{ij}^{\mathrm{out}})^2]\mathbb{E}[(x_j^{(L)})^2] = \mathbb{E}[(x_i^{(L)})^2]. \tag{A57}$$

By Eqs. (A56) and (A57), the claim is established. $\square$

**Theorem 4.1** (Properties and distributions of $\boldsymbol{J}(\boldsymbol{x}^{\mathrm{in}})$ and $\boldsymbol{a}(\boldsymbol{x}^{\mathrm{in}})$). *Suppose that the width $N$ is sufficiently large. Then, for any $\boldsymbol{x}^{\mathrm{in}} \in \mathbb{R}^d$, (I) $\boldsymbol{J}(\boldsymbol{x}^{\mathrm{in}})$ and $\boldsymbol{a}(\boldsymbol{x}^{\mathrm{in}})$ are independent. (II) each entry of $\boldsymbol{J}(\boldsymbol{x}^{\mathrm{in}})$ and $\boldsymbol{a}(\boldsymbol{x}^{\mathrm{in}})$ is i.i.d. and follows the Gaussian below:*

$$J(\boldsymbol{x}^{\mathrm{in}})_{ij} \sim \mathcal{N}\left(0, \frac{\omega^L}{d}\right), \qquad a(\boldsymbol{x}^{\mathrm{in}})_i \sim \mathcal{N}\left(0, \alpha\sigma_b^2 \sum_{k=1}^{L} \omega^{k-1}\right), \tag{7}$$

*where $\alpha := (u^2 + v^2)/2$ (cf. Defn 3.1) and $\omega$ is $\omega_{\mathrm{v}} := \alpha\sigma_w^2$ for vanilla networks and $\omega_{\mathrm{r}} := 1 + \alpha\sigma_w^2$ for residual networks.*

*Proof.* Similar to Lemma E.8, $\boldsymbol{P}^{\mathrm{in}}$ and $\boldsymbol{a}(\boldsymbol{x}^{\mathrm{in}})$ are independent. Similar to Lemmas E.10 and F.2, $\boldsymbol{J}(\boldsymbol{x}^{\mathrm{in}})$ and $\boldsymbol{a}(\boldsymbol{x}^{\mathrm{in}})$ are a zero-mean Gaussian matrix and vector, respectively, and $\boldsymbol{J}(\boldsymbol{x}^{\mathrm{in}})$ and $\boldsymbol{a}(\boldsymbol{x}^{\mathrm{in}})$ are independent. The mean squared network output can be expanded as follows:

$$\mathbb{E}[f(\boldsymbol{x}^{\mathrm{in}})_i^2]$$

$$=\mathbb{E}\left[\left(\sum_{j=1}^{d} J(\boldsymbol{x}^{\mathrm{in}})_{ij}x_j^{\mathrm{in}} + a(\boldsymbol{x}^{\mathrm{in}})_i\right)^2\right] \tag{A58}$$

$$=\sum_{j=1}^{d}\sum_{k=1}^{d} \mathbb{E}[J(\boldsymbol{x}^{\mathrm{in}})_{ij}J(\boldsymbol{x}^{\mathrm{in}})_{ik}]x_j^{\mathrm{in}}x_k^{\mathrm{in}} + 2\sum_{j=1}^{d} \mathbb{E}[J(\boldsymbol{x}^{\mathrm{in}})_{ij}a(\boldsymbol{x}^{\mathrm{in}})_i]x_j^{\mathrm{in}} + \mathbb{E}[a(\boldsymbol{x}^{\mathrm{in}})_i^2] \tag{A59}$$

$$=\sum_{j=1}^{d} \mathbb{E}[J(\boldsymbol{x}^{\mathrm{in}})_{ij}^2](x_j^{\mathrm{in}})^2 + \mathbb{E}[a(\boldsymbol{x}^{\mathrm{in}})_i^2]. \tag{A60}$$

Using the symmetry of entries of $\boldsymbol{J}(\boldsymbol{x}^{\mathrm{in}})$ and $\boldsymbol{a}(\boldsymbol{x}^{\mathrm{in}})$, the equation can be simplified as follows:

$$\mathbb{E}[f(\boldsymbol{x}^{\mathrm{in}})_i^2] = \mathbb{E}[J(\boldsymbol{x}^{\mathrm{in}})_{ij}^2]\big\|\boldsymbol{x}^{\mathrm{in}}\big\|_2^2 + \mathbb{E}[a(\boldsymbol{x}^{\mathrm{in}})_i^2]. \tag{A61}$$

Comparing Lemma E.11, Eq. (7) is obtained. Thus, the claim is established. $\square$

# F  Derivation of Thm 4.1 for residual networks

## F.1  Definitions of $J(\boldsymbol{x}^{\mathrm{in}})$ and $a(\boldsymbol{x}^{\mathrm{in}})$ for residual networks

Similar to Appx. E.2, we define (derive) $\boldsymbol{J}(\boldsymbol{x}^{\mathrm{in}})$ and $\boldsymbol{a}(\boldsymbol{x}^{\mathrm{in}})$ for residual networks. For notational simplicity, we omit the argument $\boldsymbol{x}^{\mathrm{in}}$. First, we represent $\boldsymbol{x}^{(l)}$ as follows:

$$\boldsymbol{x}^{(l)} = (\boldsymbol{I} + \boldsymbol{V}^{(l)})\boldsymbol{x}^{(0)} + \boldsymbol{c}^{(l)}, \tag{A62}$$

where $\boldsymbol{V}^{(0)} := \boldsymbol{0}$ and $\boldsymbol{c}^{(0)} := \boldsymbol{0}$. With $\boldsymbol{V}^{(l)}$ and $\boldsymbol{c}^{(l)}$, we can represent $\boldsymbol{h}^{(l)}$ as follows:

$$\boldsymbol{h}^{(l)} = \boldsymbol{W}^{(l)}\boldsymbol{x}^{(l-1)} + \boldsymbol{b}^{(l)} \tag{A63}$$

$$= \boldsymbol{W}^{(l)}((\boldsymbol{I} + \boldsymbol{V}^{(l-1)})\boldsymbol{x}^{(0)} + \boldsymbol{c}^{(l-1)}) + \boldsymbol{b}^{(l)} \tag{A64}$$

$$= \boldsymbol{W}^{(l)}(\boldsymbol{I} + \boldsymbol{V}^{(l-1)})\boldsymbol{x}^{(0)} + \boldsymbol{W}^{(l)}\boldsymbol{c}^{(l-1)} + \boldsymbol{b}^{(l)}. \tag{A65}$$

Then, we derive recurrence representations of $\boldsymbol{V}^{(l)}$ and $\boldsymbol{c}^{(l)}$ as follows:

$$\boldsymbol{x}^{(l)} =\boldsymbol{x}^{(l-1)} + \boldsymbol{P}^{(l)}\boldsymbol{\phi}(\boldsymbol{h}^{(l)}) \tag{A66}$$

$$=(\boldsymbol{I} + \boldsymbol{V}^{(l-1)})\boldsymbol{x}^{(0)} + \boldsymbol{c}^{(l-1)}$$

$$+ \boldsymbol{P}^{(l)}\boldsymbol{D}(\boldsymbol{\phi}'(\boldsymbol{h}^{(l)}))(\boldsymbol{W}^{(l)}(\boldsymbol{I} + \boldsymbol{V}^{(l-1)})\boldsymbol{x}^{(0)} + \boldsymbol{W}^{(l)}\boldsymbol{c}^{(l-1)} + \boldsymbol{b}^{(l)}). \tag{A67}$$

For reference, we denote

$$\boldsymbol{U}^{(l)} := \boldsymbol{P}^{(l)}\boldsymbol{D}(\phi'(\boldsymbol{h}^{(l)}))\boldsymbol{W}^{(l)}, \qquad\qquad \boldsymbol{d}^{(l)} := \boldsymbol{P}^{(l)}\boldsymbol{D}(\phi'(\boldsymbol{h}^{(l)}))\boldsymbol{b}^{(l)}. \qquad \text{(A68)}$$

With the notations above,

$$\boldsymbol{x}^{(l)} = (\boldsymbol{I} + \boldsymbol{V}^{(l-1)})\boldsymbol{x}^{(0)} + \boldsymbol{c}^{(l-1)} + \boldsymbol{U}^{(l)}(\boldsymbol{I} + \boldsymbol{V}^{(l-1)})\boldsymbol{x}^{(0)} + \boldsymbol{U}^{(l)}\boldsymbol{c}^{(l-1)} + \boldsymbol{d}^{(l)} \qquad \text{(A69)}$$

$$= (\boldsymbol{I} + \boldsymbol{U}^{(l)})(\boldsymbol{I} + \boldsymbol{V}^{(l-1)})\boldsymbol{x}^{(0)} + (\boldsymbol{I} + \boldsymbol{U}^{(l)})\boldsymbol{c}^{(l-1)} + \boldsymbol{d}^{(l)} \qquad \text{(A70)}$$

$$= (\boldsymbol{I} + \boldsymbol{U}^{(l)} + \boldsymbol{V}^{(l-1)} + \boldsymbol{U}^{(l)}\boldsymbol{V}^{(l-1)})\boldsymbol{x}^{(0)} + \boldsymbol{c}^{(l-1)} + \boldsymbol{U}^{(l)}\boldsymbol{c}^{(l-1)} + \boldsymbol{d}^{(l)}. \qquad \text{(A71)}$$

Thus,

$$\boldsymbol{V}^{(l)} := \boldsymbol{U}^{(l)} + \boldsymbol{V}^{(l-1)} + \boldsymbol{U}^{(l)}\boldsymbol{V}^{(l-1)}, \qquad \boldsymbol{c}^{(l)} := \boldsymbol{c}^{(l-1)} + \boldsymbol{U}^{(l)}\boldsymbol{c}^{(l-1)} + \boldsymbol{d}^{(l)}. \qquad \text{(A72)}$$

Finally, $\boldsymbol{J}(\boldsymbol{x}^{\mathrm{in}})$ and $\boldsymbol{a}(\boldsymbol{x}^{\mathrm{in}})$ are defined as follows:

$$\boldsymbol{J}(\boldsymbol{x}^{\mathrm{in}}) := \boldsymbol{P}^{\mathrm{out}}(\boldsymbol{I} + \boldsymbol{V}^{(L)})\boldsymbol{P}^{\mathrm{in}}, \qquad\qquad \boldsymbol{a}(\boldsymbol{x}^{\mathrm{in}}) := \boldsymbol{P}^{\mathrm{out}}\boldsymbol{c}^{(L)}. \qquad \text{(A73)}$$

## F.2  Main derivation

First, we note the properties of random projections in shortcuts.

*Remark* F.1.  By definition (cf. Appx. C), the following statements hold for any $l \in [L]$:

- (Gaussian matrix/vector) $\boldsymbol{P}^{(l)}$ is a zero-mean Gaussian matrix.

- (Variance) The variance of $\boldsymbol{P}^{(l)}$ is $1/N$.

- (Independence) $\boldsymbol{P}^{(l)}$ is independent of $\boldsymbol{W}^{(l)}$, $\boldsymbol{b}^{(l)}$, and $\boldsymbol{h}^{(l)}$, as is any random variable in the layer before the $l$-th.

Finally, we introduce a lemma similar to Lemma E.10. A subsequent discussion to derive Thm 4.1 is the same as the proof in Appx. E.3.

**Lemma F.2.** *Consider a residual network. For any $l \in [L]$, the following statements hold:*

*(a)* $\boldsymbol{J}^{(l)}(\boldsymbol{x}^{\mathrm{in}})$ *is a zero-mean Gaussian matrix.*

*(b)* $\boldsymbol{a}^{(l)}(\boldsymbol{x}^{\mathrm{in}})$ *is a zero-mean Gaussian vector.*

*(c)* $\boldsymbol{J}^{(l)}(\boldsymbol{x}^{\mathrm{in}})$ *and* $\boldsymbol{a}^{(l)}(\boldsymbol{x}^{\mathrm{in}})$ *are independent.*

*Proof.*  As a preliminary, please refer to Remarks E.9 and F.1 and Lemma E.10.

Note that effects of all the random variables in the layer before the $l$-th, such as $\boldsymbol{P}^{(l-1)}$, $\boldsymbol{h}^{(l-1)}$, $\boldsymbol{W}^{(l-1)}$, and $\boldsymbol{b}^{(l-1)}$, are aggregated to $\boldsymbol{x}^{(l-1)}$. By Lemma E.8, $\boldsymbol{U}^{(l)}$ and $\boldsymbol{d}^{(l)}$ are independent of $\boldsymbol{U}^{(l')}$ and $\boldsymbol{d}^{(l')}$ for any $l' \neq l$. Similarly, as $\boldsymbol{V}^{(l)}$ and $\boldsymbol{c}^{(l)}$ consist of $\boldsymbol{U}^{(1)}, \ldots, \boldsymbol{U}^{(l)}, \boldsymbol{d}^{(1)}, \ldots, \boldsymbol{d}^{(l)}, \boldsymbol{U}^{(l+1)}$ and $\boldsymbol{d}^{(l+1)}$ are independent of $\boldsymbol{V}^{(l)}$ and $\boldsymbol{c}^{(l)}$. In addition, similar to Lemma E.10, $\boldsymbol{U}^{(l)}$ and $\boldsymbol{d}^{(l)}$ are independent by Lemma E.7.

We prove the claim by induction. Assume that $\boldsymbol{V}^{(l-1)}$ is a Gaussian matrix. This holds for $l' = 1$ because of $\boldsymbol{V}^{(1)} = \boldsymbol{U}^{(1)}$. Similar to Lemma E.8, $U_{ij}$ is independent of $U_i.V_{.j}$. Since $\boldsymbol{U}^{(l)}$ and $\boldsymbol{V}^{(l-1)}$ are independent Gaussian matrices and $N$ is sufficiently large, $\boldsymbol{V}^{(l)}$ is a Gaussian matrices. Similarly, $\boldsymbol{c}^{(l)}$ is a Gaussian vector. By Lemma E.7, $\boldsymbol{V}^{(l)}$ and $\boldsymbol{c}^{(l)}$ are independent. Thus, similar to Lemma E.10, $\boldsymbol{J}(\boldsymbol{x}^{\mathrm{in}})$ and $\boldsymbol{a}(\boldsymbol{x}^{\mathrm{in}})$ are independent Gaussian matrices. $\qquad\square$

## G  Derivation of the theorems in Sec. 5

### G.1  Derivation of the theorems in Sec. 5.1

First, we introduce the following lemma, which is required to derive the maximum $\ell_\infty$ norm of a column of $\boldsymbol{J}(\boldsymbol{x}^{\mathrm{in}})$.

**Lemma G.1.** *The maximum absolute value of $n \in \mathbb{N}$ i.i.d. Gaussian variables with zero mean and variance $\sigma^2 > 0$ is smaller than $\sqrt{2\sigma^2 \ln n}$.*

Table A3: Computable combination of a $(p, q)$-operator norm [91].

| | $q = 1$ | $q = 2$ | $q = \infty$ |
|---|---|---|---|
| $p = 1$ | max. $\ell_1$ norm of a column | max. $\ell_2$ norm of a column | max. $\ell_\infty$ norm of a column |
| $p = 2$ | NP-hard | max. singular value | max. $\ell_2$ norm of a row |
| $p = \infty$ | NP-hard | NP-hard | max. $\ell_1$ norm of a row |

*Proof.* Let $z \in \mathbb{R}$ be a Gaussian variable with zero mean and variance $\sigma^2$ and $a > 0$ be a positive value. First, we compute the upper bound of probability of $z > a$ as follows:

$$\mathbb{P}[z > a] = \int_a^\infty g(z; \sigma^2) dz \tag{A74}$$

$$\leq \int_a^\infty \frac{z}{a} g(z; \sigma^2) dz \tag{A75}$$

$$= \frac{1}{a\sqrt{2\pi\sigma^2}} \left[ -\sigma^2 \exp\left(-\frac{z^2}{2\sigma^2}\right) \right]_a^\infty \tag{A76}$$

$$= \frac{1}{a}\sqrt{\frac{\sigma^2}{2\pi}} \exp\left(-\frac{a^2}{2\sigma^2}\right). \tag{A77}$$

Then, we compute the lower bound of probability of $|z| < a$ as follows:

$$\mathbb{P}[|z| < a] = \int_{-a}^a g(z; \sigma^2) dz \tag{A78}$$

$$= 2\int_0^a g(z; \sigma^2) dz + 2\int_{-\infty}^0 g(z; \sigma^2) dz - 1 \tag{A79}$$

$$= 2\int_{-\infty}^a g(z; \sigma^2) dz - 1 \tag{A80}$$

$$= 2\left(1 - \int_a^\infty g(z; \sigma^2) dz\right) - 1 \tag{A81}$$

$$= 1 - 2\int_a^\infty g(z; \sigma^2) dz \tag{A82}$$

$$\geq 1 - \frac{1}{a}\sqrt{\frac{2\sigma^2}{\pi}} \exp\left(-\frac{a^2}{2\sigma^2}\right). \tag{A83}$$

Let $\{z_i\}_{i=1}^n$ be $n$ i.i.d. Gaussian variables. Finally, we compute the upper bound of probability of $\max_{i \in [n]} |z_i| > a$ as follows:

$$\mathbb{P}\left[\max_{i \in [n]} |z_i| > a\right] = (1 - \mathbb{P}[|z| < a])^n \leq \left(\frac{1}{a}\sqrt{\frac{2\sigma^2}{\pi}} \exp\left(-\frac{a^2}{2\sigma^2}\right)\right)^n. \tag{A84}$$

In particular, when $a = \sqrt{2\sigma^2 \ln n}$,

$$\mathbb{P}\left[\max_{i \in [n]} |z_i| > \sqrt{2\sigma^2 \ln n}\right] \leq \left(\frac{1}{n\sqrt{\pi \ln n}}\right)^n \xrightarrow{n \to \infty} 0. \tag{A85}$$

Thus, the claim is established. $\qquad\square$

**Theorem 5.1** (Upper bounds of adversarial loss). *Suppose that the input dimension $d$, output dimension $K$, and width $N$ are sufficiently large. Then, for any $\boldsymbol{x}^{\text{in}} \in \mathbb{R}^d$, the following inequality holds:*

$$\mathcal{L}_{\text{adv}}(\boldsymbol{x}^{\text{in}}) \leq \epsilon\beta_{p,q}\omega^{L/2} = \begin{cases} \epsilon\beta_{p,q}(\frac{\alpha}{LN}\sum_{W \in \mathcal{W}} W^2)^{L/2} & \text{(vanilla)} \\ \epsilon\beta_{p,q}(1 + \frac{\alpha}{LN}\sum_{W \in \mathcal{W}} W^2)^{L/2} & \text{(residual)} \end{cases}, \tag{9}$$

*where $\mathcal{W} := \{W_{1,1}^{(1)}, W_{1,2}^{(1)}, \dots, W_{N,N}^{(L)}\}$ denotes the set of all network weights. The constant $\beta_{p,q}$ for each norm pair $(p, q)$ is described in Tab. 1.*

*Proof.* We note the following:

- By Thm 4.1, each entry of $\boldsymbol{J}(\boldsymbol{x}^{\mathrm{in}})$ is i.i.d. and follows a Gaussian with zero mean and variance $\mathbb{V}[J(\boldsymbol{x}^{\mathrm{in}})_{ij}] = \omega^L/d$.
- The input and output dimensions, $d$ and $K$, are sufficiently large (cf. Sec. 3.1).
- For some combinations of $(p, q)$, $(p, q)$-operator norms are computable (cf. Tab. A3) [91].

In addition, we note the following upper bound (cf. Sec. 5.1):

$$\mathcal{L}_{\mathrm{adv}}(\boldsymbol{x}^{\mathrm{in}}) \leq \epsilon \max_{\boldsymbol{x} \in \mathbb{R}^d} \|\boldsymbol{J}(\boldsymbol{x})\|_{p,q}. \tag{Ineq. (8)}$$

As a preliminary, we compute the mean absolute value of $J(\boldsymbol{x}^{\mathrm{in}})_{ij}$ as follows:

$$\mathbb{E}[|J(\boldsymbol{x}^{\mathrm{in}})_{ij}|] = \sqrt{\frac{2\omega^L}{\pi d}}. \tag{A86}$$

In the above derivation, we use the following equation

$$\mathbb{E}_{z \sim \mathcal{N}(0,\sigma^2)}[|z|] = \sqrt{\frac{2\sigma^2}{\pi}}. \tag{A87}$$

Based on Tab. A3, we compute $\|\boldsymbol{J}(\boldsymbol{x}^{\mathrm{in}})\|_{p,q}$ as follows:

**Maximum $\ell_1$ norm of a column, $\|\boldsymbol{J}(\boldsymbol{x}^{\mathrm{in}})\|_{1,1}$.**

$$\max_{j \in [d]} \sum_{i=1}^{K} |J(\boldsymbol{x}^{\mathrm{in}})_{ij}| = \max_{j \in [d]} K \frac{1}{K} \sum_{i=1}^{K} |J(\boldsymbol{x}^{\mathrm{in}})_{ij}| \tag{A88}$$

$$= \max_{j \in [d]} K \mathbb{E}[|J(\boldsymbol{x}^{\mathrm{in}})_{ij}|] \tag{A89}$$

$$= \max_{j \in [d]} K \sqrt{\frac{2\omega^L}{\pi d}} \tag{A90}$$

$$= \sqrt{\frac{2}{\pi d}} K \omega^{L/2}. \tag{A91}$$

**Maximum $\ell_2$ norm of a column, $\|\boldsymbol{J}(\boldsymbol{x}^{\mathrm{in}})\|_{1,2}$.**

$$\max_{j \in [d]} \sqrt{\sum_{i=1}^{K} J(\boldsymbol{x}^{\mathrm{in}})_{ij}^2} = \max_{j \in [d]} \sqrt{K \frac{1}{K} \sum_{i=1}^{K} J(\boldsymbol{x}^{\mathrm{in}})_{ij}^2} \tag{A92}$$

$$= \max_{j \in [d]} \sqrt{K \mathbb{E}[J(\boldsymbol{x}^{\mathrm{in}})_{ij}^2]} \tag{A93}$$

$$= \max_{j \in [d]} \sqrt{K \frac{\omega^L}{d}} \tag{A94}$$

$$= \sqrt{\frac{K}{d}} \omega^{L/2}. \tag{A95}$$

**Maximum $\ell_\infty$ norm of a column, $\|\boldsymbol{J}(\boldsymbol{x}^{\mathrm{in}})\|_{1,\infty}$.** By Lemma G.1,

$$\max_{i \in [K]} |J(\boldsymbol{x}^{\mathrm{in}})_{ij}| \leq \sqrt{2 \mathbb{V}[J(\boldsymbol{x}^{\mathrm{in}})_{ij}] \ln K} = \sqrt{\frac{2 \ln K}{d}} \omega^{L/2}. \tag{A96}$$

**Maximum singular value, $\|\boldsymbol{J}(\boldsymbol{x}^{\mathrm{in}})\|_{2,2}$.** By the Marchenko—Pastur law,

$$\|\boldsymbol{J}(\boldsymbol{x}^{\mathrm{in}})\|_2 \leq \left(1 + \sqrt{\frac{K}{d}}\right) \omega^{L/2}, \tag{A97}$$

where $\|\cdot\|_2$ denotes the spectral norm (largest singular value).

**Maximum $\ell_2$ norm of a row, $\|\boldsymbol{J}(\boldsymbol{x}^{\mathrm{in}})\|_{2,\infty}$.**

$$\max_{i\in[K]} \sqrt{\sum_{j=1}^{d} J(\boldsymbol{x}^{\mathrm{in}})_{ij}^2} = \max_{i\in[K]} \sqrt{d\frac{1}{d}\sum_{j=1}^{d} J(\boldsymbol{x}^{\mathrm{in}})_{ij}^2} \tag{A98}$$

$$= \max_{i\in[K]} \sqrt{d\mathbb{E}[J(\boldsymbol{x}^{\mathrm{in}})_{ij}^2]} \tag{A99}$$

$$= \max_{i\in[K]} \sqrt{d\frac{\omega^L}{d}} \tag{A100}$$

$$= \omega^{L/2}. \tag{A101}$$

**Maximum $\ell_1$ norm of a row, $\|\boldsymbol{J}(\boldsymbol{x}^{\mathrm{in}})\|_{\infty,\infty}$.**

$$\max_{i\in[K]} \sum_{j=1}^{d} |J(\boldsymbol{x}^{\mathrm{in}})_{ij}| = \max_{i\in[K]} d\frac{1}{d}\sum_{j=1}^{d} |J(\boldsymbol{x}^{\mathrm{in}})_{ij}| \tag{A102}$$

$$= \max_{i\in[K]} d\mathbb{E}[|J(\boldsymbol{x}^{\mathrm{in}})_{ij}|] \tag{A103}$$

$$= \max_{i\in[K]} d\sqrt{\frac{2\omega^L}{\pi d}} \tag{A104}$$

$$= \sqrt{\frac{2d}{\pi}}\omega^{L/2}. \tag{A105}$$

$\square$

In addition, we try to derive equalities rather than inequalities (upper bounds). First, we note the following equation:

$$\mathcal{L}_{\mathrm{adv}}(\boldsymbol{x}^{\mathrm{in}}) := \max_{\|\boldsymbol{\eta}\|_p \le \epsilon} \left\| \boldsymbol{f}(\boldsymbol{x}^{\mathrm{in}} + \boldsymbol{\eta}) - \boldsymbol{f}(\boldsymbol{x}^{\mathrm{in}}) \right\|_q \tag{Eq. (3)}$$

$$= \max_{\|\boldsymbol{\eta}\|_p \le \epsilon} \left\| \boldsymbol{J}(\boldsymbol{x}^{\mathrm{in}} + \boldsymbol{\eta})(\boldsymbol{x}^{\mathrm{in}} + \boldsymbol{\eta}) + \boldsymbol{a}(\boldsymbol{x}^{\mathrm{in}} + \boldsymbol{\eta}) - \boldsymbol{J}(\boldsymbol{x}^{\mathrm{in}})\boldsymbol{x}^{\mathrm{in}} - \boldsymbol{a}(\boldsymbol{x}^{\mathrm{in}}) \right\|. \tag{A106}$$

Because $\boldsymbol{J}(\boldsymbol{x}^{\mathrm{in}})$ and $\boldsymbol{a}(\boldsymbol{x}^{\mathrm{in}})$ are the slope and bias of a piecewise linear region, respectively, for sufficiently small $\boldsymbol{\eta}$, $\boldsymbol{J}(\boldsymbol{x}^{\mathrm{in}} + \boldsymbol{\eta})$ and $\boldsymbol{a}(\boldsymbol{x}^{\mathrm{in}} + \boldsymbol{\eta})$ are identical to $\boldsymbol{J}(\boldsymbol{x}^{\mathrm{in}})$ and $\boldsymbol{a}(\boldsymbol{x}^{\mathrm{in}})$, respectively. Therefore, for sufficiently small $\boldsymbol{\eta}$, we can derive the following equation:

$$\mathcal{L}_{\mathrm{adv}}(\boldsymbol{x}^{\mathrm{in}}) = \max_{\|\boldsymbol{\eta}\|_p \le \epsilon} \left\| \boldsymbol{J}(\boldsymbol{x}^{\mathrm{in}})\boldsymbol{\eta} \right\| = \epsilon \left\| \boldsymbol{J}(\boldsymbol{x}^{\mathrm{in}}) \right\|_{p,q}. \tag{A107}$$

**Proposition G.2.** *Suppose that the perturbation constraint $\epsilon$ is sufficiently small. For any $\boldsymbol{x}^{\mathrm{in}} \in \mathbb{R}^d$ and $(p, q) = (1, 1)$, $(1, 2)$, $(2, \infty)$, and $(\infty, \infty)$, the following equality holds:*

$$\mathcal{L}_{\mathrm{adv}}(\boldsymbol{x}^{\mathrm{in}}) = \epsilon\beta_{p,q}\omega^{L/2}. \tag{A108}$$

*Proof.* As shown in Eq. (A107), the adversarial loss (Eq. (3)) is equal to the $(p, q)$-operator norm of $\boldsymbol{J}(\boldsymbol{x}^{\mathrm{in}})$ under this assumption. For $(p, q) = (1, 1)$, $(1, 2)$, $(2, \infty)$, and $(\infty, \infty)$, we can obtain the equalities of the $(p, q)$-operator norms (cf. the proof of Thm 5.1). Thus, the claim is established. Note that, for $(p, q) = (1, \infty)$ and $(2, 2)$, we can derive only upper bounds (cf. the proof of Thm 5.1). $\square$

**Proposition G.3.** *Suppose that the perturbation constraint $\epsilon$ is sufficiently small and the output dimension is one. For any $\boldsymbol{x}^{\mathrm{in}} \in \mathbb{R}^d$, and $(p, q) = (2, \infty)$ and $(\infty, \infty)$, the following equality holds:*

$$\mathcal{L}_{\mathrm{adv}}(\boldsymbol{x}^{\mathrm{in}}) = \epsilon\beta_{p,q}\omega^{L/2}. \tag{A109}$$

*In addition, for any $\boldsymbol{x}^{\mathrm{in}} \in \mathbb{R}^d$ and $(p, q) = (2, 2)$, the following equality holds:*

$$\mathcal{L}_{\mathrm{adv}}(\boldsymbol{x}^{\mathrm{in}}) = \epsilon\omega^{L/2}. \tag{A110}$$

*Proof.* Similar to Prop G.2, the upper bounds for $(p, q) = (2, \infty)$ and $(\infty, \infty)$ become equalities. The upper bounds for $(p, q) = (1, 1)$ and $(1, 2)$ require sufficiently large $K$, and thus, they do not become equalities. In addition, as Tab. A3 shows, the $(2, 2)$-operator norm is a maximum singular value. When $K = 1$, the maximum singular value is identical to the Frobenius norm ($\ell_2$ norm), which is calculated as follows:

$$\left\| \boldsymbol{J}(\boldsymbol{x}^{\text{in}}) \right\|_{\text{F}} = \sqrt{\sum_{j=1}^{d} J(\boldsymbol{x}^{\text{in}})_{1j}^2} \tag{A111}$$

$$= \sqrt{d \frac{1}{d} \sum_{j=1}^{d} J(\boldsymbol{x}^{\text{in}})_{1j}^2} \tag{A112}$$

$$= \sqrt{d \mathbb{E}[J(\boldsymbol{x}^{\text{in}})_{1j}^2]} \tag{A113}$$

$$= \omega^{L/2}. \tag{A114}$$

Thus, in contrast to the proof of Thm 5.1, we can obtain the equality of the $(2, 2)$-operator norm instead of the inequality (upper bound). $\qquad \square$

## G.2 Derivation of the theorems in Sec. 5.2

In the following, we set $\mathcal{L}_{\text{adv}} := \epsilon \beta_{p,q} \omega^{L/2}$. As a preliminary, we state the following two lemmas. These are used to solve the differential equation derived from gradient flow.

**Lemma G.4.** *For $t, a, b \in \mathbb{R}$, $x : \mathbb{R} \to \mathbb{R}$, and $x(0) := x_0$, the following holds:*

$$\frac{\mathrm{d}x(t)}{\mathrm{d}t} = -ax(t)^b \Leftrightarrow x(t) = (a(b-1)t + x_0^{-(b-1)})^{-1/(b-1)}. \tag{A115}$$

*Proof.*

$$\frac{\mathrm{d}x(t)}{\mathrm{d}t} = -ax(t)^b \tag{A116}$$

$$\int x^{-b} \mathrm{d}x = -a \int \mathrm{d}t \tag{A117}$$

$$\frac{1}{-(b-1)} x^{-(b-1)} = -at + C_1 \tag{A118}$$

$$x(t) = (a(b-1)t + C_2)^{-1/(b-1)}. \tag{A119}$$

By the initial value,

$$x(0) := x_0 = C_2^{-1/(b-1)} \tag{A120}$$

$$C_2 = x_0^{-(b-1)}. \tag{A121}$$

Thus,

$$x(t) = (a(b-1)t + x_0^{-(b-1)})^{-1/(b-1)}. \tag{A122}$$

$\qquad \square$

**Lemma G.5.** *For $t, c \in \mathbb{R}$, $a, b \geq 0$, $x : \mathbb{R} \to \mathbb{R}$, $x(0) := x_0$, and $0 \leq bx(0) \ll 1$, the following holds:*

$$\frac{\mathrm{d}x(t)}{\mathrm{d}t} = -a(1 + bx(t))^c x(t) \Leftrightarrow x(t) = \frac{x_0}{(1 + bcx_0) \exp(at) - bcx_0}. \tag{A123}$$

*Proof.* By $-a(1 + bx(t))^c x(t) \leq 0$, $x(t)$ does not increase with $t$. That is, $bx(t) \leq bx(0) \ll 1$. Therefore, we can approximate $(1 + bx(t))^c$ as $1 + bcx(t)$ using the binomial theorem. Thus, we try to solve the following differential equation:

$$\frac{\mathrm{d}x(t)}{\mathrm{d}t} = -a(1 + bcx(t))x(t), \tag{A124}$$

whose solution is as follows:

$$x(t) = \frac{C_1}{bc(\exp(at) - C_1)}. \tag{A125}$$

By the initial value,

$$x(0) := x_0 = \frac{C_1}{bc(1 - C_1)} \tag{A126}$$

$$bc(1 - C_1)x_0 = C_1 \tag{A127}$$

$$bcx_0 = (1 + bcx_0)C_1 \tag{A128}$$

$$C_1 = \frac{bcx_0}{1 + bcx_0}. \tag{A129}$$

Thus,

$$x(t) = \frac{\frac{bcx_0}{1+bcx_0}}{bc(\exp(at) - \frac{bcx_0}{1+bcx_0})} \tag{A130}$$

$$= \frac{bcx_0}{bc((1 + bcx_0)\exp(at) - bcx_0)} \tag{A131}$$

$$= \frac{x_0}{(1 + bcx_0)\exp(at) - bcx_0}. \tag{A132}$$

$\square$

Then, using the update equation of parameters (Eq. (10)), we consider the differential equation of weight variance as follows:

**Lemma G.6.** *Suppose that Asm 5.2 holds. Let $\mathcal{L}_1, \ldots, \mathcal{L}_n : \mathbb{R} \to \mathbb{R}$ be the $n$ loss functions and $\mathcal{W} := \{W_{11}^{(1)}, W_{12}^{(1)}, \ldots, W_{NN}^{(L)}\}$ be the set of all network weights. A network is trained by minimizing the sum of loss functions, $\sum_{i=1}^{n} \mathcal{L}_i(\boldsymbol{x}^{\mathrm{in}})$. The network parameters are updated similarly to Eq. (10). The differential equation of $\sigma_w^2(t)$ is given by:*

$$\frac{\mathrm{d}\sigma_w^2(t)}{\mathrm{d}t} = -N\mathbb{E}_{W \in \mathcal{W}}\left[W(t)\sum_{i=1}^{n}\frac{\partial\mathcal{L}_i(\boldsymbol{x}^{\mathrm{in}})}{\partial W(t)}\right] \tag{A133}$$

*Proof.* Since Asm 5.2 holds and the number of weights, $LN^2$, is sufficiently large, $\sigma_w^2(t + \mathrm{d}t)$ can be represented as follows:

$$\sigma_w^2(t + \mathrm{d}t) = N\frac{\sigma_w^2(t + \mathrm{d}t)}{N} = N\mathbb{V}_{W \in \mathcal{W}}[W(t + \mathrm{d}t)] = N\mathbb{E}_{W \in \mathcal{W}}[W(t + \mathrm{d}t)^2]. \tag{A134}$$

We can expand $W(t + \mathrm{d}t)$ using Eq. (10) and rearrange the above equation as follows:

$$\sigma_w^2(t + \mathrm{d}t) = N\mathbb{E}_{W \in \mathcal{W}}\left[\left(W(t) - \sum_{i=1}^{n}\frac{\partial\mathcal{L}_i(\boldsymbol{x}^{\mathrm{in}})}{\partial W(t)}\mathrm{d}t\right)^2\right] \tag{A135}$$

$$= N\mathbb{E}_{W \in \mathcal{W}}\left[W(t)^2 - W(t)\sum_{i=1}^{n}\frac{\partial\mathcal{L}_i(\boldsymbol{x}^{\mathrm{in}})}{\partial W(t)}\mathrm{d}t + \mathcal{O}(\mathrm{d}t^2)\right] \tag{A136}$$

$$= \sigma_w^2(t) - N\mathbb{E}_{W \in \mathcal{W}}\left[W(t)\sum_{i=1}^{n}\frac{\partial\mathcal{L}_i(\boldsymbol{x}^{\mathrm{in}})}{\partial W(t)}\mathrm{d}t\right] + \mathcal{O}(\mathrm{d}t^2) \tag{A137}$$

$$\approx \sigma_w^2(t) - N\mathbb{E}_{W \in \mathcal{W}}\left[W(t)\sum_{i=1}^{n}\frac{\partial\mathcal{L}_i(\boldsymbol{x}^{\mathrm{in}})}{\partial W(t)}\right]\mathrm{d}t. \tag{A138}$$

Thus,

$$\frac{\mathrm{d}\sigma_w^2(t)}{\mathrm{d}t} = -N\mathbb{E}_{W \in \mathcal{W}}\left[W(t)\sum_{i=1}^{n}\frac{\partial\mathcal{L}_i(\boldsymbol{x}^{\mathrm{in}})}{\partial W(t)}\right]. \tag{A139}$$

$\square$

**Lemma G.7.** *For $\mathcal{L}_{\mathrm{adv}} := \epsilon\beta_{p,q}\omega^{L/2}$, the following equality holds:*

$$\frac{\partial\mathcal{L}_{\mathrm{adv}}}{\partial W} = \frac{\epsilon\alpha\beta_{p,q}\omega^{L/2-1}}{N}W. \tag{A140}$$

*Proof.* If the number of network weights, i.e., $LN^2$, is sufficiently large, we can represent the weight variance as follows:

$$\frac{\sigma_w^2}{N} = \frac{1}{LN^2}\sum_{W\in\mathcal{W}}W^2. \tag{A141}$$

The derivative of $\omega$ with respect to $W \in \mathcal{W}$ can be calculated as follows:

$$\frac{\partial\omega_{\mathrm{v}}}{\partial W} = \frac{\partial(\alpha\sigma_w^2)}{\partial W} = \frac{\partial(\alpha N\frac{\sigma_w^2}{N})}{\partial W} = \alpha N\frac{\partial\left(\frac{1}{LN^2}\sum_{V\in\mathcal{W}}V^2\right)}{\partial W} = \frac{2\alpha}{LN}W, \tag{A142}$$

$$\frac{\partial\omega_{\mathrm{r}}}{\partial W} = \frac{\partial(1+\alpha\sigma_w^2)}{\partial W} = \frac{2\alpha}{LN}W. \tag{A143}$$

The derivative of $\omega^{L/2}$ with respect to $W \in \mathcal{W}$ can be calculated as follows:

$$\frac{\partial\omega^{L/2}}{\partial W} = \frac{L}{2}\omega^{L/2-1}\frac{\partial\omega}{\partial W} = \frac{L}{2}\omega^{L/2-1}\frac{2\alpha}{LN}W = \frac{\alpha\omega^{L/2-1}}{N}W. \tag{A144}$$

Thus,

$$\frac{\partial\mathcal{L}_{\mathrm{adv}}}{\partial W} = \epsilon\beta_{p,q}\frac{\partial\omega^{L/2}}{\partial W} = \frac{\epsilon\alpha\beta_{p,q}\omega^{L/2-1}}{N}W. \tag{A145}$$

$\square$

**Lemma G.8.** *Suppose that Asm 5.2 holds. Let $\mathcal{L}_{\mathrm{std}} : \mathbb{R}^d \to \mathbb{R}$ be the standard loss function and $\mathcal{L}_{\mathrm{adv}} : \mathbb{R}^d \to \mathbb{R}$ be the adversarial loss function. Suppose that Asm B.1 applies to $\mathcal{L}_{\mathrm{std}}$. The adversarial loss function is defined as $\mathcal{L}_{\mathrm{adv}}(\boldsymbol{x}^{\mathrm{in}};t) := \epsilon\beta_{p,q}\omega(t)^{L/2}$. A network is trained by minimizing $\mathcal{L}_{\mathrm{std}} + \mathcal{L}_{\mathrm{adv}}$. Network parameters are updated by Eq. (10). Then, the differential equation of $\sigma_w^2(t)$ is given by:*

$$\frac{\mathrm{d}\sigma_w^2(t)}{\mathrm{d}t} = -\frac{\epsilon\alpha\beta_{p,q}}{N}\omega(t)^{L/2-1}\sigma_w^2(t). \tag{A146}$$

*Proof.* By Lemma G.6,

$$\frac{\mathrm{d}\sigma_w^2(t)}{\mathrm{d}t} = -N\mathbb{E}\left[W(t)\frac{\partial\mathcal{L}_{\mathrm{std}}}{\partial W(t)}\right] - N\mathbb{E}\left[W(t)\frac{\partial\mathcal{L}_{\mathrm{adv}}}{\partial W(t)}\right]. \tag{A147}$$

Since the standard loss satisfies Asm B.1, $\mathbb{E}\left[W(t)\frac{\partial\mathcal{L}_{\mathrm{std}}}{\partial W(t)}\right] = 0$ and

$$\frac{\mathrm{d}\sigma_w^2(t)}{\mathrm{d}t} = -N\mathbb{E}\left[W(t)\frac{\partial\mathcal{L}_{\mathrm{adv}}}{\partial W(t)}\right]. \tag{A148}$$

By Lemma G.7,

$$\frac{\mathrm{d}\sigma_w^2(t)}{\mathrm{d}t} = -N\mathbb{E}\left[W(t)\frac{\epsilon\alpha\beta_{p,q}\omega(t)^{L/2-1}}{N}W(t)\right] \tag{A149}$$

$$= -\epsilon\alpha\beta_{p,q}\omega(t)^{L/2-1}\mathbb{E}[W(t)^2] \tag{A150}$$

$$= -\frac{\epsilon\alpha\beta_{p,q}}{N}\omega(t)^{L/2-1}\sigma_w^2(t). \tag{A151}$$

$\square$

**Theorem 5.4** (Weight time evolution of vanilla network in adversarial training). *Suppose that Asms 5.2 and 5.3 hold. Then, the time evolution of $\sigma_w^2$ of a vanilla network in adversarial training is given by:*

$$\sigma_w^2(t) = \left(1 - \frac{\epsilon\alpha\beta_{p,q}\omega_{\mathrm{v}}(0)^{L/2-1}}{N}t\right)\sigma_w^2(0). \tag{11}$$

*Proof.* By Lemma G.8, the time evolution of weight variance in a vanilla network is represented as follows:

$$\frac{\mathrm{d}\sigma_w^2(t)}{\mathrm{d}t} = -\frac{\epsilon\alpha\beta_{p,q}}{N}\omega_v(t)^{L/2-1}\sigma_w^2(t) \tag{A152}$$

$$= -\frac{\epsilon\alpha\beta_{p,q}}{N}\alpha^{L/2-1}\sigma_w^2(t)^{L/2-1}\sigma_w^2(t) \tag{A153}$$

$$= -\frac{\epsilon\alpha^{L/2}\beta_{p,q}}{N}\sigma_w^2(t)^{L/2}. \tag{A154}$$

Denote $L' := L/2 - 1$. By Lemma G.4, we can obtain

$$\sigma_w^2(t) = \left(\frac{\epsilon\alpha^{L/2}\beta_{p,q}}{N}(L/2-1)t + \sigma_w^2(0)^{-(L/2-1)}\right)^{-1/(L/2-1)} \tag{A155}$$

$$= \left(\frac{\epsilon\alpha^{L'+1}\beta_{p,q}L'}{N}t + \sigma_w^2(0)^{-L'}\right)^{-1/L'} \tag{A156}$$

$$= \left(1 + \frac{\epsilon\alpha^{L'+1}\sigma_w^2(0)^{L'}\beta_{p,q}L'}{N}t\right)^{-1/L'}\sigma_w^2(0) \tag{A157}$$

$$= \left(1 + \frac{\epsilon\alpha\omega_\mathrm{v}(0)^{L'}\beta_{p,q}L'}{N}t\right)^{-1/L'}\sigma_w^2(0). \tag{A158}$$

By $t \leq T \ll N$,

$$\sigma_w^2(t) \approx \left(1 - \frac{1}{L'}\frac{\epsilon\alpha\beta_{p,q}\omega_\mathrm{v}(0)^{L'}L'}{N}t\right)\sigma_w^2(0) = \left(1 - \frac{\epsilon\alpha\beta_{p,q}\omega_\mathrm{v}(0)^{L'}}{N}t\right)\sigma_w^2(0). \tag{A159}$$

$\square$

**Theorem G.9** (Weight time evolution of a residual network in adversarial training)**.** *Suppose that Asm 5.2 holds and $\alpha\sigma_w^2(0) \ll 1$. The time evolution of $\sigma_w^2$ of a residual network in adversarial training is given by:*

$$\sigma_w^2(t) = \left(1 - (1 + \alpha L'\sigma_w^2(0))\epsilon\alpha\beta_{p,q}t/N\right)\sigma_w^2(0). \tag{A160}$$

*Proof.* By Lemma G.8, the time evolution of weight variance in a residual network is represented as follows:

$$\frac{\mathrm{d}\sigma_w^2(t)}{\mathrm{d}t} = -\frac{\epsilon\alpha\beta_{p,q}}{N}\omega(t)^{L/2-1}\sigma_w^2(t) = -\frac{\epsilon\alpha\beta_{p,q}}{N}(1 + \alpha\sigma_w^2(t))^{L/2-1}\sigma_w^2(t). \tag{A161}$$

By Lemma G.5, we can obtain

$$\sigma_w^2(t) = \frac{\sigma_w^2(0)}{(1 + \alpha(L/2-1)\sigma_w^2(0))\exp\left(\frac{\epsilon\alpha\beta_{p,q}}{N}t\right) - \alpha(L/2-1)\sigma_w^2(0)} \tag{A162}$$

$$= ((\sigma_w^2(0)^{-1} + \alpha L')\exp(\epsilon\alpha\beta_{p,q}t/N) - \alpha L')^{-1}. \tag{A163}$$

By the Maclaurin expansion of the exponential function and $t \leq T \ll N$,

$$\sigma_w^2(t) \approx \left((\sigma_w^2(0)^{-1} + \alpha L')\left(1 + \frac{\epsilon\alpha\beta_{p,q}t}{N}\right) - \alpha L'\right)^{-1} \tag{A164}$$

$$= \left(\sigma_w^2(0)^{-1} + (\sigma_w^2(0)^{-1} + \alpha L')\frac{\epsilon\alpha\beta_{p,q}t}{N}\right)^{-1} \tag{A165}$$

$$= \left(1 + (1 + \alpha L'\sigma_w^2(0))\frac{\epsilon\alpha\beta_{p,q}t}{N}\right)^{-1}\sigma_w^2(0). \tag{A166}$$

By $t \leq T \ll N$,

$$\sigma_w^2(t) = \left(1 - (1 + \alpha L'\sigma_w^2(0))\frac{\epsilon\alpha\beta_{p,q}t}{N}\right)\sigma_w^2(0). \tag{A167}$$

$\square$

Then, we consider the time evolution of weight variance, Thms 5.4 and G.9, at initialization satisfying $(M, m)$-trainability condition (Lemma 5.6). Here, we assume $\omega(0) \approx 1$ satisfying Lemma 5.6. Under this assumption, the time evolution of weight variance in vanilla networks, Eq. (11), can be rearranged as follows:

$$\sigma_w^2(t) = (1 - \epsilon\alpha\beta_{p,q}\omega_{\mathrm{v}}(0)^{L/2-1}t/N)\sigma_w^2(0) \xrightarrow{\omega_{\mathrm{v}}(0)\approx 1} (1 - \epsilon\alpha\beta_{p,q}t/N)\sigma_w^2(0). \qquad (A168)$$

In addition, the time evolution of weight variance in residual networks, Eq. (A160), can be rearranged as follows:

$$\sigma_w^2(t) = \left(1 - (1 + \alpha L'\sigma_w^2(0))\epsilon\alpha\beta_{p,q}t/N\right)\sigma_w^2(0) \xrightarrow{\omega_{\mathrm{r}}(0)\approx 1} (1 - \epsilon\alpha\beta_{p,q}t/N)\sigma_w^2(0). \qquad (A169)$$

Thus, the time evolution of weight variance is consistent in vanilla and residual networks at initialization satisfying Lemma 5.6.

## G.3  Derivation of the theorems in Sec. 5.3

**Lemma 5.6** (Vanilla and residual $(M, m)$-trainability condition). *Suppose that the width $N$ is sufficiently large. Then, the $(M, m)$-trainability conditions for vanilla and residual networks are respectively given by:*

$$m^{1/L} \leq \alpha\sigma_w^2 \leq M^{1/L} \text{ (vanilla)}, \qquad \alpha\sigma_w^2 \leq M^{1/L} - 1 \text{ (residual)}. \qquad (12)$$

*Proof.* Applying Eq. (A38) to Eqs. (2) and (A19), we can obtain the following equations:

$$\frac{\chi^{(l)}}{\chi^{(l+1)}} = \begin{cases} \sigma_w^2\mathbb{E}[\phi'(h^{(l+1)})] = \alpha\sigma_w^2 = \omega_{\mathrm{v}} & \text{(vanilla)} \\ 1 + \sigma_w^2\mathbb{E}[\phi'(h^{(l+1)})^2] = 1 + \alpha\sigma_w^2 = \omega_{\mathrm{r}} & \text{(residual)} \end{cases}. \qquad (A170)$$

Then, recursively computing the above equation, we can derive the following equations:

$$\frac{\chi^{(0)}}{\chi^{(L)}} = \omega^L = \begin{cases} (\alpha\sigma_w^2)^L & \text{(vanilla)} \\ (1 + \alpha\sigma_w^2)^L & \text{(residual)} \end{cases}. \qquad (A171)$$

Applying this equation to Defn 5.5,

$$m \leq \frac{\chi^{(0)}}{\chi^{(L)}} \leq M \Leftrightarrow m^{1/L} \leq \omega \leq M^{1/L}. \qquad (A172)$$

For a vanilla network,

$$m^{1/L} \leq \alpha\sigma_w^2 \leq M^{1/L}. \qquad (A173)$$

For a residual network,

$$m^{1/L} - 1 \leq \alpha\sigma_w^2 \leq M^{1/L} - 1. \qquad (A174)$$

In particular, for a residual network, by $m^{1/L} - 1 \leq 0$ and $\alpha\sigma_w^2 \geq 0$,

$$(0 \leq)\alpha\sigma_w^2 \leq M^{1/L} - 1. \qquad (A175)$$

$\square$

**Theorem 5.7** (Vanilla networks are not adversarially trainable). *Consider a vanilla network. Suppose that Asms 5.2 and 5.3 hold, and the $(M, m)$-trainability condition holds at $t = 0$ and $\alpha\sigma_w^2(0) = 1$. If*

$$T \geq \frac{(1 - m^{1/L})N}{\epsilon\alpha\beta_{p,q}}, \qquad (13)$$

*then there exists $0 < \tau \leq T$ such that the $(M, m)$-trainability condition does not hold for $\tau \leq t \leq T$.*

*Proof.* Let us consider $T$ breaking the $(M, m)$-trainability condition. As Thm 5.4 claims, $\sigma_w^2(t)$ decreases monotonically with $t$. Thus, we consider $T$ such that $\sigma_w^2(T)$ is less than the lower bound.

$$m^{1/L} \geq \alpha\sigma_w^2(T) = \left(1 - \frac{\epsilon\alpha\beta_{p,q}T}{N}\right)\alpha\sigma_w^2(0) = 1 - \frac{\epsilon\alpha\beta_{p,q}T}{N} \qquad (A176)$$

$$\frac{\epsilon\alpha\beta_{p,q}T}{N} \geq 1 - m^{1/L} \qquad (A177)$$

$$T \geq \frac{(1 - m^{1/L})N}{\epsilon\alpha\beta_{p,q}}. \qquad (A178)$$

$\square$

**Theorem 5.8** (Residual networks are adversarially trainable). *Consider a residual network. Suppose that Asms 5.2 and 5.3 hold, and the $(M, m)$-trainability condition holds at $t = 0$ and $\alpha\sigma_w^2(0) \ll 1$. Then, $(M, m)$-trainability condition always holds for $0 \leq t \leq T$.*

*Proof.* By Thm G.9, $\sigma_w^2(t)$ decreases monotonically. Thus, the following inequality holds, and, by Lemma 5.6, the $(M, m)$-trainability condition always holds in $0 \leq t \leq T$:

$$\alpha\sigma_w^2(t) \leq \alpha\sigma_w^2(0) \leq M^{1/L} - 1. \tag{A179}$$

$\square$

**Proposition G.10** (Residual networks are adversarially trainable without careful weight initialization). *Suppose that Asm 5.2 holds, the network is residual, and the $(M, m)$-trainability condition is **not** satisfied at $t = 0$. If*

$$T \geq \frac{(\alpha\sigma_w^2(0) - (M^{1/L} - 1))N}{\epsilon\alpha^{L/2+1}\beta_{p,q}\sigma_w^2(0)^{L/2}}, \tag{A180}$$

*there exists $0 < \tau \leq T$ such that the $(M, m)$-trainability condition hold for $\tau \leq t \leq T$.*

*Proof.* Denote the time evolution of the weight variance on a vanilla network by $\sigma_{w,\text{v}}^2$ and the variance on a residual network by $\sigma_{w,\text{r}}^2$. By Eq. (A161),

$$\frac{\text{d}\sigma_w^2(t)}{\text{d}t} = -\frac{\epsilon\alpha\beta_{p,q}}{N}(1 + \alpha\sigma_w^2(t))^{L/2-1}\sigma_w^2(t) \leq -\frac{\epsilon\alpha^{L/2}\beta_{p,q}}{N}\sigma_w^2(t)^{L/2}. \tag{A181}$$

Recall that the following differential equation represents the time evolution of weight variance on a vanilla network (cf. Thm 5.4):

$$\frac{\text{d}\sigma_w^2(t)}{\text{d}t} = -\frac{\epsilon\alpha^{L/2}\beta_{p,q}}{N}\sigma_w^2(t)^{L/2}. \tag{A182}$$

From the above, $\sigma_{w,\text{r}}^2(t) \leq \sigma_{w,\text{v}}^2(t)$ if $\sigma_w^2(0)$ is the same. Here, we consider $T$ such that $\sigma_{w,\text{v}}^2(T)$ is less than the upper bound of the $(M, m)$-trainability condition. By Thm 5.4,

$$\alpha\sigma_{w,\text{v}}^2(T) \leq M^{1/L} - 1 \tag{A183}$$

$$\left(1 - \frac{\epsilon\alpha\beta_{p,q}\omega_\text{v}(0)^{L'}T}{N}\right)\alpha\sigma_{w,\text{v}}^2(0) \leq M^{1/L} - 1 \tag{A184}$$

$$1 - \frac{\epsilon\alpha\beta_{p,q}\alpha^{L'}\sigma_w^2(0)^{L'}T}{N} \leq \frac{M^{1/L} - 1}{\alpha\sigma_w^2(0)} \tag{A185}$$

$$1 - \frac{M^{1/L} - 1}{\alpha\sigma_w^2(0)} \leq \frac{\epsilon\alpha\beta_{p,q}\alpha^{L'}\sigma_w^2(0)^{L'}T}{N} \tag{A186}$$

$$\frac{(\alpha\sigma_w^2(0) - M^{1/L} + 1)N}{\alpha\sigma_w^2(0)\epsilon\alpha\beta_{p,q}\alpha^{L'}\sigma_w^2(0)^{L'}} \leq T \tag{A187}$$

$$\frac{(\alpha\sigma_w^2(0) - M^{1/L} + 1)N}{\epsilon\alpha^{L/2+1}\beta_{p,q}\sigma_w^2(0)^{L/2}} \leq T. \tag{A188}$$

$\square$

### G.4 Derivation of the theorems in Sec. 5.4

First, we state the following two lemmas as a preliminary.

**Lemma G.11.** *Consider a vanilla network. Suppose that Asm B.1 is applied to the network output. For any $l \in [L]$, the mean squared gradient of $f_i$ with respect to $h_j^{(l)}$ is given by:*

$$\mathbb{E}\left[\left(\frac{\partial f_i}{\partial h_j^{(l)}}\right)^2\right] = \alpha\omega_\text{v}^{L-l}\chi^{(L)}. \tag{A189}$$

*Proof.*

$$\mathbb{E}\left[\left(\frac{\partial f_i}{\partial h_j^{(l)}}\right)^2\right] = \mathbb{E}\left[\left(\frac{\partial f_i}{\partial \boldsymbol{h}^{(l+1)}}\frac{\partial \boldsymbol{h}^{(l+1)}}{\partial x_j^{(l)}}\frac{\partial x_j^{(l)}}{\partial h_j^{(l)}}\right)^2\right] \tag{A190}$$

$$= \mathbb{E}\left[\left(\sum_{k=1}^{N}\frac{\partial f_i}{\partial h_k^{(l+1)}}W_{kj}^{(l+1)}\right)^2 \phi'(h_j^{(l)})^2\right]. \tag{A191}$$

By Asm B.1 and Eq. (A38),

$$\mathbb{E}\left[\left(\frac{\partial f_i}{\partial h_j^{(l)}}\right)^2\right] = \sigma_w^2\mathbb{E}[\phi'(h_j^{(l)})^2]\mathbb{E}\left[\left(\frac{\partial f_i}{\partial h_k^{(l+1)}}\right)^2\right] \tag{A192}$$

$$= \omega_{\mathrm{v}}\mathbb{E}\left[\left(\frac{\partial f_i}{\partial h_k^{(l+1)}}\right)^2\right] \tag{A193}$$

$$= \omega_{\mathrm{v}}^{L-l}\mathbb{E}\left[\left(\frac{\partial f_i}{\partial h_k^{(L)}}\right)^2\right]. \tag{A194}$$

Moreover, by Asm B.1,

$$\mathbb{E}\left[\left(\frac{\partial f_i}{\partial h_j^{(L)}}\right)^2\right] = \mathbb{E}\left[\left(\frac{\partial f_i}{\partial x_j^{(L)}}\frac{\partial x_j^{(L)}}{\partial h_j^{(L)}}\right)^2\right] = \mathbb{E}\left[\left(\frac{\partial f_i}{\partial x_j^{(L)}}\right)^2\phi'(h_j^{(L)})^2\right] = \alpha\chi^{(L)}. \tag{A195}$$

Thus,

$$\mathbb{E}\left[\left(\frac{\partial f_i}{\partial h_i^{(l)}}\right)^2\right] = \alpha\omega_{\mathrm{v}}^{L-l}\chi^{(L)}. \tag{A196}$$

$\square$

**Lemma G.12.** *Suppose that Asm B.1 is applied to the network output. For any $l \in [L]$, the mean squared gradient of $f_i$ with respect to a weight is given by:*

$$\mathbb{E}\left[\left(\frac{\partial f_i}{\partial W_{jk}^{(l)}}\right)^2\right] = \alpha\omega^{L-1}\chi^{(L)}\mathbb{E}[(x_j^{(0)})^2] + \alpha^2\omega^{L-l}\chi^{(L)}\sigma_b^2\sum_{m=1}^{l-1}\omega^{m-1}. \tag{A197}$$

*Proof.* **Vanilla.** By Asm B.1,

$$\mathbb{E}\left[\left(\frac{\partial f_i}{\partial W_{jk}^{(l)}}\right)^2\right] = \mathbb{E}\left[\left(\frac{\partial f_i}{\partial h_j^{(l)}}\frac{\partial h_j^{(l)}}{\partial W_{jk}^{(l)}}\right)^2\right] = \mathbb{E}\left[\left(\frac{\partial f_i}{\partial h_j^{(l)}}\right)^2\right]\mathbb{E}[(x_k^{(l-1)})^2]. \tag{A198}$$

By Lemmas E.11 and G.11,

$$\mathbb{E}\left[\left(\frac{\partial f_i}{\partial W_{jk}^{(l)}}\right)^2\right] = \alpha\omega_{\mathrm{v}}^{L-l}\chi^{(L)}\left(\omega_{\mathrm{v}}^{l-1}\mathbb{E}[(x_j^{(0)})^2] + \alpha\sigma_b^2\sum_{m=1}^{l-1}\omega_{\mathrm{v}}^{m-1}\right) \tag{A199}$$

$$= \alpha\omega_{\mathrm{v}}^{L-1}\chi^{(L)}\mathbb{E}[(x_j^{(0)})^2] + \alpha^2\omega_{\mathrm{v}}^{L-l}\chi^{(L)}\sigma_b^2\sum_{m=1}^{l-1}\omega_{\mathrm{v}}^{m-1}. \tag{A200}$$

**Residual.** By Asm B.1,

$$\mathbb{E}\left[\left(\frac{\partial f_i}{\partial W_{jk}^{(l)}}\right)^2\right] = \mathbb{E}\left[\left(\frac{\partial f_i}{\partial x_j^{(l)}}\frac{\partial x_j^{(l)}}{\partial h_j^{(l)}}\frac{\partial h_j^{(l)}}{\partial W_{jk}^{(l)}}\right)^2\right] \tag{A201}$$

$$= \mathbb{E}\left[\left(\frac{\partial f_i}{\partial x_j^{(l)}}\right)^2\phi'(h_j^{(l)})^2(x_k^{(l-1)})^2\right] \tag{A202}$$

$$= \mathbb{E}\left[\left(\frac{\partial f_i}{\partial x_j^{(l)}}\right)^2\right]\mathbb{E}[\phi'(h_j^{(l)})^2]\mathbb{E}[(x_k^{(l-1)})^2] \tag{A203}$$

$$= \chi^{(l)}\mathbb{E}[\phi'(h_j^{(l)})^2]\mathbb{E}[(x_k^{(l-1)})^2]. \tag{A204}$$

By Eq. (A38),

$$\mathbb{E}\left[\left(\frac{\partial f_i}{\partial W_{jk}^{(l)}}\right)^2\right] = \alpha\chi^{(l)}\mathbb{E}[(x_k^{(l-1)})^2]. \tag{A205}$$

By Eqs. (A19) and (A38),

$$\chi^{(l)} = \omega_{\mathrm{r}}^{L-l}\chi^{(L)}. \tag{A206}$$

Thus, by Lemma E.11,

$$\mathbb{E}\left[\left(\frac{\partial f_i}{\partial W_{jk}^{(l)}}\right)^2\right] = \alpha\omega_{\mathrm{r}}^{L-l}\chi^{(L)}\left(\omega_{\mathrm{r}}^{l-1}\mathbb{E}[(x_j^{(0)})^2] + \alpha\sigma_b^2\sum_{m=1}^{l-1}\omega_{\mathrm{r}}^{m-1}\right) \tag{A207}$$

$$= \alpha\omega_{\mathrm{r}}^{L-1}\chi^{(L)}\mathbb{E}[(x^{(0)})^2] + \alpha^2\omega_{\mathrm{r}}^{L-l}\chi^{(L)}\sigma_b^2\sum_{m=1}^{l-1}\omega_{\mathrm{r}}^{m-1}. \tag{A208}$$

$$\square$$

Now, we can simply represent the mean of the Fisher–Rao norm.

**Lemma G.13.** *Suppose that Asm B.1 is applied to the network output. Suppose $\|\boldsymbol{x}^{\mathrm{in}}\|_2 = \sqrt{d}$ and $\sigma_b^2 = 0$. The mean of the Fisher–Rao norm is given by:*

$$\mathbb{E}[\|\boldsymbol{w}\|_{\mathrm{FR}}] = LK\alpha\sigma_w^2\omega^{L-1}. \tag{A209}$$

*Proof.* The following discussion is based on [49]. We rearrange and expand their discussion for a ReLU-like network. We also consider it for a residual network. Note that $\boldsymbol{w} := (W_{11}^{(1)}, \dots, W_{NN}^{(L)})^\top$. As a preliminary, see Remark E.9. The Fisher–Rao norm is calculated as follows:

$$\|\boldsymbol{w}\|_{\mathrm{FR}} := \boldsymbol{w}^\top\boldsymbol{F}\boldsymbol{w} = \sum_{i=1}^{LN^2}\sum_{j=1}^{LN^2}w_iF_{ij}w_j = \sum_{i=1}^{LN^2}\sum_{j=1}^{LN^2}\sum_{k=1}^{K}\frac{\partial f_k}{\partial w_i}\frac{\partial f_k}{\partial w_j}w_iw_j. \tag{A210}$$

By Asm B.1,

$$\mathbb{E}[\|\boldsymbol{w}\|_{\mathrm{FR}}] = \sum_{i=1}^{LN^2}\sum_{j=1}^{LN^2}\sum_{k=1}^{K}\mathbb{E}\left[\frac{\partial f_k}{\partial w_i}\frac{\partial f_k}{\partial w_j}w_iw_j\right] \tag{A211}$$

$$= \sum_{i=1}^{LN^2}\sum_{j=1}^{LN^2}\sum_{k=1}^{K}\mathbb{E}\left[\frac{\partial f_k}{\partial w_i}\frac{\partial f_k}{\partial w_j}\right]\mathbb{E}[w_iw_j] \tag{A212}$$

$$= K\frac{\sigma_w^2}{N}\sum_{i=1}^{LN^2}\mathbb{E}\left[\left(\frac{\partial f_k}{\partial w_i}\right)^2\right]. \tag{A213}$$

By Lemma G.12, $\mathbb{E}[(\partial f_i/\partial W_{jk}^{(l)})^2]$ is invariant to $l$ if $\sigma_b^2 = 0$. In other words, $\mathbb{E}[(\partial f_k/\partial w_i)^2]$ does not depend on $i$ if $\sigma_b^2 = 0$. Thus,

$$\mathbb{E}[\|\boldsymbol{w}\|_{\mathrm{FR}}] = LNK\sigma_w^2\mathbb{E}\left[\left(\frac{\partial f_k}{\partial w_i}\right)^2\right] = LNK\alpha\sigma_w^2\omega^{L-1}\chi^{(L)}\mathbb{E}[(x_i^{(0)})^2]. \tag{A214}$$

Moreover,

$$\chi^{(L)} := \mathbb{E}\left[\left(\frac{\partial f_i}{\partial x_j^{(L)}}\right)^2\right] = \mathbb{E}\left[\left(\frac{\partial \sum_{k=1}^N P_{ik}^{\mathrm{out}} x_k^{(L)}}{\partial x_j^{(L)}}\right)^2\right] = \mathbb{E}[(P_{ij}^{\mathrm{out}})^2] = \frac{1}{N}, \tag{A215}$$

$$\mathbb{E}[(x_i^{(0)})^2] = \mathbb{E}\left[\left(\sum_{j=1}^d P_{ij}^{\mathrm{in}} x_j^{\mathrm{in}}\right)^2\right] = \sum_{j=1}^d \mathbb{E}[(P_{ij}^{\mathrm{in}})^2](x_j^{\mathrm{in}})^2 = 1. \tag{A216}$$

Thus,

$$\mathbb{E}[\|\boldsymbol{w}\|_{\mathrm{FR}}] = LK\alpha\sigma_w^2\omega^{L-1}. \tag{A217}$$

$\square$

**Theorem 5.9** (Adversarial training degrades network capacity). *Consider a vanilla network. Suppose that Asms 5.2 and 5.3 hold, Asm B.1 is applied to the network output, and the $(M,m)$-trainability condition holds at $t = 0$ and $\alpha\sigma_w^2(0) = 1$. Assume $\|\boldsymbol{x}^{\mathrm{in}}\|_2 = \sqrt{d}$ and $\sigma_b^2(t) = 0$. Then, the expectation of the Fisher–Rao norm is given by:*

$$\mathbb{E}[\|\boldsymbol{w}(t)\|_{\mathrm{FR}}] = LK\left(1 - \frac{\epsilon\alpha\beta_{p,q}L}{N}t\right). \tag{15}$$

*Proof.* By Lemma G.13,

$$\mathbb{E}[\|\boldsymbol{w}\|_{\mathrm{FR}}] = LK\alpha\sigma_w^2(t)\omega_{\mathrm{v}}(t)^{L-1} = LK\alpha^L\sigma_w^2(t)^L. \tag{A218}$$

By Thm 5.4,

$$\mathbb{E}[\|\boldsymbol{w}\|_{\mathrm{FR}}] = LK\alpha^L\sigma_w^2(0)^L\left(1 - \frac{\epsilon\alpha\beta_{p,q}t}{N}\right)^L = LK\left(1 - \frac{\epsilon\alpha\beta_{p,q}t}{N}\right)^L. \tag{A219}$$

By $\epsilon\alpha\beta_{p,q}t/N \ll 1$,

$$\mathbb{E}[\|\boldsymbol{w}\|_{\mathrm{FR}}] \approx LK\left(1 - \frac{L\epsilon\alpha\beta_{p,q}t}{N}\right). \tag{A220}$$

$\square$

**Theorem G.14** (Adversarial training degrades network capacity). *Consider a residual network. Suppose that Asm 5.2 holds, Asm B.1 is applied to the network output, and the $(M,m)$-trainability condition holds at $t = 0$, e.g., $\alpha\sigma_w^2(0) \ll 1$. Suppose $\|\boldsymbol{x}^{\mathrm{in}}\|_2 = \sqrt{d}$ and $\sigma_b^2(t) = 0$. Then, the mean of the Fisher–Rao norm is given by:*

$$\mathbb{E}[\|\boldsymbol{w}\|_{\mathrm{FR}}] = LK\alpha\sigma_w^2(0)\left(\begin{array}{c} 1 + (L-1)\alpha\sigma_w^2(0) \\ -(2(L-1)\alpha\sigma_w^2(0) + 1)\left(1 + \left(\frac{L}{2} - 1\right)\alpha\sigma_w^2(0)\right)\frac{\epsilon\alpha\beta_{p,q}t}{N} \end{array}\right).$$

*Proof.* By Lemma G.13,

$$\mathbb{E}[\|\boldsymbol{w}\|_{\mathrm{FR}}] = LK\alpha\sigma_w^2(t)\omega_{\mathrm{r}}(t)^{L-1} \tag{A221}$$
$$= LK\alpha\sigma_w^2(t)(1 + \alpha\sigma_w^2(t))^{L-1} \tag{A222}$$
$$\approx LK\alpha\sigma_w^2(t)(1 + (L-1)\alpha\sigma_w^2(t)). \tag{A223}$$

Temporally, denote $A := (1 + (L/2 - 1)\alpha\sigma_w^2(0))\epsilon\alpha\beta_{p,q}t/N$. By Thm G.9,

$$\mathbb{E}[\|\boldsymbol{w}\|_{\mathrm{FR}}] = LK\alpha(1 - A)\sigma_w^2(0)(1 + (L-1)\alpha(1 - A)\sigma_w^2(0)) \tag{A224}$$

$$= LK\alpha\sigma_w^2(0)(1 - A)(1 + (L-1)\alpha\sigma_w^2(0) - (L-1)A\alpha\sigma_w^2(0)) \tag{A225}$$

$$= LK\alpha\sigma_w^2(0)\begin{pmatrix} 1 - A + (L-1)(1-A)\alpha\sigma_w^2(0) \\ -(L-1)A\alpha\sigma_w^2(0) + \mathcal{O}(A^2) \end{pmatrix}. \tag{A226}$$

By $\mathcal{O}(1/N^2) \approx 0$, $\mathcal{O}(A^2) \approx 0$. Thus,

$$\mathbb{E}[\|\boldsymbol{w}\|_{\mathrm{FR}}] \approx LK\alpha\sigma_w^2(0)(1 - A + (L-1)(1-A)\alpha\sigma_w^2(0) - (L-1)A\alpha\sigma_w^2(0)) \tag{A227}$$

$$= LK\alpha\sigma_w^2(0)(1 - A + (L-1)\alpha\sigma_w^2(0) - 2(L-1)A\alpha\sigma_w^2(0)) \tag{A228}$$

$$= LK\alpha\sigma_w^2(0)(1 + (L-1)\alpha\sigma_w^2(0) - (2(L-1)\alpha\sigma_w^2(0) + 1)A) \tag{A229}$$

$$= LK\alpha\sigma_w^2(0)\begin{pmatrix} 1 + (L-1)\alpha\sigma_w^2(0) \\ -(2(L-1)\alpha\sigma_w^2(0) + 1)\left(1 + \left(\dfrac{L}{2} - 1\right)\alpha\sigma_w^2(0)\right)\dfrac{\epsilon\alpha\beta_{p,q}t}{N} \end{pmatrix}. \tag{A230}$$

$\square$

## H Comparison with matrix decomposition approaches

Here, we analyze Ineq. (8) with another approach, which decomposes $\boldsymbol{J}(\boldsymbol{x}^{\mathrm{in}})$ into products of matrices, and consider norm of each matrix. As related studies based on this approach, we cite the literature on certified adversarial defense [3, 18, 93] and spectral regularization [60, 108]. We consider the case of $(p, q) = (2, 2)$ for simplicity. In comparison to them, our approach is different in two ways. First, their bound is not tractable due to a deterministic approach. For example, they consider the spectral norm of $\boldsymbol{J}(\boldsymbol{x}^{\mathrm{in}})$ by decomposing $\boldsymbol{J}(\boldsymbol{x}^{\mathrm{in}})$ and computing the norm of each matrix consisting $\boldsymbol{J}(\boldsymbol{x}^{\mathrm{in}})$ as follows:

$$\left\|\boldsymbol{J}(\boldsymbol{x}^{\mathrm{in}})\right\|_2 = \left\|\boldsymbol{P}^{\mathrm{out}}\boldsymbol{D}(\phi'(\boldsymbol{h}^{(L)}(\boldsymbol{x}^{\mathrm{in}})))\boldsymbol{W}^{(L)}\cdots\boldsymbol{D}(\phi'(\boldsymbol{h}^{(1)}(\boldsymbol{x}^{\mathrm{in}})))\boldsymbol{W}^{(1)}\boldsymbol{P}^{\mathrm{in}}\right\|_2 \tag{A231}$$

$$\leq \left\|\boldsymbol{P}^{\mathrm{out}}\right\|_2\left\|\boldsymbol{D}(\phi'(\boldsymbol{h}^{(L)}(\boldsymbol{x}^{\mathrm{in}})))\right\|_2\left\|\boldsymbol{W}^{(L)}\right\|_2$$

$$\cdots\left\|\boldsymbol{D}(\phi'(\boldsymbol{h}^{(1)}(\boldsymbol{x}^{\mathrm{in}})))\right\|_2\left\|\boldsymbol{W}^{(1)}\right\|_2\left\|\boldsymbol{P}^{\mathrm{in}}\right\|_2 \tag{A232}$$

$$\leq \max(|u|, |v|)^L\left\|\boldsymbol{P}^{\mathrm{in}}\right\|_{\mathrm{F}}\left\|\boldsymbol{P}^{\mathrm{out}}\right\|_{\mathrm{F}}\prod_{l=1}^{L}\left\|\boldsymbol{W}^{(l)}\right\|_{\mathrm{F}}. \tag{A233}$$

Because Eqs. (A232) and (A233) are hard to theoretically manage, we cannot derive some theorems in this study such as Thms 5.4, 5.7 and 5.9. Second, because we calculate the spectral norm of $\boldsymbol{J}(\boldsymbol{x}^{\mathrm{in}})$ directly in Thm 5.1 instead of considering the norm of each matrix and the spectral norm is submultiplicative, our upper bounds are always tighter than those derived from Eq. (A232).

Here, we proceed with the discussion of tightness of their bound using probabilistic theory. One approach is the rearrangement of Eq. (A232) using the Marchenko–Pastur law. As a preliminary, we note the following:

$$\left\|\boldsymbol{P}^{\mathrm{in}}\right\|_2 \leq 1 + \sqrt{\frac{N}{d}}, \qquad \left\|\boldsymbol{P}^{\mathrm{out}}\right\|_2 \leq 1 + \sqrt{\frac{K}{N}}, \qquad \left\|\boldsymbol{W}^{(l)}\right\|_2 \leq 2\sqrt{\sigma_w^2}. \tag{A234}$$

Using the above equations, Eq. (A232) can be rearranged as follows:

$$\left\|\boldsymbol{J}(\boldsymbol{x}^{\mathrm{in}})\right\|_2 \leq \max(|u|, |v|)^L\left(1 + \sqrt{\frac{N}{d}}\right)\left(1 + \sqrt{\frac{K}{N}}\right)2^L(\sigma_w^2)^{L/2}. \tag{A235}$$

This is exponentially looser than Thm 5.1.

Another approach is the rearrangement of Eq. (A233) using the assumption that each matrix is sufficiently large. As a preliminary, we note the following:

$$\left\|\boldsymbol{P}^{\mathrm{in}}\right\|_{\mathrm{F}} = \sqrt{Nd\frac{1}{Nd}\sum_{i=1}^{N}\sum_{j=1}^{d}(P_{ij}^{\mathrm{in}})^2} = \sqrt{Nd\mathbb{E}[(P_{ij}^{\mathrm{in}})^2]} = \sqrt{Nd\frac{1}{d}} = \sqrt{N}, \tag{A236}$$

$$\left\|\boldsymbol{P}^{\mathrm{out}}\right\|_{\mathrm{F}} = \sqrt{KN\frac{1}{KN}\sum_{i=1}^{K}\sum_{j=1}^{N}(P_{ij}^{\mathrm{out}})^2} = \sqrt{KN\mathbb{E}[(P_{ij}^{\mathrm{out}})^2]} = \sqrt{KN\frac{1}{N}} = \sqrt{K}, \tag{A237}$$

$$\left\|\boldsymbol{W}^{(l)}\right\|_{\mathrm{F}} = \sqrt{N^2\frac{1}{N^2}\sum_{i=1}^{N}\sum_{j=1}^{N}(\boldsymbol{W}_{ij}^{(l)})^2} = \sqrt{N^2\mathbb{E}[(\boldsymbol{W}_{ij}^{(l)})^2]} = \sqrt{N^2\frac{\sigma_w^2}{N}} = \sqrt{N\sigma_w^2}. \tag{A238}$$

Using the above equations, Eq. (A233) can be rearranged as follows:

$$\left\|\boldsymbol{J}(\boldsymbol{x}^{\mathrm{in}})\right\|_2 \leq \max(|u|,|v|)^L \sqrt{N}\sqrt{K}\sqrt{N\sigma_w^2}^L = \max(|u|,|v|)^L K^{1/2} N^{(L+1)/2}(\sigma_w^2)^{L/2}. \tag{A239}$$

This is also exponentially looser than Thm 5.1.

# I  Comparison with $\ell_2$ weight regularization

As stated in Secs. 5.1 and 5.2, adversarial training serves the role of a weight regularizer. Here, we compare adversarial training with $\ell_2$ weight regularization. First, we define $\ell_2$ weight regularization as follows:

$$\mathcal{L}_{\mathrm{w}}(\mathcal{W};\lambda) := \lambda \sum_{W\in\mathcal{W}} W^2, \tag{A240}$$

where $\lambda > 0$ is a scaling factor. Then, we determine the scaling factor $\lambda$ based on the concept of gradient vanishing and explosion. The mean of $\mathcal{L}_{\mathrm{w}}(\mathcal{W};\lambda)$ is calculated as follows:

$$\mathbb{E}[\mathcal{L}_{\mathrm{w}}(\mathcal{W};\lambda)] = \lambda \sum_{W\in\mathcal{W}} \mathbb{E}[W^2] = \lambda LN^2 \mathbb{E}[W^2] = \lambda LN^2 \frac{\sigma_w^2}{N} = \lambda LN\sigma_w^2. \tag{A241}$$

To prevent (gradient) vanishing and explosion of $\mathbb{E}[\mathcal{L}_{\mathrm{w}}(\mathcal{W};\lambda)]$ under sufficiently large $L$ and $N$, $\lambda$ should be $1/(LN)$. Moreover, for simplicity of the derivation, we set $\lambda := 1/(2LN)$. Finally, $\ell_2$ regularized training tries to minimize the following loss function:

$$\mathcal{L} := \mathcal{L}_{\mathrm{std}} + \frac{1}{2LN} \sum_{W\in\mathcal{W}} W^2 \tag{A242}$$

Next, we consider the time evolution of $\sigma_w^2$ with $\ell_2$ weight regularization. Similar to Thms 5.4 and G.9, we can derive the following proposition:

**Proposition I.1** (Weight time evolution with $\ell_2$ weight regularization)**.** *Suppose that Asm 5.2 holds. Let $\mathcal{L}_{\mathrm{std}}$ be the standard loss function and $\mathcal{L}_{\mathrm{w}}(\mathcal{W}) := 1/(2LN)\sum_{W\in\mathcal{W}} W^2$ be the $\ell_2$ regularization loss function. Suppose that Asm B.1 applies to $\mathcal{L}_{\mathrm{std}}$, but not to $\mathcal{L}_{\mathrm{w}}$. A network is trained by minimizing $\mathcal{L}_{\mathrm{std}} + \mathcal{L}_{\mathrm{w}}(\mathcal{W})$. Then, the time evolution of $\sigma_w^2$ is given by:*

$$\sigma_w^2(t) = \left(1 - \frac{t}{LN}\right)\sigma_w^2(0). \tag{A243}$$

*Proof.* Similar to Lemma G.8,

$$\frac{\mathrm{d}\sigma_w^2(t)}{\mathrm{d}t} = -N\mathbb{E}\left[W(t)\frac{\partial\mathcal{L}_{\mathrm{w}}(\mathcal{W})}{\partial W(t)}\right] \tag{A244}$$

$$= -N\mathbb{E}\left[W(t)\frac{W(t)}{LN}\right] \tag{A245}$$

$$= -\frac{1}{L}\mathbb{E}[W(t)^2] \tag{A246}$$

$$= -\frac{1}{LN}\sigma_w^2(t). \tag{A247}$$

Thus,

$$\sigma_w^2(t) = \exp\left(-\frac{t}{LN}\right)\sigma_w^2(0). \tag{A248}$$

By $t \leq T \ll N$, using Maclaurin expansion of the exponential function,

$$\sigma_w^2(t) = \left(1 - \frac{t}{LN}\right)\sigma_w^2(0). \tag{A249}$$

$\square$

Here, we consider the adversarial loss defined as $\mathcal{L}_{\mathrm{adv}}(\boldsymbol{x}^{\mathrm{in}}) := \epsilon\beta_{p,q}\omega^{L/2}$ (cf. Thm 5.1). The difference between adversarial training and $\ell_2$ regularization lies in the scale: $\epsilon\alpha\beta_{p,q}/N$ in adversarial training (cf. Thm 5.4) and $1/(LN)$ in $\ell_2$ regularized training. Adversarial training with strong adversarial examples (e.g., $\ell_\infty$ constrained ones) reduces $\sigma_w^2(t)$ more drastically than $\ell_2$ regularized training. For example, with $L = 100$, $N = 1,000$, $d = 3 \times 224 \times 224$, $\epsilon = 0.1$, $\alpha = 1/2$, $p = \infty$, $q = \infty$, then $\Theta(\epsilon\alpha\beta_{p,q}/N) = 10^{-2}$ and $\Theta(1/(LN)) = 10^{-5}$.

Finally, we compare adversarial training and $\ell_2$ regularization in terms of stability of gradient descent. The derivations of both losses with respect to the weight $W \in \mathcal{W}$ are given by (cf. Lemma G.7):

$$\frac{\partial\mathcal{L}_{\mathrm{adv}}}{\partial W} = \frac{\epsilon\alpha\beta_{p,q}\omega^{L/2-1}W}{N}, \qquad\qquad \frac{\partial\mathcal{L}_{\mathrm{w}}}{\partial W} = \frac{W}{LN}. \tag{A250}$$

This indicates that the derivation of $\ell_2$ regularizer is smooth across the entire weight space, whereas the derivation of adversarial loss changes significantly at the $\omega = 1$ boundary. Note that $\omega$ evolves during training (cf. the definition in Thm 4.1). The steep derivation in adversarial training prevents the gradient descent from reaching a minimum.

## J   Revisiting trainability condition in adversarial training

Here, we revisit Defn 5.5. Recall that this condition can be derived exclusively under Asm B.1. This is because $\chi^{(l)}$ can only be computed under Asm B.1 [81]. In this study, we assume that Asm B.1 applies to the standard loss $\mathcal{L}_{\mathrm{std}}$, but not to the adversarial loss $\mathcal{L}_{\mathrm{adv}}$. Consequently, the composite loss $\mathcal{L} := \mathcal{L}_{\mathrm{std}} + \mathcal{L}_{\mathrm{adv}}$ does not satisfy Asm B.1, implying that strictly speaking, Defn 5.5 is not valid for $\mathcal{L}$ and considering Defn 5.5 in adversarial training might not be entirely accurate.

However, we must emphasize that Defn 5.5 continues to be a determining factor in the success of adversarial training, or in other words, training with $\mathcal{L}$. This is primarily due to the fact that if Defn 5.5 is not met, networks will be unable to adequately fit the training dataset owing to gradient vanishing or explosion.

Examining this from a mathematical standpoint, the necessity of Defn 5.5 is underlined by the linearity of the differential operator. The update equation for the parameter $\theta(t)$ is formulated as follows:

$$\frac{\mathrm{d}\theta(t)}{\mathrm{d}t} := -\frac{\partial\mathcal{L}}{\partial\theta(t)} = -\frac{\partial\mathcal{L}_{\mathrm{std}}}{\partial\theta(t)} - \frac{\partial\mathcal{L}_{\mathrm{adv}}}{\partial\theta(t)} \tag{A251}$$

As demonstrated by the equation above, updates with $\mathcal{L}_{\mathrm{std}}$ and $\mathcal{L}_{\mathrm{adv}}$ are conducted independently. Although Defn 5.5 may not be considered for $\mathcal{L}_{\mathrm{adv}}$, it can be for $\mathcal{L}_{\mathrm{std}}$. In order to forestall gradient vanishing or explosion in the standard loss and to ensure classification ability for clean images, Defn 5.5 remains a necessity. Therefore, it can be concluded that Defn 5.5 continues to dictate the success of training, even within the context of adversarial training.

## K   Other theoretical results

**Scaled effect of input and output dimensions.**   In Tab. 1, we showed a wide effect of the input dimension $d$ and the output dimension $K$ on the upper bounds of the adversarial loss. In practice, the perturbation budget $\epsilon$ and the adversarial loss are scaled based on the chosen $p$ and $q$. For example, we can rearrange the definition of the adversarial loss (Eq. (3)) as follows:

$$\mathcal{L}_{\mathrm{adv}}(\boldsymbol{x}^{\mathrm{in}}) := \max_{\|\boldsymbol{\eta}\|_p \leq \lambda_p\epsilon} \lambda_q\big\|\boldsymbol{f}(\boldsymbol{x}^{\mathrm{in}} + \boldsymbol{\eta}) - \boldsymbol{f}(\boldsymbol{x}^{\mathrm{in}})\big\|_q, \tag{A252}$$

Table A4: Values of $\beta'_{p,q}$. The description is the same as Tab. 1

|  | $q = 1$ | $q = 2$ | $q = \infty$ |
|---|---|---|---|
| $p = 1$ | $\sqrt{\frac{2d}{\pi}}^{\dagger}$ | $\sqrt{d}^{\dagger}$ | $\sqrt{2d \ln K}$ |
| $p = 2$ |  | $\sqrt{\frac{d}{K}}^{\diamond} + 1$ | $\sqrt{d}^{\dagger\diamond}$ |
| $p = \infty$ |  |  | $\sqrt{\frac{2d}{\pi}}^{\dagger\diamond}$ |

where $\lambda_p$ and $\lambda_q$ denote the scaling factors. In this context, the upper bound (Thm 5.1) can be rearranged as follows:

$$\mathcal{L}_{\mathrm{adv}}(\boldsymbol{x}^{\mathrm{in}}) \leq \epsilon \cdot \lambda_p \lambda_q \beta_{p,q} \cdot \omega^{L/2} = \epsilon \beta'_{p,q} \omega^{L/2}, \tag{A253}$$

where $\beta'_{p,q} := \lambda_p \lambda_q \beta_{p,q}$. For example, we consider the following $\lambda_p$ and $\lambda_q$:

$$\lambda_p := \begin{cases} d & (p = 1) \\ \sqrt{d} & (p = 2) \\ 1 & (p = \infty) \end{cases}, \qquad \lambda_q := \begin{cases} 1/K & (q = 1) \\ 1/\sqrt{K} & (q = 2) \\ 1 & (q = \infty) \end{cases}. \tag{A254}$$

Under these conditions, $\beta'_{p,q}$ corresponds to Tab. A4. This table reveals that (i) the input dimension impacts the upper bounds with $\Theta(\sqrt{d})$, but the output dimension has little or no effects; (ii) although the dimensions generally exert a similar influence on the bounds, the specific factors vary widely; (iii) for $(p,q) = (1,\infty)$ and $(2,2)$, the output dimension behaves in a distinctive manner. In conclusion, our upper bounds, even inclusive of scaling effects, provide several insights into the relationship between the adversarial loss and input/output dimensions.

**Other metrics of capacity.** Here, we consider capacity metrics other than the Fisher–Rao norm. Norm-based metrics, such as the path norm [66], the group norm [67], and spectral norm [8], have a similar definition to the Fisher–Rao norm. They are constructed by the sum and product of the weights of a network. The trajectory length [73] is defined by the arc length of a network output with change of parameters. A similar definition based on curvature was also used in [71]. These metrics concluded that capacity increases with the weight variance of a network, despite differences in speed. Therefore, we can derive similar results as in Sec. 5.4 for metrics other than the Fisher–Rao norm.

**Mitigation of adversarial risk.** Here, we consider the mitigation of adversarial risk, i.e., the mitigation of the upper bounds of the adversarial loss. There are three solutions: (i) sample each entry of $\boldsymbol{P}^{\mathrm{in}}$ from $\mathcal{N}(0, 1/d^2)$ instead of $\mathcal{N}(0, 1/d)$; (ii) sample each entry of $\boldsymbol{W}^{(l)}$ from $\mathcal{N}(0, \sigma_w^2/N^2)$ instead of $\mathcal{N}(0, \sigma_w^2/N)$ for $l \in [L]$; (iii) sample each entry of $\boldsymbol{P}^{\mathrm{out}}$ from $\mathcal{N}(0, 1/N^2)$ instead of $\mathcal{N}(0, 1/N)$.

First, let us examine scenario (i). In this setting, the variance of $J(\boldsymbol{x}^{\mathrm{in}})$ is transformed into $\mathbb{V}[J(\boldsymbol{x}^{\mathrm{in}})_{ij}] = \omega^L/d^2$ from $\mathbb{V}[J(\boldsymbol{x}^{\mathrm{in}})_{ij}] = \omega^L/d$ (cf. the proof of Thm 4.1). In addition, $\beta_{\infty,\infty}$ changes to $\Theta(\beta_{\infty,\infty}) = 1$ from $\Theta(\beta_{\infty,\infty}) = \sqrt{d}$ (cf. the proof of Thm 5.1). These modifications suggest a potential reduction in adversarial risk in scenario (i).

However, this leads to a training failure. Let us consider the variance of a network output under no bias (for simplicity). For sufficiently large input dimension $d$ and $\boldsymbol{x}^{\mathrm{in}} \in [0,1]^d$, it can be calculated as follows:

$$\mathbb{V}[\boldsymbol{f}(\boldsymbol{x}^{\mathrm{in}})] = \mathbb{V}[\boldsymbol{J}(\boldsymbol{x}^{\mathrm{in}})\boldsymbol{x}^{\mathrm{in}}] = \mathbb{V}[J(\boldsymbol{x}^{\mathrm{in}})_{ij}]\|\boldsymbol{x}^{\mathrm{in}}\|_2^2 = \frac{\|\boldsymbol{x}^{\mathrm{in}}\|_2^2}{d^2} \approx 0. \tag{A255}$$

This equation implies that the network always outputs a zero vector for any input $\boldsymbol{x}^{\mathrm{in}}$. In other words, input information is lost during signal propagation in the network. In this situation, networks cannot learn the data structure well, and thus training does not proceed (cf. [71, 81, 105]).

Situations (2) and (3) are also similar to (1); they can mitigate the adversarial risk, but they break input information during forward and backward. Therefore, we conclude that it is not feasible to mitigate the adversarial effect without compromising the network's trainability.

**Extension to Lipschitz continuous activations.** In this study, we primarily established our theorems for networks employing ReLU-like activations. Here, we attempt to generalize these theorems to encompass networks with Lipschitz continuous activations. However, this extension involve looser upper bounds or potentially unrealistic assumptions.

First, we examine the upper bounds for $p = 2$ and $q = 2$. Since $\boldsymbol{J}(\boldsymbol{x}^{\text{in}})$ is a Jacobian at $\boldsymbol{x}^{\text{in}}$, Ineq. (8) remains valid for Lipschitz continuous activations. Denote the Lipschitz constant by $k \geq 0$. Similar to the discussion in Appx. H, we can derive the following upper bound:

$$\left\| \boldsymbol{J}(\boldsymbol{x}^{\text{in}}) \right\|_2 \leq k^L \left( 1 + \sqrt{\frac{N}{d}} \right) \left( 1 + \sqrt{\frac{K}{N}} \right) 2^L (\sigma_w^2)^{L/2}. \tag{A256}$$

An important observation here is that $\omega^{L/2}$, which dictates the training properties, is common to networks with both ReLU-like and Lipschitz continuous activations. Consequently, we claim that our theorems also extend to Lipschitz continuous activations, despite differences in the factor. However, this bound is exponentially loose, casting doubt on its ability to accurately represent the properties of adversarial loss.

Second, we use the analysis of a network Jacobian [69]. For a comprehensive comparison between our approach and theirs, see Sec. 5.1. Here, we adopt their assumption where the variance $\mathbb{V}[h^{(l)}]$ is constant for any $l \in [L]$. Drawing from the results expressed in Eq. (22) of [69], we can assert that $\left\| \boldsymbol{J}(\boldsymbol{x}^{\text{in}}) \right\|_2 \leq \Theta((\sigma_w^2)^{L/2})$ holds. Similar to the first proposition, this implies that our theorems hold to general activations including Lipschitz continuous activations. However, the assumption clearly does not hold in our theorems such as Thm 5.4.

**Single-gradient descent attack can find adversarial examples.** Here, we demonstrate that a single-gradient descent attack, such as the fast gradient sign method [38], can find adversarial examples. This is not a substantially novel contribution. For example, see [7, 11, 24, 25, 61, 84, 112]. However, to introduce a new approach based on Thm 4.1, we attempt it.

**Proposition K.1** (Single-gradient descent attack can find adversarial examples). *Suppose that* $\left\| \boldsymbol{x}^{\text{in}} \right\|_2 = \sqrt{d}$, *the perturbation constraint* $\epsilon$ *is sufficiently small, and the input dimension* $d$ *is sufficiently large. Then, the single-gradient descent attack finds* $\ell_\infty$ *constrained adversarial examples that flip the prediction of a single-output random deep neural network with high probability*

$$\text{erf} \left( \epsilon \sqrt{\frac{\omega^L d}{\pi \left( \omega^L + \alpha \sigma_b^2 \sum_{k=1}^L \omega^{k-1} \right)}} \right) \xrightarrow{d \to \infty} 1, \tag{A257}$$

*where* erf *is the error function.*

*Proof.* Since $\boldsymbol{\eta}$ is sufficiently small, the same linear regions in a ReLU-like network encompass both $\boldsymbol{x}^{\text{in}}$ and $\boldsymbol{x}^{\text{in}} + \boldsymbol{\eta}$, i.e., $\boldsymbol{J}(\boldsymbol{x}^{\text{in}}) = \boldsymbol{J}(\boldsymbol{x}^{\text{in}} + \boldsymbol{\eta})$. When an $\ell_\infty$ constrained adversarial example flips the prediction of a classifier, the inequality $|\boldsymbol{J}(\boldsymbol{x}^{\text{in}})\boldsymbol{x}^{\text{in}} + \boldsymbol{a}(\boldsymbol{x}^{\text{in}})| < |\boldsymbol{J}(\boldsymbol{x}^{\text{in}})\boldsymbol{\eta}|$ holds for $|\boldsymbol{\eta}|_\infty \leq \epsilon$. The perturbation generated by a single-gradient descent is defined as $\boldsymbol{\eta} := \epsilon \, \text{sgn}(\boldsymbol{J}(\boldsymbol{x}^{\text{in}}))$. Note that $\boldsymbol{J}(\boldsymbol{x}^{\text{in}})$ is a one-dimensional vector in single-output networks. Using this perturbation, by Eq. (A87) and Thm 4.1, the right term of the inequality can be transformed into:

$$|\boldsymbol{J}(\boldsymbol{x}^{\text{in}})\boldsymbol{\eta}| \leq \epsilon d \frac{1}{d} \sum_{i=1}^d |J(\boldsymbol{x}^{\text{in}})_i| = \epsilon d \mathbb{E}[|J(\boldsymbol{x}^{\text{in}})_i|] = \epsilon d \sqrt{\frac{2\omega^L}{\pi d}} = \epsilon \sqrt{\frac{2\omega^L d}{\pi}}. \tag{A258}$$

Then, let us consider $|\boldsymbol{J}(\boldsymbol{x}^{\text{in}})\boldsymbol{x}^{\text{in}} + \boldsymbol{a}(\boldsymbol{x}^{\text{in}})|$. By the reproductive property of the Gaussian, $\boldsymbol{J}(\boldsymbol{x}^{\text{in}})\boldsymbol{x}^{\text{in}} + \boldsymbol{a}(\boldsymbol{x}^{\text{in}})$ follows a Gaussian $\mathcal{N}(0, \left\| \boldsymbol{x}^{\text{in}} \right\|_2^2 \mathbb{V}[J(\boldsymbol{x}^{\text{in}})] + \mathbb{V}[a(\boldsymbol{x}^{\text{in}})])$. The variance can be rearranged as follows:

$$\left\| \boldsymbol{x}^{\text{in}} \right\|_2^2 \mathbb{V}[J(\boldsymbol{x}^{\text{in}})_{ij}] + \mathbb{V}[a(\boldsymbol{x}^{\text{in}})_i] = \frac{\left\| \boldsymbol{x}^{\text{in}} \right\|_2^2 \omega^L}{d} + \alpha \sigma_b^2 \sum_{k=1}^L \omega^{k-1} = \omega^L + \alpha \sigma_b^2 \sum_{k=1}^L \omega^{k-1}. \tag{A259}$$

Note that the c.d.f of the folded normal distribution based on the Gaussian $\mathcal{N}(0, \sigma^2)$ is given by $\text{erf}(x/\sqrt{2\sigma^2})$. Thus, we can compute the probability such that $|\boldsymbol{J}(\boldsymbol{x}^{\text{in}})\boldsymbol{x}^{\text{in}} + \boldsymbol{a}(\boldsymbol{x}^{\text{in}})|$ is less than

$\epsilon \sqrt{2\omega^L d/\pi}$ as follows:

$$\text{erf}\left(\frac{\epsilon\sqrt{2\omega^L d/\pi}}{\sqrt{2\left(\omega^L + \alpha\sigma_b^2 \sum_{k=1}^{L} \omega^{k-1}\right)}}\right) = \text{erf}\left(\epsilon\sqrt{\frac{\omega^L d}{\pi\left(\omega^L + \alpha\sigma_b^2 \sum_{k=1}^{L} \omega^{k-1}\right)}}\right) \tag{A260}$$

$$\xrightarrow{d\to\infty} 1. \tag{A261}$$

$\square$

## L   Other experimental results

### L.1   Setting

We focused on ReLU networks, i.e., $u = 1$, $v = 0$, and $\alpha = 1/2$. Adversarial examples were generated using auto projected gradient descent [20] and the loss function defined in Eq. (3). Networks were initialized to satisfy the $(M, m)$-trainability condition (Lemma 5.6), i.e., $\sigma_w^2 = 2$ for vanilla networks and $\sigma_w^2 = 0.1$ or $\sigma_w^2 = 0.01$ for residual networks. The initial bias variance was set to $\sigma_b^2 = 0.01$. We employed MNIST [26] and Fashion-MNIST [99] as the training dataset. All experiments are conducted on an NVIDIA A100.

### L.2   Verification of Thm 4.1 (new mean field-based framework)

As shown in Fig. 1, the empirical distribution of $\boldsymbol{J}(\boldsymbol{x}^{\text{in}})$ in the vanilla network aligned well with Thm 4.1. To further verify Thm 4.1, we provide additional results. We sampled $10,000$ vanilla networks with $d = 1,000$ and $K = 1$. For vanilla networks, we set $\sigma_w^2 = 2$, and for residual networks, we set $\sigma_w^2 = 0.01$. The network width $N$ and depth $L$ were set to $5,000$ and $10$, respectively.

The empirical distributions of $\boldsymbol{J}(\boldsymbol{x}^{\text{in}})$ and $\boldsymbol{a}(\boldsymbol{x}^{\text{in}})$ for different network depths, $L = 10$ and $100$, and two randomly generated inputs, $\boldsymbol{x}_1^{\text{in}}$ and $\boldsymbol{x}_2^{\text{in}}$, are shown in Figs. A5 and A6. The accuracy of Thm 4.1 in predicting these distributions remains consistent across varying network depths. Moreover, the distributions of $\boldsymbol{J}(\boldsymbol{x}^{\text{in}})$ and $\boldsymbol{a}(\boldsymbol{x}^{\text{in}})$ did not depend on the input $\boldsymbol{x}^{\text{in}}$, even though they are defined as functions of $\boldsymbol{x}^{\text{in}}$.

We should note that our theory operates under the assumption of infinite network width. To assess the influence of network width on Thm 4.1, please refer to Fig. A7. We found that for smaller values of $N$, for instance $N = 10$, the empirical distribution of $\boldsymbol{J}(\boldsymbol{x}^{\text{in}})$ and $\boldsymbol{a}(\boldsymbol{x}^{\text{in}})$ deviates from the theoretical prediction as provided by Thm 4.1. As network width increases, the alignment between empirical results and predictions from Thm 4.1 improves.

It is important to clarify that while Thm 4.1 asserts that the distributions of $\boldsymbol{J}(\boldsymbol{x}^{\text{in}})$ and $\boldsymbol{a}(\boldsymbol{x}^{\text{in}})$ do not depend on $\boldsymbol{x}^{\text{in}}$, it does not necessarily imply that random variables $\boldsymbol{J}(\boldsymbol{x}^{\text{in}})$ and $\boldsymbol{a}(\boldsymbol{x}^{\text{in}})$ are independent of $\boldsymbol{J}(\boldsymbol{y}^{\text{in}})$ and $\boldsymbol{a}(\boldsymbol{y}^{\text{in}})$, respectively, when $\boldsymbol{x}^{\text{in}} \neq \boldsymbol{y}^{\text{in}}$. For slightly different inputs ($\boldsymbol{y}^{\text{in}} := \boldsymbol{x}^{\text{in}} \times 0.999$, $\boldsymbol{x}^{\text{in}} \times 0.99$, and $\boldsymbol{x}^{\text{in}} \times 0.5$), we computed the correlation coefficients between $\boldsymbol{J}(\boldsymbol{x}^{\text{in}})$, $\boldsymbol{a}(\boldsymbol{x}^{\text{in}})$ and $\boldsymbol{J}(\boldsymbol{y}^{\text{in}})$, $\boldsymbol{a}(\boldsymbol{y}^{\text{in}})$, respectively.[4] These findings can be found in Figs. A8 and A9, and clearly demonstrate that more similar inputs result in higher correlation coefficients. However, this does not contravene the claims made in Thm 4.1. Further, for two randomly generated inputs, $\boldsymbol{x}_1^{\text{in}}$ and $\boldsymbol{x}_2^{\text{in}}$, the random variables $\boldsymbol{J}(\boldsymbol{x}^{\text{in}})$, $\boldsymbol{a}(\boldsymbol{x}^{\text{in}})$ and $\boldsymbol{J}(\boldsymbol{x}_2^{\text{in}})$, $\boldsymbol{a}(\boldsymbol{x}_2^{\text{in}})$ were found to be relatively uncorrelated, respectively. The theoretical prediction for this correlation is currently unclear, which is a topic for future work.

To validate the independence of different entries of $\boldsymbol{J}(\boldsymbol{x}^{\text{in}})$ and $\boldsymbol{a}(\boldsymbol{x}^{\text{in}})$, we computed the correlation coefficient between two distinct entries.[4] As presented in Fig. A10, we found that the two unique entries of $\boldsymbol{J}(\boldsymbol{x}^{\text{in}})$ and $\boldsymbol{a}(\boldsymbol{x}^{\text{in}})$ were indeed uncorrelated, corroborating the claims in Thm 4.1. Furthermore, Fig. A10 illustrates that the entries of $\boldsymbol{J}(\boldsymbol{x}^{\text{in}})$ and $\boldsymbol{a}(\boldsymbol{x}^{\text{in}})$ were uncorrelated, which also aligned with Thm 4.1.

An examination of the distributions of $\boldsymbol{J}(\boldsymbol{x}^{\text{in}})$ and $\boldsymbol{a}(\boldsymbol{x}^{\text{in}})$ after training may yield interesting insights. However, as trained networks have different weight and bias variances, the observed $\boldsymbol{J}(\boldsymbol{x}^{\text{in}})$ and

---

[4]While the correlation coefficient is not necessarily indicative of dependence, we regard it as a conveniently measurable value. For more information, please refer to Remark E.3.

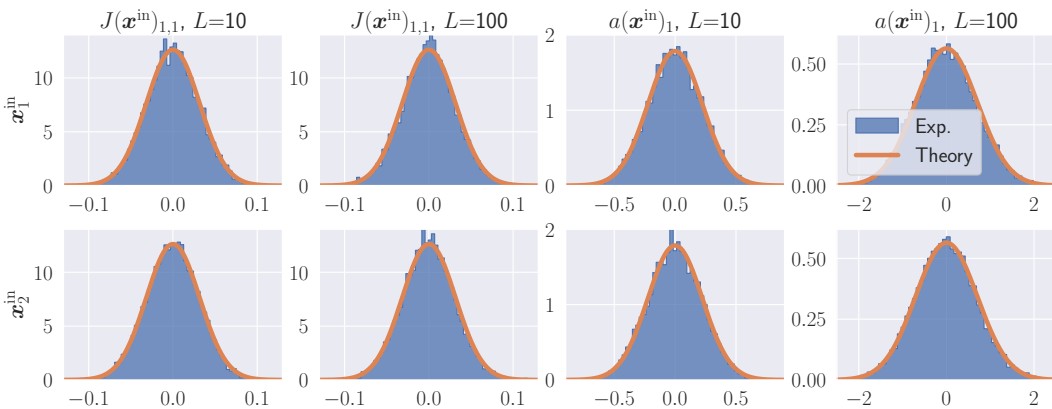

Figure A5: Distributions of $J(\boldsymbol{x}^{\text{in}})_{1,1}$ and $a(\boldsymbol{x}^{\text{in}})_1$ in the vanilla ReLU network with $d = 1,000$, $K = 1$, $N = 5,000$, $\sigma_w^2 = 2$, and $\sigma_b^2 = 0.01$. The description is the same as Fig. 1.

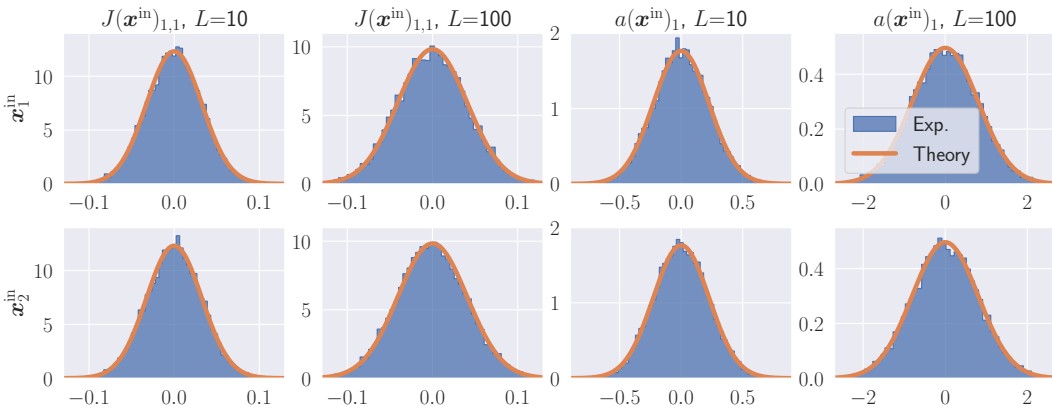

Figure A6: Distributions of $J(\boldsymbol{x}^{\text{in}})_{1,1}$ and $a(\boldsymbol{x}^{\text{in}})_1$ in the residual ReLU network with $d = 1,000$, $K = 1$, $N = 5,000$, $\sigma_w^2 = 0.01$, and $\sigma_b^2 = 0.01$. The description is the same as Fig. 1.

$\boldsymbol{a}(\boldsymbol{x}^{\text{in}})$ in each network do not follow the same distribution. Consequently, empirically evaluating the validity of Thm 4.1 for trained networks presents substantial challenges. Despite this, we consider, based on experimental findings such as those presented in Fig. A18, that Thm 4.1 offers an accurate representation of the early stages of the training process.

### L.3 Verification of Thm 5.1 (upper bounds of adversarial loss)

As shown in Fig. 2, the upper bounds in Thm 5.1 indicates the significant tightness in vanilla networks. To further verify Thm 5.1, we provide additional results. We generated 100 adversarial examples and calculated the adversarial losses defined in Eq. (3). For vanilla networks, we set $\sigma_w^2 = 2$, and for residual networks, we set $\sigma_w^2 = 0.01$. The network width $N$ was set to $40,000$ for vanilla networks and $35,000$ for residual networks, and the network depth $L$ was set to 3. We set the perturbation constraint $\epsilon$ to 0.1 and iterations of projected gradient descent to 50.

In Fig. 2, we illustrate the tightness of the bounds with varying input dimensions in vanilla networks. We provide further results for residual networks, as shown in Fig. A11. Additionally, Figs. A12 and A13 demonstrate the adversarial loss as a function of varying output dimensions in vanilla and residual networks, respectively. Overall, our findings affirm the tightness of our bounds across both network types. For some $(p, q)$ combinations, empirically observed adversarial loss samples exceeded the theoretical upper bounds. We consider that this discrepancy can be mitigated by expanding the network width, a topic elaborated upon in the following paragraph.

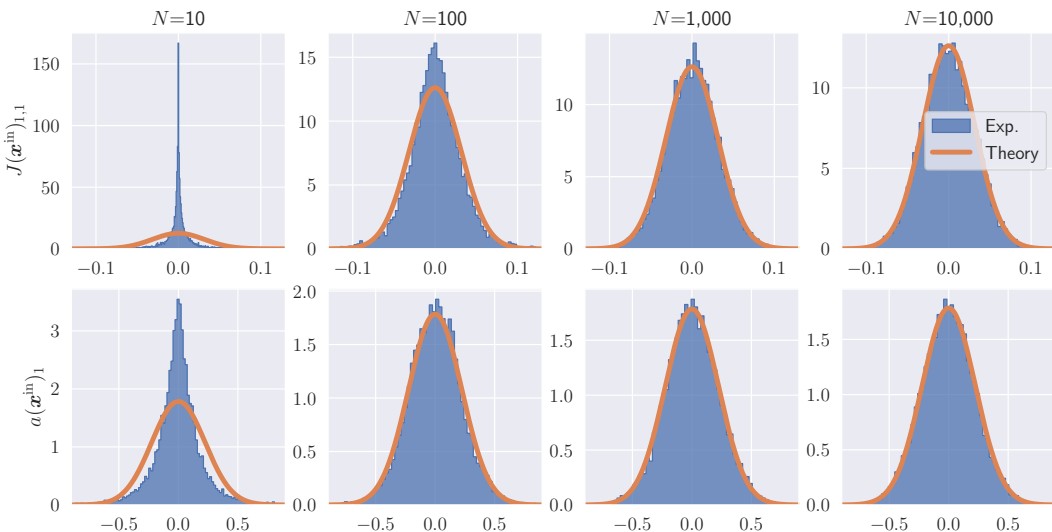

Figure A7: Distributions of $J(\boldsymbol{x}^{\text{in}})_{1,1}$ and $a(\boldsymbol{x}^{\text{in}})_1$ in the vanilla ReLU network with $d = 1,000$, $K = 1$, $L = 10$, $\sigma_w^2 = 0.01$, and $\sigma_b^2 = 0.01$. The description is the same as Fig. 1.

The derivation of Thm 5.1 assumes a network of infinite width. However, in practical implementation, an infinite network width is unachievable, leading to instances where empirical results do not coincide with Thm 5.1. Specifically, certain samples were observed to exceed the theoretical upper bounds, as shown in Fig. 2. To evaluate the influence of network width, we varied it while sampling the adversarial losses, with the results displayed in Fig. A14. In the case of $(p, q) = (2, 2)$, it was observed that a wider network width diminished the discrepancy between sampled losses and theoretical bounds. For other $(p, q)$ pairings, even with a relatively small width (e.g., $N = 100$), empirical results aligned with Thm 5.1.

The impact of the perturbation constraint $\epsilon$ can be found in Fig. A15. It was confirmed that larger $\epsilon$ values corresponded to a greater divergence between empirically sampled adversarial losses and theoretical bounds. This is because for larger $\epsilon$, it was harder for the projected gradient descent to tune adversarial examples well.

The effect of the network depth $L$ can be confirmed in Fig. A16. For $(p, q) = (2, 2)$, as the number of layers increased, the value exceeded the upper bounds. In contrast, for $(p, q) = (\infty, \infty)$, the value fell below the bounds. These differences can be attributed to the varying complexities of optimizing the adversarial loss for each combination of $p$ and $q$. That is, for $(p, q) = (2, 2)$, the optimization of adversarial examples is relatively straightforward, enabling the generation of adversarial losses that exceed the upper bound, which is imposed by the constraints of finite width. However, for $(p, q) = (\infty, \infty)$, the optimization is challenging, and it is difficult to generate high-quality adversarial examples. The underlying reasons for these disparities in optimization complexity remain a topic for future work.

Furthermore, we assessed adversarial loss during training on the MNIST dataset, as shown in Fig. A17. Vanilla networks were trained using stochastic gradient descent with a learning rate of $0.001$. While Thm 5.1 broadly holds for $(p, q) = (1, 2)$, the disparity between theoretical prediction and empirical results expands as training progresses for $(p, q) = (2, 2)$, $(2, \infty)$, and $(p, q) = (\infty, \infty)$. Determining a theoretical prediction that provides a tight bound on the adversarial loss for fully trained deep neural networks remains an open challenge for future research.

### L.4   Verification of Thms 5.4 and G.9 (time evolution of weight variance)

As shown in Fig. 3, the empirical weight variance observed in the vanilla network aligns well with Thm 5.4. To verify Thm G.9, we provide Fig. A18. We trained 10-layers vanilla and residual networks. The vanilla networks were initialized with $\sigma_w^2(0) = 2$ and $\sigma_b^2(0) = 0.01$. Residual networks are initialized with $\sigma_w^2(0) = 0.1$ and $\sigma_b^2(0) = 0.01$. For standard training, $N$ was set to $1,000$. For

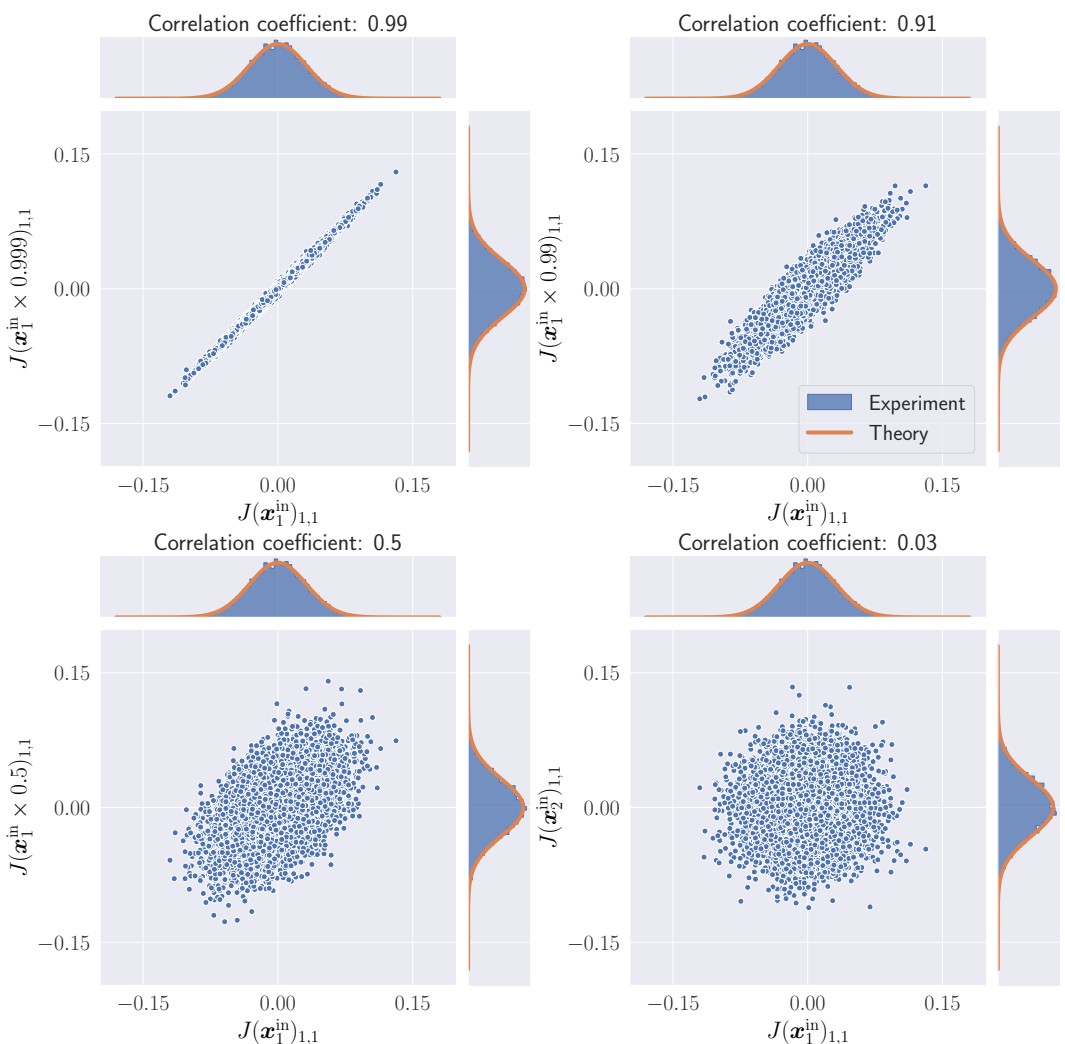

Figure A8: Correlation coefficient of $J(\cdot)_{1,1}$ for two inputs in the vanilla ReLU network with $d = 1,000$, $K = 1$, $N = 5,000$, $L = 10$, $\sigma_w^2 = 2$, and $\sigma_b^2 = 0.01$. The description of the histogram is the same as Fig. 1.

adversarial training, we designated $p = \infty$, $q = \infty$, and $\epsilon = 0.3$. In both training, we used stochastic gradient descent with a small learning rate, $0.001$. Note that gradient flow assumes an infinitesimal learning rate (cf. Eq. (10)). A theoretically defined training step $t$ under gradient flow is not equal to a training step in the experiment. We have linked them by $t := \text{training steps} \times \text{learning rate}$. In practice, setting the weight variance precisely is not feasible. For example, in a vanilla network, $\sigma_w^2(0)$ might be set to $1.999$ even though we tried to initialize it to satisfy $\sigma_w^2(0) = 2$. Thus, for visibility, the experimental results (curves) were shifted parallel to meet $\sigma_w^2(0) = 2$ in vanilla networks and $\sigma_w^2(0) = 0.1$ in residual networks.

From Fig. A18, it is evident that adversarial training facilitates weight regularization for both vanilla and residual networks. Compared to the $\ell_2$ weight regularization discussed in Appx. I, it offers stronger regularization. We can verify that Thm 5.1 can predict these empirical behaviors in the initial stage of training. The validity of Thms 5.4 and G.9, which underpin our theorems such as Thms 5.7 and 5.9, lends credence to our theoretical assertions.

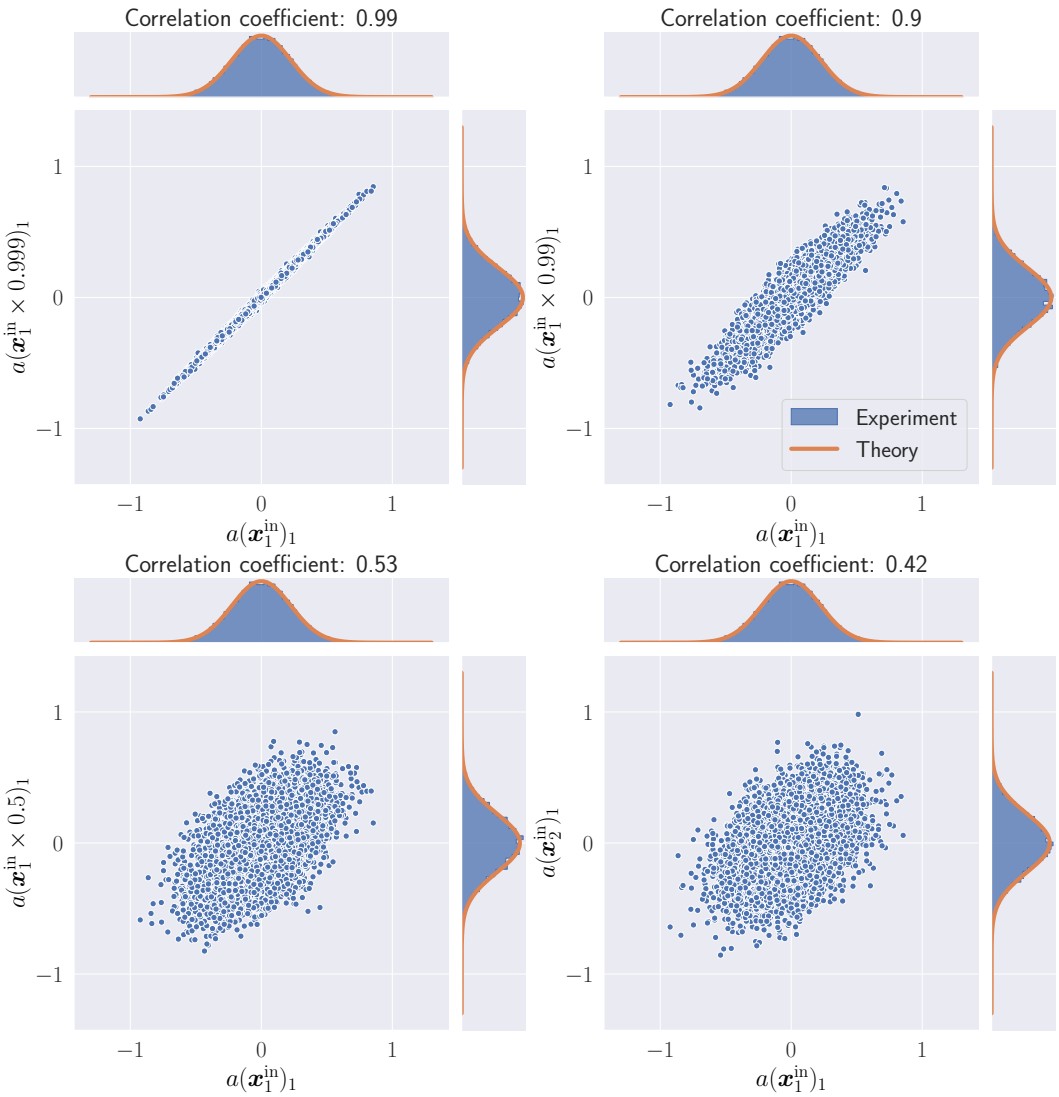

Figure A9: Correlation coefficient of $a(\cdot)_1$ for two inputs in the vanilla ReLU network with $d = 1,000$, $K = 1$, $N = 5,000$, $L = 10$, $\sigma_w^2 = 2$, and $\sigma_b^2 = 0.01$. The description of the histogram is the same as Fig. 1.

## L.5 Verification of Thms 5.7 and 5.8 (adversarial trainability)

As shown in Fig. 4, vanilla networks with small width and large depth were not adversarially trainable, which aligned well with Thm 5.7. To verify Thms 5.7 and 5.8, we provide Fig. A19. We trained vanilla and residual networks under various width and depth settings, and observed training accuracy. For fast training convergence, we used Adam [51]. The training was stopped if training accuracy was not improved in the last 200 steps. We set the learning rates to 0.001 or 0.0001, adopting the highest training accuracy at the final training step. The vanilla networks were initialized with $\sigma_w^2(0) = 2$ and $\sigma_b^2(0) = 0.01$. The residual networks were initialized with $\sigma_w^2(0) = 0.01$ and $\sigma_b^2(0) = 0.01$. This initialization satisfied the $(M, m)$-trainability conditions (Lemma 5.6). In adversarial training, we set $p = \infty$, $q = \infty$, $\epsilon = 0.1$, and the iterations of projected gradient descent to 10.

From Fig. A19, we can confirm that vanilla networks with large depth and small width were not adversarially trainable and the training difficulty was unique to adversarial training of such vanilla networks. Note that Prop G.10 could not be verified as the condition in which training persists without machine errors and the trainability condition not being satisfied at $t = 0$ is challenging to implement in practical scenarios.

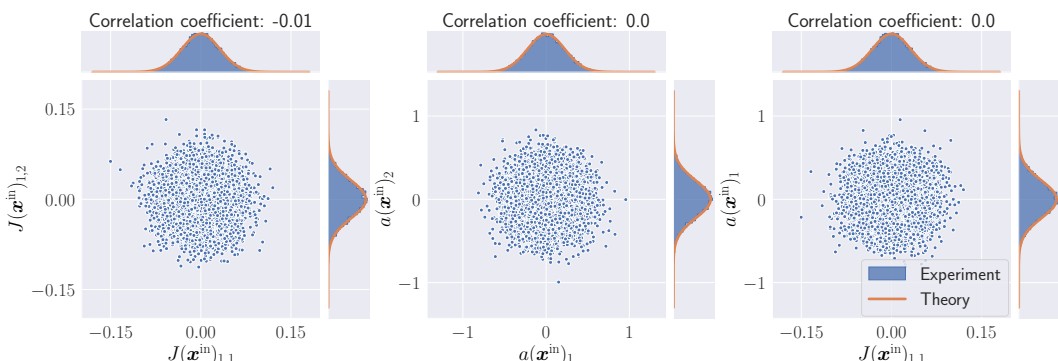

Figure A10: Correlation coefficient of two different entries of $\boldsymbol{J}(\boldsymbol{x}^{\text{in}})$ in the vanilla ReLU network with $d = 1,000$, $K = 1$, $N = 5,000$, $L = 10$, $\sigma_w^2 = 2$, and $\sigma_b^2 = 0.01$. The description of the histogram is the same as Fig. 1.

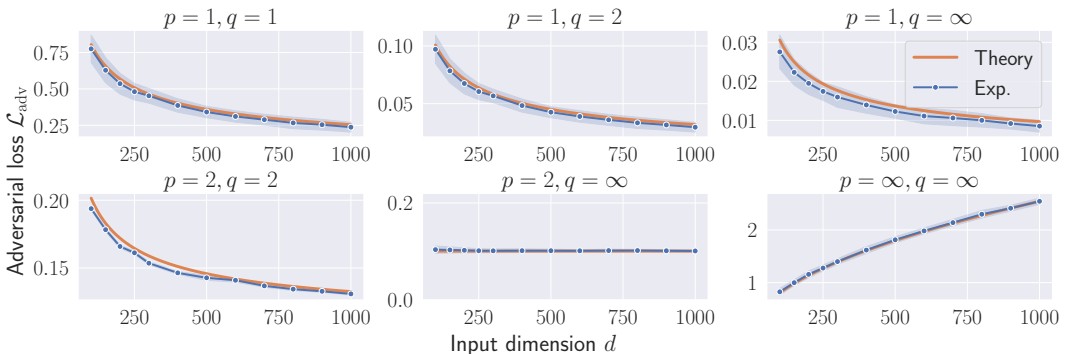

Figure A11: Adversarial loss (Eq. (3)) in residual networks with $N = 35,000$, $K = 100$, $L = 3$, $\sigma_w^2 = 0.01$, $\sigma_b^2 = 0.01$, and $\epsilon = 0.1$. The description is the same as Fig. 2.

### L.6 Verification of Thms 5.9 and G.14 (capacity degradation)

To verify the degradation of network capacity during adversarial training, we trained vanilla and residual networks on MNIST with $p = \infty$, $q = \infty$, and $\epsilon = 0.1$, and observed the Fisher–Rao norm as a measure of capacity. The training steps were set to 500. As Thms 5.9 and G.14 are derived under sufficiently small learning rate (cf. Eq. (10)), we employed small learning rate, 0.0001. Vanilla networks were initialized with $\sigma_w^2(0) = 2$ and $\sigma_b^2(0) = 0.01$. Residual networks were initialized with $\sigma_w^2(0) = 0.01$ and $\sigma_b^2(0) = 0.01$. During the derivation of Thms 5.9 and G.14, using Asm B.1, we calculated $\mathbb{E}[\frac{\partial f}{\partial w_i} \frac{\partial f}{\partial w_j} w_i w_j] = \mathbb{E}[\frac{\partial f}{\partial w_i} \frac{\partial f}{\partial w_j}]\mathbb{E}[w_i w_j] = 0$, for $i \neq j$. Nevertheless, in practice, this calculation does not always hold. Therefore, for the computation of the Fisher–Rao norm, we employed a diagonal matrix of $\boldsymbol{F}$ instead of $\boldsymbol{F}$ itself.

As shown in Figs. A20 and A21, large network depths (large $L$) produced high Fisher–Rao norms but degraded them drastically. Moreover, it was observed that a wider network width could maintain the Fisher–Rao norm at its initial state. The discrepancy between these experimental values and the values predicted by Thms 5.9 and G.14 is considered to originate from Asm B.1, which does not hold in the case where a diagonal matrix of $\boldsymbol{F}$ is used instead of $\boldsymbol{F}$ itself.

Additionally, we assessed the influence of network width on the robust test accuracy of fully-trained networks. The robust test accuracy was determined using $\ell_\infty$-AutoAttack with $\epsilon = 0.1$. The adversarial training was conducted under $p = \infty$, $q = \infty$, $\epsilon = 0.1$, and 10 iterations. We employed Adam with a learning rate of 0.001 and epochs set to 200.

From Tabs. A5 and A6, it is evident that the robust test accuracy depends on network width rather than depth. The same superscripts in the tables indicate results from networks with an identical number of parameters. For example, the accuracy for $(N, L) = (250, 5)$ significantly surpasses that

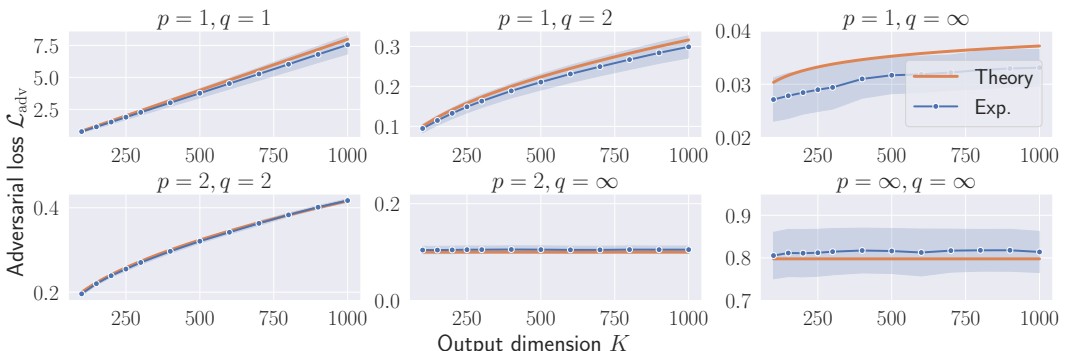

Figure A12: Adversarial loss (Eq. (3)) in vanilla networks with $d = 100$, $N = 40,000$, $L = 3$, $\sigma_w^2 = 2$, $\sigma_b^2 = 0.01$, and $\epsilon = 0.1$. The description is the same as Fig. 2.

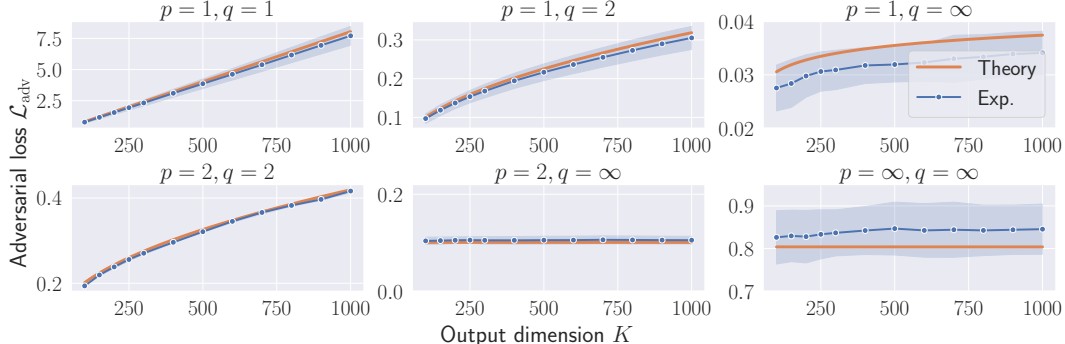

Figure A13: Adversarial loss (Eq. (3)) in residual networks with $d = 100$, $N = 35,000$, $L = 3$, $\sigma_w^2 = 0.01$, $\sigma_b^2 = 0.01$, and $\epsilon = 0.1$. The description is the same as Fig. 2.

for $(N, L) = (125, 20)$. Although Thms 5.9 and G.14 is primarily applicable to the early stages of training, these theorems, emphasizing the importance of network width, hold true for fully-trained networks.

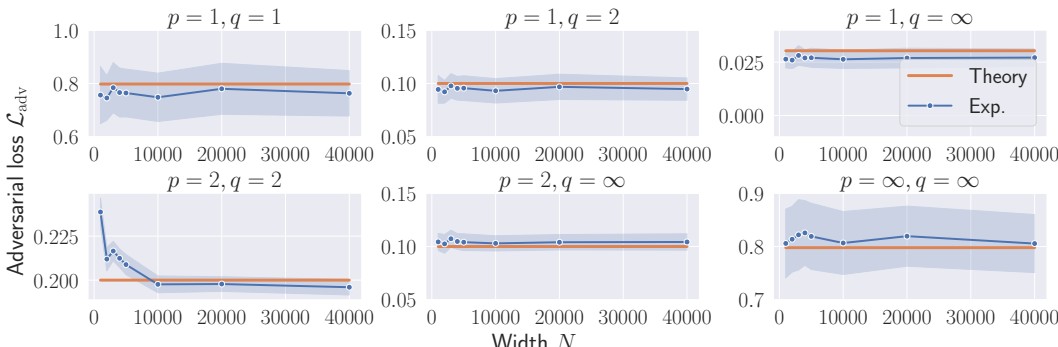

Figure A14: Adversarial loss (Eq. (3)) in vanilla networks with $d = 500$, $K = 100$, $L = 3$, $\sigma_w^2 = 2$, $\sigma_b^2 = 0.01$, $p = 2$, $q = 2$, and $\epsilon = 0.1$. The description is similar to Fig. 2.

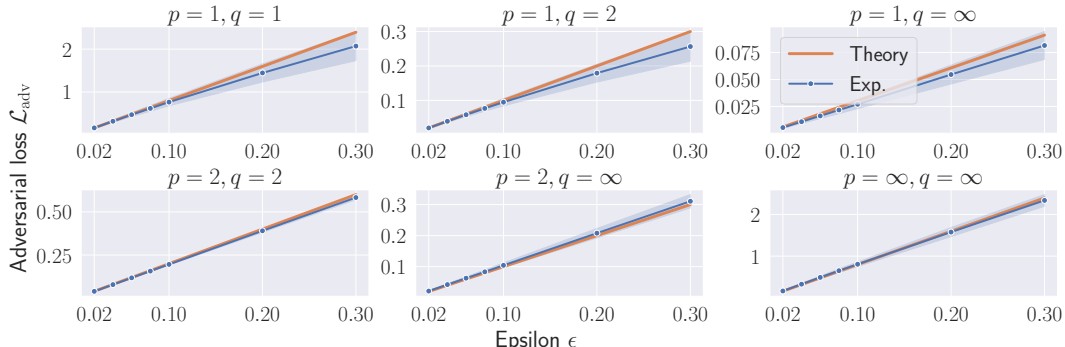

Figure A15: Adversarial loss (Eq. (3)) in vanilla networks with $d = 500$, $N = 40,000$, $K = 100$, $L = 3$, $\sigma_w^2 = 2$, $\sigma_b^2 = 0.01$, $p = 2$, and $q = 2$. The description is similar to Fig. 2.

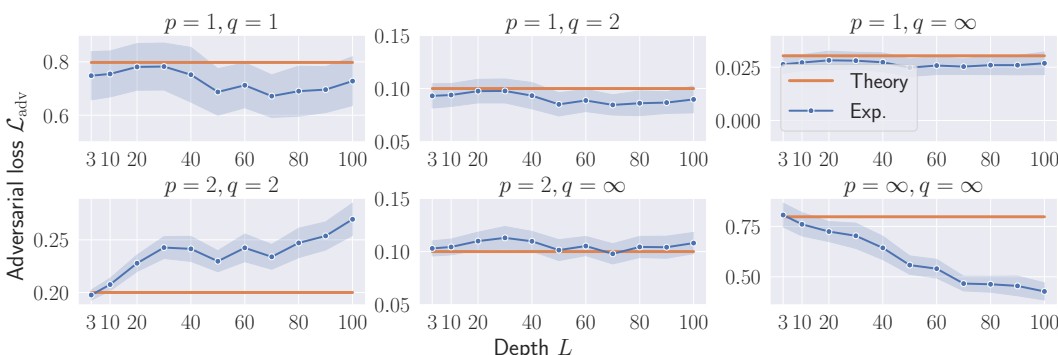

Figure A16: Adversarial loss (Eq. (3)) in vanilla networks with $d = 500$, $N = 40,000$, $K = 100$, $\sigma_w^2 = 2$, $\sigma_b^2 = 0.01$, $p = 2$, $q = 2$, and $\epsilon = 0.1$. The description is similar to Fig. 2.

Table A5: Test accuracy (%) on MNIST [26] under $\ell_\infty$-AutoAttack with the perturbation constraint 0.1. We used residual networks with $\sigma_w^2 = 0.01$ and $\sigma_b^2 = 0.01$. For adversarial attack in training, we used $p = \infty$, $q = \infty$, $\epsilon = 0.1$, and 10 iterations. We set an optimizer to Adam, a learning rate to 0.001, and epochs to 200. Values marked with the same superscript denote results from networks with the same number of parameters $LN^2$.

|  | $L = 5$ | $L = 10$ | $L = 15$ | $L = 20$ |
|---|---|---|---|---|
| $N = 125$ | 13.8 | 13.1 | 9.79 | 8.69♣ |
| $N = 250$ | 45.8♣ | 49.8 | 47.0 | 48.1† |
| $N = 500$ | 79.2† | 76.5 | 75.3 | 73.4◇ |
| $N = 1,000$ | 89.8◇ | 88.4 | 87.3 | 87.9 |

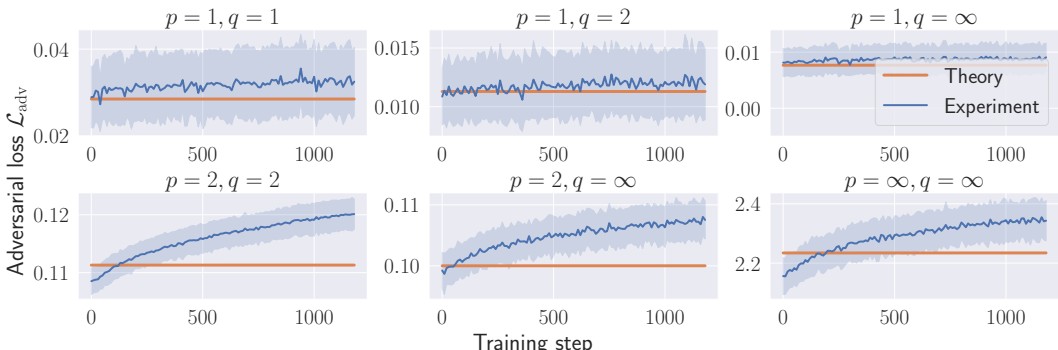

Figure A17: Adversarial loss (Eq. (3)) calculated with $\epsilon = 0.1$. We used vanilla networks with $d = 28 \times 28$, $N = 10,000$, $K = 10$, $L = 3$, $\sigma_w^2(0) = 2$, and $\sigma_b^2(0) = 0.01$. We trained the vanilla networks normally (not adversarially) with the learning rate $0.001$. The description is similar to Fig. 2.

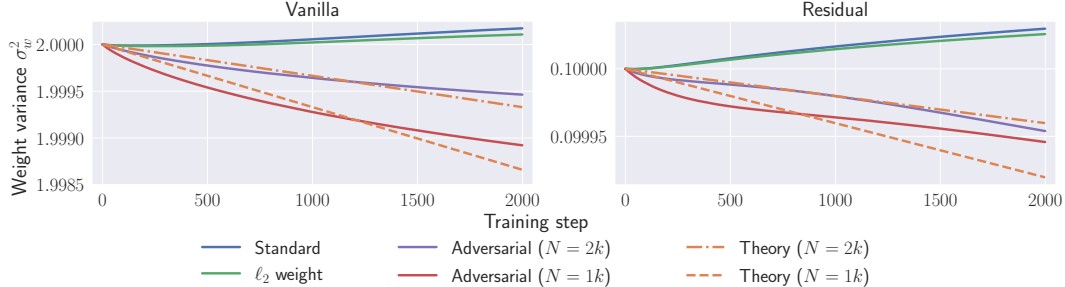

Figure A18: Time evolution of the weight variance in the vanilla and residual network with $L = 10$ during adversarial training with $p = \infty$, $q = \infty$, and $\epsilon = 0.3$. In standard and adversarial training, we set the learning rate to $0.0001$. In standard training with or without $\ell_2$ weight regularization, we set $N = 1,000$. In adversarial training, we set $p = \infty$, $q = \infty$, and $\epsilon = 0.3$. The vanilla network was initialized with $\sigma_w^2(0) = 2$ and $\sigma_b^2(0) = 0.01$. The residual network was initialized with $\sigma_w^2(0) = 0.1$ and $\sigma_b^2(0) = 0.01$. The orange dashed lines are predicted by Thms 5.4 and G.9. To derive the theoretical predictions, we calculated $t := \text{training steps} \times \text{learning rate}$. For visibility, the experimental results were shifted parallel to satisfy $\sigma_w^2(0) = 2$ on the vanilla networks and $\sigma_w^2(0) = 0.1$ on the residual networks.

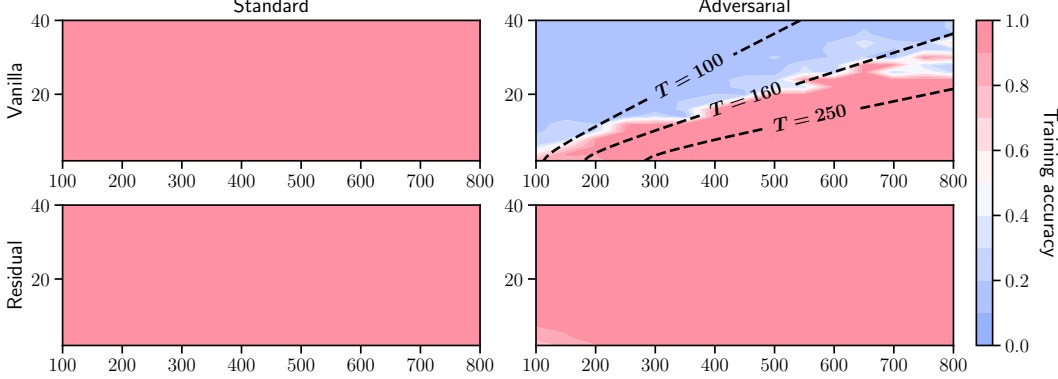

Figure A19: Heat map of the training accuracy on the vanilla and residual networks trained on MNIST. The description is the same as Fig. 4. The vanilla network was initialized with $\sigma_w^2(0) = 2$ and $\sigma_b^2(0) = 0.01$. The residual network was initialized with $\sigma_w^2(0) = 0.1$ and $\sigma_b^2(0) = 0.01$. The horizontal axis represents the width of the network and the vertical axis represents the depth.

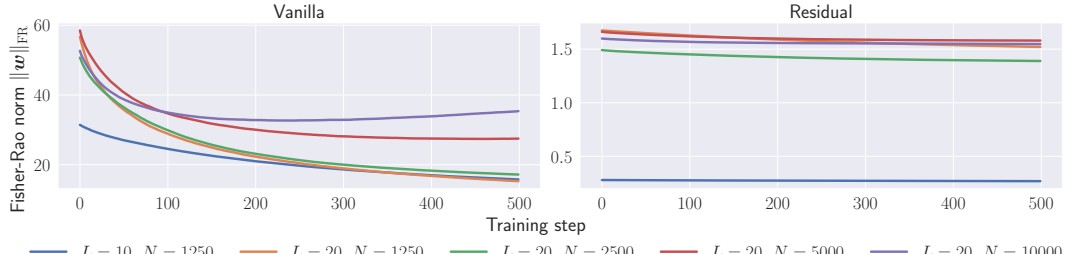

Figure A20: Fisher–Rao norm of vanilla and residual networks adversarially trained on MNIST. We set $p = \infty$, $q = \infty$, $\epsilon = 0.1$, and the learning rate to 0.0001. The vanilla network was initialized with $\sigma_w^2(0) = 2$ and $\sigma_b^2(0) = 0.01$. The residual network was initialized with $\sigma_w^2(0) = 0.1$ and $\sigma_b^2(0) = 0.01$.

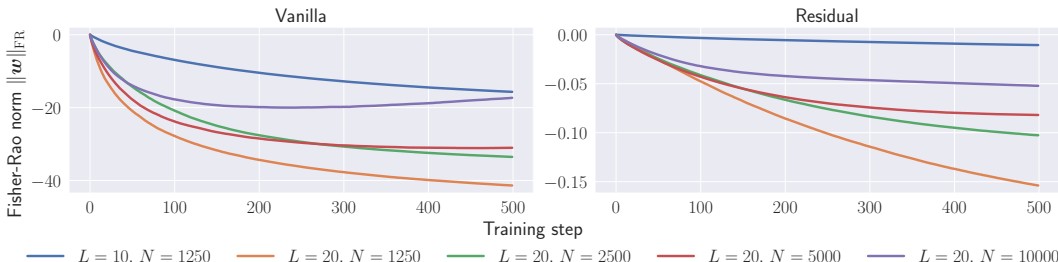

Figure A21: Fig. A20 with the origin shifted parallel to zero.

Table A6: Test accuracy (%) on Fashion-MNIST [99] under $\ell_\infty$-AutoAttack. The description is the same as Tab. A5.

|           | $L = 5$ | $L = 10$ | $L = 15$ | $L = 20$ |
|-----------|---------|----------|----------|----------|
| $N = 125$ | 7.78 | 9.22 | 10.4 | 11.0♣ |
| $N = 250$ | 33.1♣ | 33.6 | 32.6 | 34.0† |
| $N = 500$ | 51.7† | 51.7 | 51.6 | 50.8◇ |
| $N = 1,000$ | 67.9◇ | 67.3 | 66.7 | 66.8 |

