# OpenReview forum: "Adversarial Training from Mean Field Perspective"
_NeurIPS.cc/2023/Conference — NeurIPS 2023 spotlight_

### Official Review · Reviewer_QtQE · 2023-07-05

**Soundness:** 4 excellent
**Presentation:** 4 excellent
**Contribution:** 4 excellent
**Rating:** 8
**Confidence:** 4

**Summary:**

The authors proposed a new theoretical framework based on the mean field theory to analyse adversarial training from several perspectives, such as the upper bounds of adversarial loss, the time evolution of weight variance and adversarially trainable conditions. Besides the theoretical analysis, the authors conducted several experiments verifying the proposed theoretical framework.
Generally speaking, the proposed theoretical framework provides a new perspective to analyse adversarial training and is highly versatile and even can extend to other training methods.


**Strengths:**

The paper is organised well and easy to follow.
The proposed theoretical framework seems inspiring and intriguing and gives a new perspective to analyse adversarial training from several aspects, which may serve as a good guidance for future work. Besides the proposed theory, verification experiments were also conducted to prove its effectiveness further.



**Weaknesses:**

1.	The verification experiments were only conducted on the easy dataset (MNIST); it may strengthen the findings if additional experiments are conducted on more challenging datasets.
2.	 Including a more systematic evaluation of the results may be beneficial, such as adversarial loss in residual networks vs. training steps for normally or adversarially training.
3.	Several conclusions match the previous works; it could be more convincing if the reference could be given in the main content, such as ‘’the square sum of the weights in Ineq. (9) suggests that adversarial training exhibits a weight regularisation effect,’’ is consistent with [1], ‘’to achieve high capacity in adversarial training, it is necessary to increase not only the number of layers L but also the width N to keep L^2/N constant’’ is somehow consistent with [2].
4.	The authors mentioned in Line 260 that ‘’This result suggests that residual networks are better suited for adversarial training. However, one of the previous studies indicated that residual networks are more vulnerable to transfer attacks [54] than vanilla networks. ’’ However, the proposed theoretical framework did not explain such a transfer attack phenomenon. Could the authors explain or give more comments about this?
5. in Equation 3 and 4, it seems that the minimize part is not shown, perhaps it would be more reasonable to change the form of min-max in equation 3 and 4.

[1] A unified gradient regularization family for adversarial examples, in: IEEE International Conference on Data Mining (ICDM), 2015.
[2] Do wider neural networks really help adversarial robustness? Advances in Neural Information Processing Systems, 34.


**Questions:**

See Weakness above.



**Limitations:**

The authors have clearly discussed the limitations of the proposed framework, such as, ‘’some theorems begin to diverge from the actual behaviour‘’ and ’’ the mean field theory assumes infinite network width, which is practically infeasible’’.
From my point of view, this article does not involve any potential negative social impact.

---

> ### Author Rebuttal · Authors · 2023-08-08
>
> We would like to thank your careful reading.
>
> > The verification experiments were only conducted on the easy dataset (MNIST); it may strengthen the findings if additional experiments are conducted on more challenging datasets.
>
> We have only tested our theory on simple datasets. This is because fully connected networks struggle to achieve high training accuracy and generalization performance on more complex datasets, such as CIFAR-10 and ImageNet, even in standard training. This complicates a fair comparison between standard and adversarial training. To perform both theoretical and experimental analysis of adversarial training on complex datasets, our theoretical results would need to be extended to more practical architectures, such as convolutional networks. Note that Table A8 shows results for Fashion-MNIST that are consistent with our theoretical findings. We also highlight that using MNIST for verification is a common practice in several theoretical approaches to adversarial defense, such as certified adversarial defense [3,4].
>
> [3] Certified defenses against adversarial examples, ICLR18.
> [4] Semidefinite relaxations for certifying robustness to adversarial examples, NeurIPS18.
>
> > Including a more systematic evaluation of the results may be beneficial, such as adversarial loss in residual networks vs. training steps for normally or adversarially training.
>
> Our extensive experiments cover each of our theoretical claims for vanilla and residual networks. Some results were sent to Appendix due to the page limitation. However, thanks to the reviewer’s comments, we found that some results for residual networks were not included, such as how the adversarial loss in residual networks changes during adversarial training. We will include these results in the updated version, along with more visually clear illustrations and comparisons.
>
> > Several conclusions match the previous works; it could be more convincing if the reference could be given in the main content, such as "the square sum of the weights in Ineq. (9) suggests that adversarial training exhibits a weight regularisation effect," is consistent with [1], "to achieve high capacity in adversarial training, it is necessary to increase not only the number of layers $L$ but also the width $N$ to keep $L^2/N$ constant" is somehow consistent with [2].
>
> We appreciate the reviewer’s introduction to related work and will include comparisons with these studies in the main text.
>
> [1] suggested that adversarial training with the fast gradient sign method regularizes network weights around training instances, which aligns with Theorems 5.3 and F.9. However, their research relies on a strong assumption about the network's Jacobian, which does not strictly hold for realistic data distributions. Our study does not require such an assumption. Furthermore, we have provided explicit details regarding the time evolution of weight variance and the impacts of network width, depth, and other structures that were not provided in [1].
>
> From a perturbation instability perspective, [2] demonstrated that increasing network width does not necessarily improve robustness. This may seem to contradict our result, which suggests that a wider network can help the model maintain capacity during adversarial training, implying greater robustness in wider networks. However, these two claims are compatible. Robustness is determined by both perturbation instability (negative effect) and network capacity (positive effect). While the negative effect of width appears dominant in [2]'s experiments on CIFAR-10 and WideResNet, the positive effect appeared more prevalent in our experiments on MNIST, Fashion-MNIST, and fully connected networks with or without shortcuts. The dominant factor may depend on the dataset and model architectures.
>
> > The authors mentioned in Line 260 that "This result suggests that residual networks are better suited for adversarial training. However, one of the previous studies indicated that residual networks are more vulnerable to transfer attacks [54] than vanilla networks." However, the proposed theoretical framework did not explain such a transfer attack phenomenon. Could the authors explain or give more comments about this?
>
> In practice, adversarial training of residual networks is more common than that of vanilla networks. We mentioned transfer attacks as a motivation for analyzing adversarial training of vanilla networks. However, as several reviewers have pointed out, our study does not address transfer attacks. To avoid confusion, we have decided to remove this section in the updated version.
>
> > in Equation 3 and 4, it seems that the minimize part is not shown, perhaps it would be more reasonable to change the form of min-max in equation 3 and 4.
>
> Equations 3 and 4 do not include minimization because they define adversarial loss. The minimization of this loss with the standard loss defines adversarial training, which is described below Equations 3 and 4 (perhaps we should highlight this point more clearly). Many of our results and proofs refer to the adversarial loss, and we would like to maintain its current form without minimization.

---

> > ### Comment · Reviewer_QtQE · 2023-08-15
> > **Keep my previous rating**
> >
> > I appreciate the responses given by the authors which resolve some of my concerns. I would like to keep my previous positive rating.

---

### Official Review · Reviewer_CfQ9 · 2023-07-24

**Soundness:** 3 good
**Presentation:** 2 fair
**Contribution:** 3 good
**Rating:** 6
**Confidence:** 3

**Summary:**

The authors propose a mean field based framework for theoretically analyzing the training dynamics of adversarial training for MLP and residual networks with ReLU non-linearities. Based on this framework, the authors provide tight bounds on the adversarial loss (squared loss between clean and adversarial example output) and investigate trainability of MLP and residual networks.

**Strengths:**

- While I am not very familiar with related work on mean field theory for (adversarially) training networks, the proposed framework seems to address limitations of prior work quite elegantly.
- The paper includes quite a number of interesting results using the proposed framework (loss bounds, training dynamics, trainability, impact of width).
- Formulation includes MLP and residual network with a class of “ReLU-like” non-linearities; is also includes adversarial training with clean loss (as e.g. done by TRADES).
- I feel the framework can be quite insightful and helpful for future work in understanding or improving adversarial training.
- Many claims are empirically supported on MNIST.

**Weaknesses:**

My main concern about the paper is that it is incredibly dense (due to the number of included claims) and its structure does not make checking the proofs and claims easy. Even though I invested significantly more time in this review compared to other reviews, I was unable to fully follow all derivations and proofs. I believe this is mainly due to the extremely convoluted way of presenting the theoretical results across main paper and appendix. Unfortunately, I feel that this also makes me less enthusiastic about the results. Here are some more detailed comments and suggestions:
- Starting with 4.2, the reader is somewhat forced to jump to the appendix at least once just to see that the authors simply reformulate the ReLU using its derivative and D for J and a. This trend continues throughout the paper and appendix. Especially in the appendix, one is forced to jump around a lot just to follow 2-3 sentences.
- The proofs are structured starting with simple lemmata and building up to the actual theorems. This is generally fine, but again, the referencing is overdone. For Thm 4.1 in Appendix E, there are 9 pages of derivation with >20 individual lemmata. So if I want to follow the proof of Thm 4.1, I am forced to go through many but not all of these lemmeta. Many have proofs of only 1-2 lines, but reference 2+ other lemmata or remarks and I need to remember the numbers or jump back and forth 2+ times just to read a single sentence. I feel the root cause for this is that many of the results are over-generalized and compartmentalized too much. I appreciate the thorough job of the authors in establishing many of the independence results, but as a reviewer and reader my #1 interest is following Thm 4.1, nothing more and nothing less. Everything that complicates this job is – in my opinion – bad for the paper. For me, the ideal solution would be a separate, easier to follow section for the proof of Thm 4.1 - even if it restates many of the lemmata and remarks, and moving the other results to a separate section for the (very very) interested reader.
- The main paper includes so many results that there is basically no discussion of each individual result. Often it feels that every sentence refers to some additional result in the appendix. I think for me, and many readers, actually discussing the results informally, in words, and taking more time and space to introduce the required notation would be more beneficial than including the current amount of results. I would prefer to have fewer results well-described and the remaining ones being in the appendix.
- Empirical results are discussed twice – after the corresponding theorems as well as in Section 6 – the space of the latter could be used to address one of the points above.

Comments and questions unrelated to structure and writing:
- In the introduction, contribution (d) is unclear to me – what theoretical result does it refer to, the trainability?
- The use of “probabilistic properties” is a bit unclear until the discussion of training dynamics. It would be helpful if the meaning would be detailed earlier in the paper.
- In 3.1, why is having $P^{in}$ and $P^{out}$ important, i.e., why do we need these fixed layers?
- Usually, the adversarial loss is also cross-entropy or something similar. While I saw papers arguing for a squared loss, TRADES does not use it AFAIK. Instead, the common setting is adversarial cross-entropy loss only, combined with clean cross-entropy loss or clean cross-entropy + KL as in TRADES. This makes me ask how assuming an adversarial cross-entropy loss would impact the results? Can similar results be derived?
- In 4, the assumption of independence is also unclear. I feel making this more explicit, e.g., by informally providing a short result on independence from the appendix, could be useful.
- The authors also highlight broader applicability; I am wondering if the authors derived similar results for standard training as reference? Or has this been done in previous work with other frameworks? This also related to the statement in l280.
- In 5.3 l261, I can’t follow how transfer attacks are relevant here? Transfer attacks should be weaker than the general attack modeled in the paper …

Conclusion:
I think that the paper has many interesting contributions and will be valued by the NeurIPS community. However, as I was not able to follow all derivations in a reasonable time, I will closely follow what the other reviewers have to say about the theoretical results. Also, I believe that the paper in its current form would have better fitted a long-format journal. For NeurIPS, I hope the authors invest some time in simplifying the main paper and restructuring the appendix. I think the current format will limit the audience of the paper to those very familiar with related work or willing to invest hours jumping back and forth between main paper and appendix. Some restructuring could really make this paper more accessible to the broad audience at NeurIPS.

**Questions:**

See weaknesses.

**Limitations:**

See weaknesses.

---

> ### Author Rebuttal · Authors · 2023-08-08
>
> ## Structure and writing
> We extend our deepest appreciation to you for the considerable time and effort you have devoted to reviewing our paper. Taking the reviewer’s comments into account, we plan to improve the manuscript as follows.
>
> **Simplified version of Theorem 4.1.**
>
> We plan to present a simplified claim and its proof of Theorem 4.1 for one-hidden-layer vanilla networks. If readers can understand this, we believe that they can more easily follow the claim for multi-layer networks (i.e., Theorem 4.1). This is because its proof relies on mathematical induction with respect to the layers.
>
> In addition, one factor that complicates the proof arises from a well-known but sometimes overlooked fact: uncorrelated Gaussian variables are not necessarily independent. We have prepared many lemmas to discuss this rigorously in our context. Considering the reviewer’s feedback, as a stepping stone, we decided to prove Theorem 4.1 with an incorrect claim (i.e., uncorrelated Gaussian variables are independent) as well as the formal proof.
>
> Note that Proposition A.1 was provided to highlight the essence of Theorem 4.1, i.e., the independence of the distributions of $J$ and $a$ from the input $x$. However, we acknowledge that this has not been sufficiently emphasized.
>
> **Separation of proofs for vanilla and residual networks.**
>
> Currently, some of our lemmas apply only to residual networks, which may not be relevant to readers solely interested in the results for vanilla networks. We intend to separate the formal proofs for vanilla and residual networks.
>
> **Generality and over-compartmentalization of lemmas.**
>
> We plan to reconsider the generality of all claims to improve readability. However, maintaining a rigorous discussion while doing so is not easy. Therefore, we will initially focus on the previously mentioned step-by-step presentations. We acknowledge the over-compartmentalization and try to group lemmas with similar claims (e.g., Lemma E20--23).
>
> **Simplification of main text.**
>
> Although we have struggled with this, we believe that all claims in the main text are essential. We are not considering removing an entire theorem. Instead, we intend to increase intuitive explanations and informal presentations to improve readability. Following the reviewer’s suggestion, we also plan to remove some unnecessary references to experimental results to create space.
>
> ## Contents
> > In the introduction, contribution (d) is unclear to me – what theoretical result does it refer to, trainability?
>
> The content in "Other contributions" can be found in Section J. Contribution (d) corresponds to the paragraph "Mitigation of adversarial risk" in Section J. We will explicitly mention this in the updated version.
>
> > The use of "probabilistic properties" is a bit unclear until the discussion of training dynamics.
>
> In the updated version, we will improve the clarity of this part as follows (e.g., l30):
>
> Before: existing mean field-based approaches cannot manage the probabilistic properties of an entire network ...
>
> After: existing mean field-based approaches cannot manage the probabilistic properties of an entire network (e.g., the distribution of a network Jacobian) ...
>
> > Why do we need $P^{in}$ and $P^{out}$?
>
> They are introduced to discuss networks with general input and output sizes. They also slightly simplify the calculations as it allows us to focus on trainable layers of fixed size $N\times N$. For example, without these layers, $\chi^{(0)}/\chi^{(L)}=\omega^L$ would become $\chi^{(0)}/\chi^{(L)}=N\omega^L/d$. Note that they do not harm training.
>
> > Can similar results be derived for adversarial cross-entropy loss?
>
> Based on the reviewer’s suggestion, we have considered the case with cross-entropy loss, and found that similar results can be obtained. We appreciate the comment and share some of the results. Suppose that the class label of $x$ is one and that the network has no biases. The adversarial loss is defined as:
> $$\mathcal{L}(x):=\max\_{\|\eta\|\_\infty\leq\epsilon}\left(-\ln\frac{\exp(J(x+\eta)\_{1\cdot}(x+\eta))}{\sum^K\_{k=1}\exp(J(x+\eta)\_{k\cdot}(x+\eta))}\right).$$
> For simplicity, we assume that (I) $x=0$. (II) $\epsilon$ is sufficiently small, and thus $J(x+\eta)=J(x)$ (cf. l931). (III) the input dimension is sufficiently large. Then,
> $$\mathcal{L}(0)\leq\ln\left(\sum^K\_{k=1}\exp\left(\max\_{\|\eta\|\_\infty\leq\epsilon}J(0)\_{k\cdot}\eta\right)\right)+\max\_{\|\eta\|\_\infty\leq\epsilon}J(0)\_{1\cdot}\eta.$$
> For any $k\in\\{1,\ldots,K\\}$ (cf. (A105)),
> $$\max\_{\|\eta\|\_\infty\leq\epsilon}J(0)\_{k\cdot}\eta=\mathcal{O}(\omega^{L/2}).$$
> Thus,
> $$\mathcal{L}(0)\leq\mathcal{O}(\omega^{L/2}).$$
> This is consistent with Theorem 5.1, which also gives $\mathcal{O}(\omega^{L/2})$. Thus, we consider that a similar discussion can be applied to Theorem 5.3, as well as those based on it (i.e., Theorems 5.6-5.8).
>
> > In 4, the assumption of independence is unclear.
>
> In this paper, "independence" was used in the probabilistic sense: two random variables are called independent if their joint probability equals the product of their probabilities.
>
> > The authors also highlight broader applicability; the authors derived similar results for standard training? Or has this been done in previous work with other frameworks?
>
> We did not apply the proposed framework to standard training because this has been done in previous studies using other mean field-based approaches. However, previous approaches are not applicable to adversarial training, which is why we developed a new framework.
>
> By "broader applicability" we mean that the utility of Theorem 4.1 is not limited to adversarial training. Theorem 4.1 provides a simple view of a ReLU-like network that should be useful for analyzing other training methods (e.g., contrastive learning).
>
> > In l261, how transfer attacks are relevant?
>
> A similar question was raised by Reviewer QtQE. Please kindly refer to our reply to Revewer QtQE (due to character limitation).

---

> > ### Comment · Reviewer_CfQ9 · 2023-08-14
> > **Thanks for the response**
> >
> > I appreciate the authors' clarifications. I have no follow-up questions and given the other positive reviews, my rating remains positive, as well.

---

### Official Review · Reviewer_55dm · 2023-07-26

**Soundness:** 3 good
**Presentation:** 3 good
**Contribution:** 2 fair
**Rating:** 6
**Confidence:** 2

**Summary:**

The paper provides a mean field analysis on relu networks for adversarial training. The main insight is that networks without residual connections are not most likely inevitably suffer from gradient explosion or vanishing and thus are not adversarially trainable, unlike vanilla network training.

**Strengths:**

Analyzing adversarial training performance is an important problem both theoretically.

The insight from the analysis that adversarial training vanilla relu networks is more likely to suffer from gradient explosion/vanishing is an interesting insight.

**Weaknesses:**

The verification of the theorems might need more effort. E.g., Figure 4 is showing accuracy of vanilla network, it would be helpful to also show the curves for residual networks.



**Questions:**

Could the authors elaborate a bit on why residual connections could make a bigger difference in adversarial training than vanilla training?

**Limitations:**

Limitations are well discussed.

---

> ### Author Rebuttal · Authors · 2023-08-08
>
> We would like to appreciate your fruitful suggestion and question.
>
> > The verification of the theorems might need more effort. E.g., Figure 4 is showing accuracy of vanilla network, it would be helpful to also show the curves for residual networks.
>
> The training accuracy of adversarial training in residual networks is presented in Figure A19. Due to the complexity of the heatmap design, we believe that combining them into a single image would be challenging. In the updated version, we plan to clarify in the caption of Figure 4 that the results for residual networks can be found in Figure A19.
>
> > Could the authors elaborate a bit on why residual connections could make a bigger difference in adversarial training than vanilla training?
>
> This is because adversarial training has a strong weight regularization effect (cf. Theorem 5.3), while vanilla training (standard training) does not. As adversarial training progresses, the network weights become smaller. Therefore, without residual connections, the small weights lead to gradient vanishing as the input signals diminish during the forward pass. Note that network weights also decay during standard training with L2 regularization. However, adversarial training causes the weights to decay at a significantly faster rate than L2 regularization (cf. Section H in Appendix).

---

### Official Review · Reviewer_e8jY · 2023-07-26

**Soundness:** 4 excellent
**Presentation:** 4 excellent
**Contribution:** 4 excellent
**Rating:** 7
**Confidence:** 3

**Summary:**

The theoretical understanding of adversarial training is an important and valuable topic. This work proposes a new theoretical framework for this based on mean field theory. With the proposed framework, the authors analyze the properties of adversarial training from multiple aspects, including the upper bounds of adversarial loss, the time evolution of weight variance, the adversarial trainable conditions, and the degradation of network capacity. These results could be helpful for the understanding of adversarial training and inspire more efforts on this topic.

**Strengths:**

1. It proposes a new framework to analyze adversarial training theoretically based on mean field theory.
2. Based on the proposed framework, it presents several theoretical results for adversarial training.
3. The proposed framework and the presented theoretical results are non-trivial and helpful for the understanding of adversarial training.

**Weaknesses:**

It studies several different adversarial training characteristics in the main paper. Is there any correlation between these different characteristics？ Why do we choose these aspects for analysis? Further, is it possible to provide a global diagram to better see which properties can be analyzed and which cannot be analyzed at present based on the proposed framework?

**Questions:**

I am not an expert on mean field theory and didn't check all the proofs. I give my rating by considering that 1) the theoretical understanding of adversarial training is important and the progress on this is very helpful for the community and 2) the proposed framework seems to be generic and may inspire more work in the future.

**Limitations:**

yes

---

> ### Author Rebuttal · Authors · 2023-08-08
>
> We would like to thank your insightful comments.
>
> > It studies several different adversarial training characteristics in the main paper. Is there any correlation between these different characteristics？ Why do we choose these aspects for analysis?
>
> Our theoretical results on adversarial training can be divided into two categories: those related to training dynamics (Theorems 5.6 and 5.7) and those related to test time performance (Theorem 5.8). Both are fundamental aspects of machine learning. More specifically, Theorems 5.6 and 5.7 answer why vanilla networks cannot achieve high training accuracy in some situations, while residual networks always can. Theorem 5.8 addresses the degradation of network capacity, which is tied to the generalization performance of networks, i.e., test accuracy. Similar studies have been conducted on standard training [1,2]. However, the untrainability of vanilla networks and the loss of capacity are unique to adversarial training. Thus, our choice of analysis allows us to highlight the significant differences between adversarial and standard training.
>
> Theorems 5.1 and 5.3 present the upper bounds of the adversarial loss and the time evolution of weight variance, respectively. While they were originally formulated to derive Theorems 5.6-5.8, they are also interesting results in their own right.
>
> [1] Deep information propagation, ICLR17.
> [2] Universal statistics of Fisher information in deep neural networks: Mean field approach, AISTATS19.
>
> > Further, is it possible to provide a global diagram to better see which properties can be analyzed and which cannot be analyzed at present based on the proposed framework?
>
> The proposed framework can analyze the following properties of adversarial training: (i) the upper bounds of the adversarial loss, (ii) time evolution of weight variance, (iii) adversarial trainability, and (iv) degradation of network capacity. Moreover, we believe that our approach can be applied to other deep learning methods besides adversarial training (cf. Section 4.2).
>
> A major limitation of our analysis, which is common to all mean field-based analyses, is its theoretical restriction to the early stages of training, making it not directly applicable to full training process. Nevertheless, mean field-based analyses offer superior flexibility compared to other approaches; they can handle multi-layer perceptrons (including shortcuts) with nonlinear activations without imposing any assumptions on the data distribution. We have discussed this in Section 7, but will strive to make it clearer in the updated version.
>
> Note that it has been observed empirically that the theoretical results from the early stages of training align well with fully trained networks [3,4]. In this study, for example, our theoretical prediction in Theorems 5.8 and F.16 - network width strongly influences generalization performance in adversarial training - holds even for fully trained networks (cf. Tables A7 and A8). Therefore, even though our framework is not strictly applicable to fully trained networks, we believe that it can still provide important insights for fully trained networks.
>
> [3] First-order adversarial vulnerability of neural networks and input dimension, ICML19.
> [4] Adversarial robustness guarantees for random deep neural, ICML21.

---

### Author Rebuttal · Authors · 2023-08-08

We appreciate the reviewers' critical reading, constructive comments, and overall positive scores. We have carefully taken into account all the comments and questions. Please kindly refer to the response to each reviewer. If our answers require further explanation or clarification, we are more than willing to provide them.

---

### Decision · Program_Chairs · 2023-09-21

**Decision:**

Accept (spotlight)

**Comment:**

All reviewers agree that the paper has made a good theoretical contribution on understanding adversarial training and keep positive opinions on the paper. I therefore recommend acceptance of the paper. Please revise the paper accordingly according to the reviewers' comments in the camera-ready version.